# STATE SPACE MODELS ARE PROVABLY COMPARABLE TO TRANSFORMERS IN DYNAMIC TOKEN SELECTION

**Naoki Nishikawa**
The University of Tokyo
RIKEN AIP
nishikawa-naoki259@g.ecc.u-tokyo.ac.jp

**Taiji Suzuki**
The University of Tokyo
RIKEN AIP
taiji@mist.i.u-tokyo.ac.jp

## ABSTRACT

Deep neural networks based on state space models (SSMs) are attracting significant attention in sequence modeling since their computational cost is much smaller than that of Transformers. While the capabilities of SSMs have been demonstrated through experiments in various tasks, theoretical understanding of SSMs is still limited. In particular, most theoretical studies discuss the capabilities of SSM layers without nonlinear layers, and there is a lack of discussion on their combination with nonlinear layers. In this paper, we explore the capabilities of SSMs combined with fully connected neural networks, and show that they are comparable to Transformers in extracting the essential tokens depending on the input. As concrete examples, we consider two synthetic tasks, which are challenging for a single SSM layer, and demonstrate that SSMs combined with nonlinear layers can efficiently solve these tasks. Furthermore, we study the nonparametric regression task, and prove that the ability of SSMs is equivalent to that of Transformers in estimating functions belonging to a certain class.

## 1 INTRODUCTION

Foundation models based on Transformers have achieved remarkable success in various sequence modeling tasks such as natural language processing (Vaswani et al., 2017), computer vision (Dosovitskiy et al., 2020), and speech recognition (Radford et al., 2023). The superior performance of Transformers is attributed to the self-attention mechanism, which enables the model to aggregate information from the input sequence.

In contrast to its success, the self-attention mechanism has a potential problem that it requires a large amount of computation and memory. To deal with this issue, many studies have attempted to develop efficient models that can replace Transformers. Among them, *State Space Models* (SSMs) have garnered considerable interest recently. One advantage of SSMs is that the output can be computed with a significantly small time using convolution via the FFT algorithm or recursive computation. Based on the original SSMs, many improvements have been proposed, such as HiPPO-based intialization (Gu et al., 2021) and architectures using gated convolutions (Fu et al., 2022; Poli et al., 2023).

Networks based on SSMs have achieved high performance in various applications such as gene analysis (Nguyen et al., 2024), audio generation (Goel et al., 2022) and speech recognition (Saon et al., 2023). On the other hand, some of the recent studies pointed out the limitations of SSMs to solve tasks. For example, Merrill et al. (2024) show that SSMs cannot solve sequential problems from the view of computational complexity theory. Additionally, Jelassi et al. (2024) demonstrate that SSMs are inferior to Transformers in solving the task to copy the input sequence.

One of the major differences between SSMs and Transformers lies in how they aggregate information from the input sequence tokens. The output of Transformers is computed as a weighted sum of the input tokens, where the weights are determined depending on the input. This allows the Transformer to *dynamically determine which tokens to extract based on the input*, leading to its high performance. SSMs also compute their output as a weighted sum of the input tokens, but the *weights do not depend on the input*. Therefore, a single SSM layer cannot perform dynamic token extraction, which limits its capability.

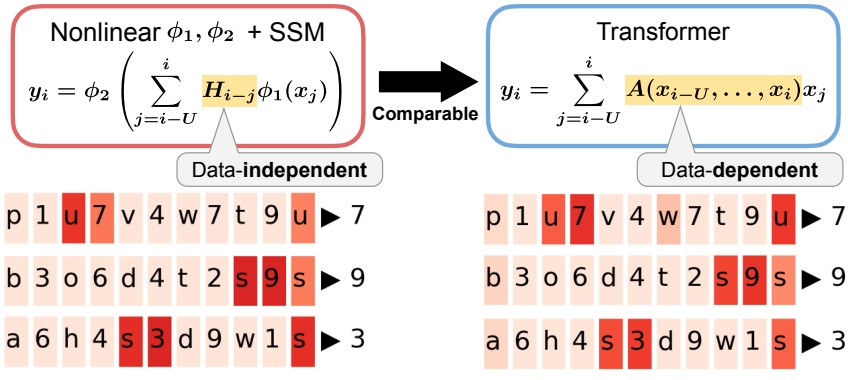

Figure 1.1: Conceptual illustrations of our theory. The abilities of SSMs are said to be limited since their filter is not data-dependent. However, when combined with nonlinear layers, SSMs are comparable to Transformers in terms of dynamic token selection. Indeed, experiments on associative recall tasks show that SSMs capture the important tokens in the sequence depending on the input, which is similar to the behavior of Transformers. The heatmap in the figure represents the importance of the token when the model predicts the output. Note that these are not artificial figures, but the actual results of the experiments.

However, in typical architectures, SSMs are repeatedly applied alternately with fully connected neural networks (FNNs), similar to the attention mechanism. This raises the following question:

*Can SSMs combined with FNN layers perform dynamic token selection similar to Transformers?*

This paper provides a positive theoretical result for this question. Specifically, we demonstrate that SSMs, when combined with FNNs, can exhibit dynamic token selection with performance equivalent to Transformers.

To demonstrate the claim, we consider two synthetic tasks (input copying and associative recall), and a non-parametric regression problem. *Input copying* is the task of generating the same sequence as the given input. Jelassi et al. (2024) intensively studied this task and showed that a single SSM layer underperforms compared to Transformers in solving this task. In this paper, we show that two-layer SSMs combined with FNN layers can achieve performance comparable to Transformers in this task. In *associative recall*, the models are required to infer the answer from a pair of words provided as input. We demonstrate that the performance of SSMs combined with FNN layers is better than that of SSMs without FNNs shown in Massaroli et al. (2024). As for the *non-parametric regression*, we consider the estimation of piecewise $\gamma$-smooth functions, which is defined in Takakura & Suzuki (2023). Takakura & Suzuki (2023) shows that Transformers can efficiently estimate functions in this class. We show that SSMs combined with FNN layers can achieve the same convergence rate as the rate for Transformers shown in Takakura & Suzuki (2023).

In solving the three problems above, the models have to determine which tokens to extract based on the input data. Therefore, these results imply that SSMs possess dynamic token extraction capabilities comparable to Transformers. We give some examples of associative recall task in Figure 1.1. We trained SSMs and Transformers on the task and draw heatmaps to show which tokens the trained models focus on. From the figures, we can observe that both models pay attention to similar parts of the input. This verifies that SSMs possess dynamic feature extraction capabilities comparable to Transformers. See Section 3.2 for more details of the associative recall task.

The contributions of this paper are summarized as follows:

1. We theoretically study the abilities of SSMs to solve two artificial tasks, input copying and associative recall, and prove that SSMs + FNNs can solve these tasks efficiently (Section 3.1, 3.2).

2. To prove the above results, we provide a theoretical result that shows SSMs combined with FNNs can mimic the dynamic token selection mechanism of Transformers (Section 3.3).

3. As a more general example, we also consider the non-parametric regression for the function class defined in Takakura & Suzuki (2023), and demonstrate that SSMs can achieve the same estimation error as Transformers (Section 4).

**Other related works.** Some studies have theoretically investigated the abilities of SSMs recently. For instance, Wang & Xue (2023) show that SSMs are universal approximators for continuous sequence-to-sequence functions. Moreover, Cirone et al. (2024) studied the abilities of SSMs using rough path theory. Furthermore, Alonso et al. (2024) analyzed the abilities of SSMs with the use of the control theory. However, these studies (i) do not give quantitative evaluations comparing SSMs to Transformers, and/or (ii) only consider SSM layers without nonlinear layers.

There are some previous studies for the estimation error bound for non-parametric regression, for example, Suzuki (2018); Schmidt-Hieber (2020); Suzuki & Nitanda (2021) for FNNs and Okumoto & Suzuki (2021) for CNNs. These studies do not consider the case where the positions of essential features change depending on the input. The function classes with piecewise smoothness are also considered in Petersen & Voigtlaender (2018) and Imaizumi & Fukumizu (2019). They do not consider anisotropic smoothness or sequence inputs, whereas we consider such situations.

**Notations.** For $l, r \in \mathbb{Z}$ ($l \leq r$), let $[l]$ be the set $\{1, \ldots, l\}$, and $[l : r]$ be the set $\{l, \ldots, r\}$. For a set $S \subseteq \mathbb{R}$ and $d, V \in \mathbb{N}$, let $S^{d \times [-V:0]} := \left\{ [s_{-V}, \ldots, s_0] \mid s_i \in S^d \right\}$ and let $S^{d \times \infty} := \left\{ [\ldots, s_{-2}, s_{-1}, s_0] \mid s_i \in S^d \right\}$. For $F : \Omega \to \mathbb{R}^l$, let $\|F\|_\infty := \sup_{X \in \Omega} \|F(X)\|_\infty$. For a matrix $A$, let $\|A\|_0 = |\{(i, j) \mid A_{ij} \neq 0\}|$. For $X \in \mathbb{R}^{d \times \infty}$, $X_{i,:} \in \mathbb{R}^{1 \times \infty}$ represents its $i$-th row.

## 2 THE DEFINITION OF DEEP NEURAL NETWORKS WITH SSMS

In this section, we provide the formal definition of deep neural networks with SSMs. State space models with the input $[u_t]_{t=-L}^0$, the latent vectors $[x_t]_{t=-L}^0$ and the output $[y_t]_{t=-L}^0$ ($u_t \in \mathbb{R}$, $x_t \in \mathbb{R}$, $y_t \in \mathbb{R}$), are represented as follows:

$$x_{t+1} = \mathsf{A}x_t + \mathsf{B}u_t, \quad y_t = \mathsf{C}x_t + \mathsf{D}u_t \quad (t = -L, \ldots, -1),$$

where $\mathsf{A}, \mathsf{B}, \mathsf{C}, \mathsf{D} \in \mathbb{R}$ are learnable parameters. Then, the output $y_t$ can be written explicitly as $y_t = \sum_{n=-L}^t \left( \mathsf{C}\mathsf{A}^{t-n}\mathsf{B} + \mathsf{D}\delta_{t-n} \right) u_n$. By setting $h_t := \mathsf{C}\mathsf{A}^t\mathsf{B} + \mathsf{D}\delta_t$ and $h = [h_t]_{t=0}^L$, we can rewrite the output as $y_t = (h * u)_t := \sum_{n=-L}^t h_{t-n} u_n$ using convolution operation $*$. If the *filter* $[h_t]_{t=0}^L$ is precomputed, the output can be computed with $O(L \log L)$ time complexity using FFT algorithm, which is much faster than the computation cost of Transformers, $O(L^2)$.

In this paper, we consider the architectures consisting of the following three types of layers: (i) FNN layers, (ii) convolution layers, and (iii) an embedding layer.

**(i) FNN layer** An FNN with depth $L$ and width $W$ is defined as

$$f(x) := (A_L \eta(\cdot) + b_L) \circ \cdots \circ (A_1 x + b_1),$$

where $\eta = \mathrm{ReLU}$, and $A_i \in \mathbb{R}^{d_{i+1} \times d_i}, b_i \in \mathbb{R}^{d_{i+1}}$ with $\max_i d_i \leq W$. Then, we define the class of FNN with depth $L$, width $W$, norm bound $B$ and sparsity $S$ by

$$\Psi(L, W, S, B) := \left\{ f \mid \max_i \left\{ \|A_i\|_\infty, \|b_i\|_\infty \right\} \leq B, \sum_{i=1}^L \|A_i\|_0 + \|b_i\|_0 \leq S \right\}.$$

**(ii) Convolution layer** Next, we define the convolution layers. Let $W \in \mathbb{R}^{D \times D}$ be learnable weights, and $D$ be the embedding dimension. Then, given the input $X \in \mathbb{R}^{D \times \infty}$, the output of the convolution layer $g : \mathbb{R}^{D \times \infty} \to \mathbb{R}^{D \times \infty}$ with window size $U$ is computed as

$$g(X) := H * (WX), \quad H_{k,j} := c_{1,k} \cos\left( \frac{2\pi j \cdot a_{1,k}}{U + 1} \right) + c_{2,k} \sin\left( \frac{2\pi j \cdot a_{2,k}}{U + 1} \right),$$

where $H \in \mathbb{R}^{D \times (U+1)}$ is a filter controlled by learnable parameters $c_1, c_2, a_1, a_2 \in \mathbb{R}^D$. The operator $* : \mathbb{R}^{D \times (U+1)} \times \mathbb{R}^{D \times \infty} \to \mathbb{R}^{D \times \infty}$ represents element-wise convolution. Note that we assume the finite window size $U$, i.e., the output of position $-i$ ($i \in \mathbb{N}_{\geq 0}$) is computed with the tokens at position $-i, \ldots, -i - U$. Such setting is also considered in the analysis of Transformers (Takakura & Suzuki, 2023). Then, we define the class of convolution layers with window size $U$, embedding dimension $D$ and norm constraint $B$ by

$$\mathcal{C}(U, D, B) := \{ g \mid \max \left\{ \|W\|_\infty, \|a_1\|_\infty, \|a_2\|_\infty, \|c_1\|_\infty, \|c_2\|_\infty \right\} \leq B \}.$$

The definition of the convolution filter includes several important settings. First, our setting includes the case where we use the convolution filter $h_t = \mathsf{C}\mathsf{A}^t\mathsf{B} + \mathsf{D}\delta_t$ of ordinary SSMs. Indeed, by

constructing the parameters A, B, C, D in the filter appropriately, we can obtain the same architecture as the convolution layer we consider (see Appendix A for the details). Moreover, our setting includes the filter used in Hyena (Poli et al., 2023), in which the filter is defined by neural networks with $\sin$-activation.

**(iii) Embedding layer**    Finally, the embedding layer with embedding dimension $D$ is defined as
$$\mathrm{Emb}(X) = [E_1 X_i + E_2]_{i=-\infty}^{\infty},$$
where $E_1 \in \mathbb{R}^{D \times d}$ and $E_2 \in \mathbb{R}^{D}$ are learnable parameters.

We first feed the sequence into $\mathrm{Emb}$, and then alternately apply convolution layers and FNN layers. Note that the FNN layers are applied in a token-wise manner. Thus, we define a class of deep neural networks using SSMs, denoted as $\mathcal{S}$, as follows:

$$\mathcal{S}(M, U, D, L, W, S, B) := \left\{ f_M \circ g_M \circ \cdots \circ f_1 \circ g_1 \circ \mathrm{Emb} \left| \begin{array}{l} f_i \in \Psi(L, W, S, B), \\ g_i \in \mathcal{C}(U, D, B), \\ \|E_1\|_\infty \le B, \ \|E_2\|_\infty \le B. \end{array} \right. \right\}.$$

We remark that the number of parameters of the model is $O(M(LW^2 + D^2))$, and does not depend on the window size $U$, since the parameters are shared among the tokens.

## 3    Synthetic Tasks: Input Copying and Associative Recall

In this section, we consider the two tasks of (i) input copying and (ii) associative recall, and show that SSMs combined with FNNs can solve these tasks efficiently. In both of these tasks, the position of the important token is different for each input. Therefore, the fact that SSMs can solve these tasks suggests that SSMs have *dynamic token extraction capabilities* comparable to those of Transformers.

Throughout this section, we assume that an input is given as a sequence of words in the dictionary $\mathcal{W}$, where $\mathcal{W}$ is a finite set. Each word in the sequence is first converted into a $|\mathcal{W}|$-dimensional one-hot vector. Then, the tokens are transformed to $D$-dimensional vectors by the embedding layer, and fed into subsequent convolution layers and FNN layers. Thus, we obtain a sequence of $D$-dimensional tokens as a model's output. We consider the setting where the model generates sequences by autoregressively outputting the words, i.e., next-token prediction. To do this, we introduce an additional decoding layer, which we denote by $\mathrm{Dec}$. This layer has a learnable parameter $W_{\mathrm{Dec}} \in \mathbb{R}^{D \times |\mathcal{W}|}$, and linearly transforms the final token of the model's output into a vector in $\mathbb{R}^{|\mathcal{W}|}$. Then, the word corresponding to the largest component of this vector is regarded as the model's prediction.

### 3.1    Input Copying

The input copying is a task in which the model is required to output exactly the same sequence as the input via autoregressive inference. As an example, consider the situation where the model receives the input sequence "⟨BOS⟩ c a d b e ⟨COPY⟩", where a, b, c, d, e are the words in $\mathcal{W}$, and ⟨BOS⟩ and ⟨COPY⟩ are special tokens, which are also included in $\mathcal{W}$. Then, the model first needs to output "c". Next, the model receives the input "⟨BOS⟩ c a d b e ⟨COPY⟩ c", and the model is required to generate "a". This process is repeated until the number of tokens the model generates becomes equal to the number of words in the input sequence.

To evaluate the model for this task, we consider the probability that the model generates the correct sequence for a certain input distribution. More precisely, suppose that input sequences are given in the form of "⟨BOS⟩ $x_1 x_2 \cdots x_V$ ⟨COPY⟩", where $x_1, x_2, \ldots, x_V$ are independently generated from the uniform distribution on $\mathcal{W} \setminus \{⟨BOS⟩, ⟨COPY⟩\}$. Then, we set $y_1, \ldots, y_V$ as the sequence generated by the model, and evaluate the model by the metric $\mathrm{err}_V$ defined as follows, which measures the probability that the model does not correctly copy the input sequence:

$$\mathrm{err}_V := \mathbb{P}[(y_1, \ldots, y_V) \ne (x_1 \ldots, x_V)].$$

Jelassi et al. (2024) intensively studied the capabilities of SSMs and Transformers to solve this task. They theoretically showed that Transformers with $O(\log(V/\epsilon) \log|\mathcal{W}|)$ parameters can achieve $\mathrm{err}_V \le \epsilon$. In their proof, they leverage the dynamic token extraction ability of the attention mechanism. More specifically, they demonstrate that Transformers can solve the copying task by looking up the $n$ sequential tokens that match the last $n$ tokens. In addition, they discuss the lower bound of the accuracy of SSMs, and showed that SSMs need $O(L \log|\mathcal{W}|)$ memory size to achieve

$\text{err}_V \leq 1/2$. These results suggest that a single SSM layer is inferior to Transformers in terms of the dynamic token extraction abilities.

In contrast to these results, we investigate the case where nonlinear layers are placed before and after the SSM layer, and obtain the following result.

**Theorem 3.1.** *For any $\epsilon > 0$, there exists an SSM $\hat{F} \in \mathcal{S}(M, U, D, L, W, S, B)$ with*

$$M = 2, \quad U = V, \quad D, \ L, \ W, \ S, \ \log B \lesssim \log^{37} \epsilon^{-1} \log^{42} V \log^8 |\mathcal{W}|,$$

*and decoding layer Dec with $\|W_{\text{Dec}}\|_\infty \leq 1$ such that $\sup_{V' \in [V]} \text{err}_{V'} \leq \epsilon$.*

This result shows that, two layer SSMs combined with FNNs, we can achieve the error $\epsilon$ with $\text{poly} \log(\epsilon^{-1}, V, |\mathcal{W}|)$ parameters, and avoid the necessity of the memory size increasing linearly with $V$.

In the proof of this theorem, it is essential that there are FNN layers before and after the second SSM layer. As we will show later in Lemma 3.3, the SSM combined with the preceding and following FNN layers has the ability to extract tokens based on the inner product of keys and queries, similar to a Transformer. Jelassi et al. (2024) demonstrates that a two-layer Transformer can solve the input copying task. Thus, by replacing the attention layer with FNN + SSM + FNN, the above theorem can be proved. The detailed proof can be found in Appendix H.

### 3.2 ASSOCIATIVE RECALL

Next, we investigate the task called associative recall. In this task, we assume that the set of words $\mathcal{W}$ is divided into two disjoint sets $\mathcal{W}_{\text{key}}$ and $\mathcal{W}_{\text{value}}$. The input sequence is given in the form of "$k_1$ $v_1 \cdots k_V \ v_V \ q$", where $k_1, \ldots, k_V \in \mathcal{W}_{\text{key}}$ ($k_i \neq k_j$ if $i \neq j$), $v_1, \ldots, v_V \in \mathcal{W}_{\text{value}}$ and $q \in \mathcal{W}_{\text{key}}$ matches one of the $k_i$ ($i \in [V]$). Then, the model is required to output the corresponding $v_i$ for the given $q = k_i$. For example, if the model receives the input "c 2 a 5 d 1 b 4 a", the model needs to output "5".

Similarly to the input copying task, this task also requires the model to dynamically extract the important tokens. Indeed, to solve this task, the model needs to focus on three tokens: (i) the last token of the input, (ii) the token with the same word as the last one, and (iii) the token that follows (ii). Since the locations of (ii) and (iii) change depending on (i), the model needs to pay attention to different locations depending on the input.

We proved the following theorem, which shows that $O(\text{poly} \log(|\mathcal{W}|))$ parameters are sufficient to solve the associative recall task when using SSMs combined with FNNs.

**Theorem 3.2.** *There exists an SSM $\hat{F} \in \mathcal{S}(M, U, D, L, W, S, B)$ with*

$$M = 2, \quad U = 2V, \quad D, \ L, \ W, \ S, \ \log B \lesssim \log^{13}(|\mathcal{W}|),$$

*and decoding layer Dec with $\|W_{\text{Dec}}\|_\infty \leq 1$ such that, for any input sequences of associative recall task, the model generates the correct output.*

Massaroli et al. (2024) showed that Hyena-based SSMs can solve the associative recall task with $O(\sqrt{|\mathcal{W}|} \log^2 |\mathcal{W}|)$ parameters. In contrast, we proved that we require only $O(\text{poly} \log |\mathcal{W}|)$ parameters, thanks to the nonlinear layers. This indicates that combining SSMs with nonlinear layers can improve the dynamic token extraction ability of SSMs.

In this theorem as well, the essence of the proof is the ability of the SSM with preceding and following FNN layers, as described in Lemma 3.3. The proof can be found in Appendix H.

### 3.3 SSMs MIMIC ATTENTION MECHANISMS TO SELECT IMPORTANT TOKENS

In this subsection, we discuss why SSMs combined with FNNs can perform dynamic token selection similar to Transformers.

The dynamic token selection abilities of Transformers stem from the attention mechanisms. Indeed, attention mechanisms compute the weighted sum of the values (i.e., projected input tokens), where the weights are determined based on the input. The weights are computed by applying the softmax function to the inner product between keys and queries. This means that the attention mechanism *prioritizes the tokens where the key and query have a high inner product*. The following statement shows that SSMs combined with FNN layers can mimic this functionality.

**Lemma 3.3** (Dynamic Token Selection by SSMs). *Suppose that the input sequence $X$ is given by*

$$X = \begin{bmatrix} * & \cdots & * & q \\ k_{-V} & \cdots & k_{-1} & k_0 \\ v_{-V} & \cdots & v_{-1} & v_0 \end{bmatrix} \in \mathbb{R}^{(2d'+d)\times[-V:0]},$$

*where $q, k_j \in \mathbb{R}^{d'}$ and $v_j \in \mathbb{R}^d$ with $\|q\|_\infty \le 1, \|k_j\|_\infty \le 1, \|v_j\|_\infty \le 1$ $(j = -V, \ldots, 0)$. Let $\mu_j = q^\top k_j$ $(j = -V, \ldots, 0)$ and $j^* = \arg\max_{j=-V,\ldots,0} \mu_j$. Suppose that, for any $j \in [-V : 0] \setminus \{j^*\}$, it holds $\mu_j \le \mu_{j^*} - \delta$ for some $\delta > 0$. Then, for any $\epsilon > 0$, there exist FNN layers $f_1, f_2 \in \Psi(L, W, S, B)$ and a convolution layer $g \in \mathcal{C}(U, D, B)$ with*

$$U = V, \quad D = d'^3 \delta^{-2}\big(\log^2 \epsilon^{-1} + \log^2 V\big),$$

$$L \lesssim d'^8 \delta^{-5}\big(\log^5 \epsilon^{-1} + \log^5 V\big), \quad W \lesssim d'^3 \delta^{-3}\big(\log^3 \epsilon^{-1} + \log^3 V\big),$$

$$S \lesssim d'^8 \delta^{-5}\big(\log^5 \epsilon^{-1} + \log^5 V\big), \quad \log B \lesssim d'^3 \delta^{-2}\big(\log^3 \epsilon^{-1} + \log^3 V\big),$$

*such that $\|y_0 - v_{j^*}\|_\infty \le \epsilon$, where $[y_{-V}, \ldots, y_{-1}, y_0] := f_2 \circ g \circ f_1(X)$.*

We provide the proof in Appendix G. From this theorem, we can see that SSMs preceded and followed by FNN layers can dynamically change the positions of tokens to focus on based on the value of the inner product between the key and query, similar to the attention mechanism. Thus, SSMs demonstrate *dynamic token selection abilities* similar to Transformers.

## 4 NONPARAMETRIC REGRESSION PROBLEM

In this section, we consider a non-parametric regression problem with sequence inputs, and show that SSMs are comparable to Transformers in estimating piecewise $\gamma$-smooth functions. As we will explain later, in functions belonging to this class, the positions of important tokens vary depending on the input. Therefore, the dynamic token selection ability is essential for estimating a piecewise $\gamma$-smooth function. In Appendix E, we provide a visual explanation of the motivation for considering this class of functions.

Due to the technical convenience to analyze the estimation error, we consider the setting where the output of the network is bounded. For this purpose, we assume that the output of the model is fed into the function $\text{clip}_R$ defined by $\text{clip}_R(x) := \max\{-R, \min\{R, x\}\}$. Since $\text{clip}_R$ can be implemented by the FNN with depth 1 and width 2, this assumption is not far from the practical setting. Furthermore, to predict a real value, the final token (i.e., index 0) in the sequence output by the model is regarded as the predicted value. We define the class of clipped networks $\mathcal{S}'$ by

$$\mathcal{S}'(M, U, D, L, W, S, B) = \{(\text{clip}_R \circ F)_0 \mid F \in \mathcal{S}(M, U, D, L, W, S, B)\}.$$

### 4.1 PROBLEM SETTING

We consider the situation where the input $X := [x_i]_{i=-\infty}^0 \in \mathbb{R}^{d\times\infty}$ is a sequence of $d$-dimensional tokens, and they are generated from a probability measure $P_X$ on $([0,1]^{d\times\infty}, \mathcal{B}([0,1]^{d\times\infty}))$. In the following, we denote $\Omega := \text{supp}\, P_X$ and define the norm $\|\cdot\|_{p,P_X}$ by $\|f\|_{p,P_X} = \left(\int_\Omega \|f(X)\|_p^p \, \mathrm{d}P_X\right)^{1/p}$.

As in the usual nonparametric regression setting, suppose that we observe $n$ i.i.d. inputs $X^{(i)} \sim P_X$ $(i = 1, \ldots, n)$ and the corresponding outputs $Y^{(i)} \in \mathbb{R}$ generated by $Y^{(i)} = F^\circ(X^{(i)}) + \xi^{(i)}$, where $\xi^{(i)} \in \mathbb{R}$ is the i.i.d. noise generated from $\mathcal{N}(0, \sigma^2)$ $(\sigma > 0)$. We further assume that $\{\xi^{(i)}\}_{i=1}^n$ is independent of the inputs $\{X^{(i)}\}_{i=1}^n$. Given the pairs of inputs and outputs $\{(X^{(i)}, Y^{(i)})\}_{i=1}^n$, we obtain the estimator $\hat{F}$ of the target function $F$ through empirical risk minimization:

$$\hat{F} := \arg\min_{F \in \mathcal{S}'} \frac{1}{n}\sum_{i=1}^n \left(Y^{(i)} - F(X^{(i)})\right)^2,$$

where $\mathcal{S}'$ is the class of networks that we defined above. To measure the statistical performance of the estimator $\hat{F}$, we utilize mean squared error (MSE) defined by

$$R(\hat{F}, F^\circ) = \mathbb{E}\left[\left\|\hat{F}(X) - F^\circ(X)\right\|_{2,P_X}^2\right],$$

where the expectation is taken for $\left\{X^{(i)}\right\}_{i=1}^n$ and $\left\{\xi^{(i)}\right\}_{i=1}^n$.

## 4.2 PIECEWISE $\gamma$-SMOOTH FUNCTION CLASS

To compare the estimation ability of SSMs with that of Transformers, we assume that the target function $F^\circ$ belongs to the function class called *piecewise $\gamma$-smooth*. The function class was introduced in Takakura & Suzuki (2023), and they showed the estimation error bound of Transformers for the functions in the class. In this function class, *the importance of the tokens (or coordinates) is characterized by the smoothness of the function*. We describe the details in the following.

**$\gamma$-smooth function class.** Before introducing the piecewise $\gamma$-smooth function class, we first define the $\gamma$-smooth function class, which was first proposed by Okumoto & Suzuki (2021). This function class can be seen as an *extension of the well-known function spaces*, such as the mixed-Besov space and (anisotropic) Sobolev space.

First, for $r \in \mathbb{Z}_0^{d \times \infty}$, we define $\psi_{r_{ij}} : [0,1] \to \mathbb{R}$ by $\psi_{r_{ij}}(x) := \begin{cases} \sqrt{2}\cos(2\pi|r_{ij}|x) & (r_{ij} \le 0), \\ \sqrt{2}\sin(2\pi|r_{ij}|x) & (r_{ij} > 0), \end{cases}$

and $\psi_r : [0,1]^{d \times \infty} \to \mathbb{R}$ by $\psi_r(X) = \prod_{i=1}\prod_{j=1}\psi_{r_{ij}}(X_{ij})$. Then, $\{\psi_r\}_{r \in \mathbb{Z}_0^{d \times \infty}}$ forms a complete orthonormal system of $L^2([0,1]^{d \times \infty})$, Therefore, any $f \in L^2([0,1]^{d \times \infty})$ can be expanded as $f = \sum_{r \in \mathbb{Z}_0^{d \times \infty}} \langle f, \psi_r \rangle \psi_r$. For $s \in \mathbb{N}_0^{d \times \infty}$, we define

$$\delta_s(f) := \sum_{r \in \mathbb{Z}_0^{d \times \infty}, \lfloor 2^{s_{ij}-1} \rfloor \le r_{ij} < 2^{s_{ij}}} \langle f, \psi_r \rangle \psi_r.$$

Then, we define the $\gamma$-smooth function class as follows.

**Definition 4.1** ($\gamma$-smooth function class). For a given $\gamma : \mathbb{N}_0^{d \times \infty} \to \mathbb{R}$ which is monotonically non-decreasing with respect to each coordinate and $p \ge 2, \theta \ge 1$, we define the $\gamma$-smooth function space as follows:

$$\mathcal{F}_{p,\theta}^\gamma([0,1]^{d \times \infty}) := \left\{ f \in L^2([0,1]^{d \times \infty}) \mid \|f\|_{\mathcal{F}_{p,\theta}^\gamma} < \infty \right\},$$

where the norm $\|f\|_{\mathcal{F}_{p,\theta}^\gamma}$ is defined as $\|f\|_{\mathcal{F}_{p,\theta}^\gamma} := \left( \sum_{s \in \mathbb{N}_0^{d \times \infty}} 2^{\theta\gamma(s)} \|\delta_s(f)\|_{p,P_X}^\theta \right)^{1/\theta}$. We also define the finite dimensional version $\mathcal{F}_{p,\theta}^\gamma([0,1]^{d \times l})$ for $l \in \mathbb{N}$ in the same way.

Note that $\delta_s(f)$ can be seen as the frequency component of $f$ with frequency $|r_{ij}| \sim 2^{s_{ij}}$ for each coordinate $(i,j)$. Therefore, we can interpret that $\gamma$ controls the amplitude of each frequency component through weighting the term $\|\delta_s(f)\|_{p,P_X}$ in the definition of the norm $\|\cdot\|_{\mathcal{F}_{p,\theta}^\gamma}$. In other words, if $\gamma(s)$ is larger, the norm of frequency component $\delta_s(f)$ is smaller.

As a special case of $\gamma$, we consider the following two types of smoothness:

$$\gamma(s) = \begin{cases} \langle a, s \rangle & \text{(Mixed smoothness)}, \\ \max\{a_{ij}s_{ij} \mid i \in [d], j \in \mathbb{Z}\} & \text{(Anisotropic smoothness)}, \end{cases}$$

where $a \in \mathbb{R}_{>0}^{d \times \infty}$ is the *smoothness parameter*, which determines the smoothness of the function for each coordinate. To provide the intuition of the smoothness, let us consider the extreme case, $a_{ij} \to \infty$ for $(i,j)$ with $s_{ij} \ne 0$. Then, it holds $\gamma(s) \to \infty$, both for mixed and anisotropic smoothness, which implies $2^{\gamma(s)} \to \infty$. This indicates that a strong "penalty" is imposed on the component $\delta_s(f)$ and the function $f$ does not have the frequency component $s_{ij}$ along the direction of $(i,j)$. Since the norm $\|f\|_{\mathcal{F}_{p,\theta}^\gamma}$ has to be finite, it holds $\|\delta_s(f)\|_{p,P_X} \to 0$. This means that the function $f$ has to be smooth for the input coordinate $(i,j)$. Therefore, large $a_{ij}$ implies that the function is smooth towards the coordinate $(i,j)$, and this indicates that the value of function does not change much for the input coordinate $(i,j)$, which means that $X_{ij}$ is *not an important feature*. In contrast, small $a_{ij}$ implies that the coordinate $X_{ij}$ is *an important feature*.

As we stated above, the function class $\mathcal{F}_{p,\theta}^\gamma([0,1]^{d \times \infty})$ can be seen as an extension of some well-known function spaces to the infinite-dimensional setting. Indeed, if $P_X$ is a uniform distribution on $[0,1]^{1 \times l}$ and $p < \infty$, then $\mathcal{F}_{p,\theta}^\gamma([0,1]^{1 \times l})$ with mixed smoothness is equivalent to the mixed-Besov space. Moreover, if $P_X$ is a uniform distribution, then the anisotropic Sobolev space is included in the unit ball of $\mathcal{F}_{2,2}^\gamma$ with anisotropic smoothness.

**Piecewise $\gamma$-smooth function class.** Now, we are ready to define the piecewise $\gamma$-smooth function class. The functions in this class have different smoothness depending on the input unlike $\gamma$-smooth functions. Therefore, the models have to *choose appropriate tokens to extract depending on the input* when estimating a function in this class.

The rigorous definition of piecewise $\gamma$-smooth function class is given as follows. In the following, we denote $X_{-i} := [\ldots, x_{-i-1}, x_{-i}]$ for $i \in \mathbb{N}$ and $X = [\ldots, x_{-1}, x_0]$.

**Definition 4.2** (Piecewise $\gamma$-smooth function class). Let $\mu$ be a function that belongs to $\mathcal{F}_{p,\theta}^{\gamma}([0,1]^{d \times \infty})$, which we call the *importance function*. Additionally, let $\pi_X : [-V : 0] \to [-V : 0]$ be the map that sorts the indices $i \in [-V : 0]$ in ascending order of the importance $\mu(X_i)$, i.e.,

$$\mu(X_{\pi_X(-V)}) < \cdots < \mu(X_{\pi_X(0)}).$$

Then, we define the map $\Pi : [0,1]^{d \times \infty} \to [0,1]^{d \times [-V:0]}$ by

$$\Pi(X) := [x_{\pi_X(-V)}, \ldots, x_{\pi_X(0)}].$$

Then, for $p \geq 2, \theta \geq 1$ and $\gamma : \mathbb{N}_0^{d \times \infty} \to \mathbb{R}$, the function class $\mathcal{P}_{p,\theta}^{\gamma}([0,1]^{d \times \infty})$ with piecewise $\gamma$-smoothness is defined as follows:

$$\mathcal{P}_{p,\theta}^{\gamma}([0,1]^{d \times \infty}) := \left\{ g = f \circ \Pi \mid f \in \mathcal{F}_{p,\theta}^{\gamma}([0,1]^{d \times [-V:0]}), \|g\|_{\mathcal{P}_{p,\theta}^{\gamma}} < \infty \right\},$$

where the norm $\|g\|_{\mathcal{P}_{p,\theta}^{\gamma}}$ is defined by $\|g\|_{\mathcal{P}_{p,\theta}^{\gamma}} := \left( \sum_{s \in \mathbb{N}_0^{d \times [-V:0]}} 2^{\theta \gamma(s)} \|\delta_s(f) \circ \Pi\|_{p,P_X}^{\theta} \right)^{1/\theta}$.

In simple terms, when an input $X = [\ldots, x_{-V}, \ldots, x_0]$ is fed into a function $g \in \mathcal{P}_{p,\theta}^{\gamma}([0,1]^{d \times \infty})$, the tokens $x_{-V}, \ldots, x_0$ are first sorted in ascending order of the importance $\mu(X_{-V}), \ldots, \mu(X_0)$, and the resulting sequence $[x_{\pi_X(-V)}, \ldots, x_{\pi_X(0)}]$ is fed into the $\gamma$-smooth function $f$. The importance of each token $x_{-i}$ is determined by the preceding tokens, i.e., $x_{-i}, x_{-i-1}, \ldots$. Since the order of the sorted tokens changes depending on the input, the smoothness of the function $g = f \circ \Pi \in \mathcal{G}_{p,\theta}^{\gamma}$ differs for different inputs.

In the definition, tokens with higher importance are placed closer to index 0 after sorting. In the latter subsection, we assume that the smoothness of the function $f$ for a coordinate is smaller for the tokens with indices closer to 0, which implies that the tokens with higher importance are more essential to estimate the function $f$.

As in Takakura & Suzuki (2023), we assume that an importance function $\mu$ is *well-separated*, i.e., for some constant $c, \beta > 0$, $\mu$ satisfies $\mu(X_{\pi_\lambda(-i)}) \geq \mu(X_{\pi_\lambda(-i-1)}) + ci^{-\beta}$ for any $X \in \Omega_\lambda$. This implies that $X$ satisfies $\mu(X_{-i}) \simeq \mu(X_{-j})$ ($i \neq j$) with probability zero. A similar assumption can be found in the literature of statistics such as Hall & Horowitz (2007).

### 4.3 Approximation and Estimation Ability of SSMs

In this subsection, we show the theoretical results on the ability of SSMs to approximate and estimate the function in the piecewise $\gamma$-smooth function class.

To establish the theories, we make the following assumptions, all of which are also imposed in Takakura & Suzuki (2023).

**Assumption 4.3.** *The true function $F^\circ$ belongs to $\mathcal{G}_{p,\theta}^{\gamma}$, where $\gamma$ is mixed or anisotropic smoothness. Moreover, $F^\circ$ and the importance function $\mu$ satisfy the following conditions:*

*(i) $\|F^\circ\|_{\mathcal{F}_{p,\theta}^{\gamma}} \leq 1, \|F^\circ\|_\infty \leq R_0, \|\mu\|_{\mathcal{F}_{p,\theta}^{\gamma}} \leq 1, \|\mu\|_\infty \leq 1$ for some constant $R_0 > 0$.*

*(ii) For the smoothness parameter $a$, it holds $a_{ij} = \Omega(\log(|j| + 1))$ for $\mu \in \mathcal{F}_{p,\theta}^{\gamma}$, and $a_{ij} = \Omega(j^\alpha)$ for $F^\circ \in \mathcal{G}_{p,\theta}^{\gamma}$, where $\alpha > 0$ is a constant. Moreover, it holds $\|a\|_{wl^\alpha} \leq 1$ for both $\mu$ and $F^\circ$, where $\|a\|_{wl^\alpha} := \sup_j j^\alpha \bar{a}_j^{-1}$ and $\bar{a}_j$ is the $j$-th smallest element of $a$.*

*(iii) If $\gamma$ is mixed smoothness, we assume $\bar{a}_1 < \bar{a}_2$.*

**Remark 4.4.** The assumption $\|a\|_{wl^\alpha} \leq 1$ implies that the $j$-th smallest element of smoothness parameter $a$ increases polynomially with respect to $j$, which indicates the *sparsity* of the important features. This assumption is natural in real-world applications, as we check in Section 5. Moreover, the assumptions $a_{ij} = \Omega(\log(|j| + 1))$ and $a_{ij} = \Omega(j^\alpha)$ mean that the token placed far from the final token is less important. Note that the condition $a_{ij} = \Omega(j^\alpha)$ for $F^\circ = f \circ \Pi \in \mathcal{G}_{p,\theta}^{\gamma}$ is imposed on the function $f \in \mathcal{F}_{p,\theta}^{\gamma}$, and the input of the function $f$ is sorted by the importance.

Then, we have the following theorem on the approximation ability of SSMs for piecewise $\gamma$-smooth functions.

**Theorem 4.5.** *Let $F^\circ$ be a function satisfying Assumption 4.3. Then, for any $T > 0$, there exists a SSM $\hat{F} \in \mathcal{S}'(M, U, D, L, W, S, B)$ with*

$$M \lesssim T^{1/\alpha}, \quad U = V, \quad D \lesssim T^{c_{\alpha,\beta}} \log^2 V, \quad L \lesssim T^{c_{\alpha,\beta}} \log^5 V,$$

$$W \lesssim 2^{T/a^\dagger} T^{c_{\alpha,\beta}} \log^3 V, \quad S \lesssim 2^{T/a^\dagger} T^{c_{\alpha,\beta}} \log^5 V, \quad \log B \lesssim T^{c_{\alpha,\beta}} \log^3 V, \tag{4.1}$$

*such that $\left\| F^\circ - \hat{F} \right\|_{2, P_X} \lesssim 2^{-T}$. Here, $c_{\alpha,\beta}$ is a constant depending on $\alpha$ and $\beta$ such that $c_{\alpha,\beta} \leq 5 + 2/\alpha + 5\beta/\alpha$.*

This result reveals that the number of parameters to achieve the error $\epsilon$ is $O((1/\epsilon)^{\frac{1}{a^\dagger}} \operatorname{poly} \log(1/\epsilon, V))$, which is the same as that of Transformers shown by Takakura & Suzuki (2023). Similarly to Theorem 3.1 and Theorem 3.2, to prove this theorem, we utilize the similar argument as Lemma 3.3, which establishes the dynamic token selection ability of SSMs combined with FNNs. The detailed proof can be found in Appendix I.

Using the approximation theory above, we obtain the following results, which state the estimation ability of SSMs for piecewise $\gamma$-smooth functions.

**Theorem 4.6.** *Suppose that the target function $F^\circ$ satisfies Assumption 4.3. Let $a^\dagger = \bar{a}_1$ for mixed smoothness, and $a^\dagger = \left( \sum_{i=1}^\infty \bar{a}_i^{-1} \right)^{-1}$ for anisotropic smoothness. Moreover, let $\hat{F}$ be an ERM estimator in $\mathcal{S}(M, U, D, L, W, S, B)$, with $M, U, D, L, W, S, B$ defined as (4.1) for $T = \frac{a^\dagger}{2a^\dagger + 1}$. Then, it holds*

$$R(\hat{F}, F^\circ) \lesssim n^{-\frac{2a^\dagger}{2a^\dagger + 1}} (\log n)^{c'_{\alpha,\beta}} \log^{13} V,$$

*where $c'_{\alpha,\beta}$ is a constant such that $c'_{\alpha,\beta} \leq 21 + 10/\alpha + 20\beta/\alpha$.*

The proof can be found in Appendix J. We can see that the convergence rate with respect to $n$ matches that of Transformers shown in Takakura & Suzuki (2023). This indicates that SSMs possess the ability to *select important tokens based on the inputs*, similarly to Transformers. Moreover, since the estimation error bound depends on $V$ with only poly-log factor, if $V = \operatorname{poly}(n)$, the estimation error rate does not change up to poly-log factor. This also aligns with Transformers and shows that, even when estimating functions that depend on long ranges of the input sequence, SSMs are as efficient as Transformers. Overall, we can conclude that SSMs can estimate the function in the piecewise $\gamma$-smooth function class with the same efficiency as Transformers.

## 5 EXPERIMENTS: SPARSITY OF IMPORTANT TOKENS

In the tasks we theoretically study, the number of essential tokens in the sequence is small. We conducted simple numerical experiments and confirmed that (i) such sparsity of important tokens also holds in real-world tasks, and that (ii) SSMs can indeed extract these important tokens depending on the input.

We use the dataset of DNA base sequences in Genomic Benchmark Dataset (Grešová et al., 2023). The base sequences are treated as sequences where each nucleic acid is considered a single token, and we consider a binary classification task based on the role of the base sequences. We employ the pre-trained Hyena provided by Nguyen et al. (2024), and fine-tune it on the dataset.

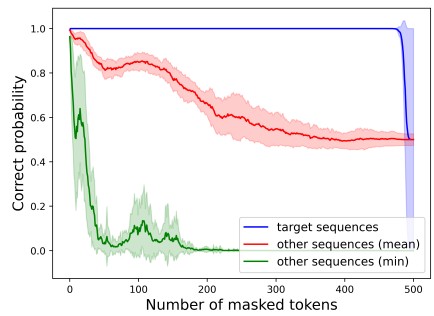

Figure 5.1: The transition of the probability of correct classification when we repeatedly mask the input tokens.

To investigate which tokens the model focuses on for classification, we selected a correctly classified data sample (we refer to this as the *target sequence*) and masked unimportant tokens one by one. More precisely, we repeatedly masked the token that causes the smallest decrease in accuracy when masked. The change in accuracy is shown by the blue line in Figure 5.1. We can see that, even when most of the tokens are masked, the model is still able to classify correctly. This indicates that the important tokens in the input sequence are sparse.

Furthermore, to investigate the ability of SSMs to dynamically extract important tokens, we examined how the accuracy changes by masking the tokens at the same positions as in the target sequence for other data samples. The average and minimum changes in accuracy are shown with the red and green lines, respectively. The figure shows that although the accuracy was high before masking, it decreases as more tokens are masked. This demonstrates that the positions of important tokens vary across different data samples, and that SSMs are able to dynamically extract important tokens based on the input to perform correct classifications.

## 6  CONCLUSION

In this study, we theoretically investigated the capabilities of SSMs compared to Transformers. Specifically, we focused on the ability to dynamically extract tokens based on the input, which is an essential strength of Transformers, and clarified that SSMs combined with FNN layers can emulate such mechanism. Using this insight, we analyzed three cases: input copying, associative recall, and nonparametric regression, and showed that SSMs exhibit performance comparable to Transformers.

**Limitations and future work**   We studied approximation and estimation abilities of SSMs to solve the tasks, and did not discuss whether SSMs can be optimized efficiently. Analyzing how the optimization algorithm works for SSMs is a possible direction for future work. Additionally, we did not investigate the other types of efficient sequence models, such as SSMs with data-dependent filters (like Mamba (Gu & Dao, 2023)) and linear attention (Katharopoulos et al., 2020). Future research could focus on the comparison of those models and SSMs. Moreover, we did not consider a specific parameterization known in practical applications of SSMs. Specifically, we did not impose constraints such as $A$ being a diagonal matrix in the filter $CA^{t-n}B + D\delta_{t-n}$. Instead, as described in Appendix A, we considered cases where $A$ is a block diagonal matrix. It remains future work to explore how we can constrain the structure of $A$ to solve the tasks.

## ACKNOWLEDGMENT

NN was partially supported by JST, ACT-X (JPMJAX24CK). TS was partially supported by JSPS KAKENHI (24K02905) and JST CREST (JPMJCR2015).

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

# —— **Appendix** ——

## A  EXTENSION TO ORDINARY SSM FILTER

In this section, we describe how to extend our setting to the ordinary SSM filter. More specifically, our setting with embedding dimension $D$ can be extended to the ordinary SSM filter with embedding dimension $4D$.

For simplicity, we consider the case $D = 1$. We constuct the parameters $\mathsf{A}, \mathsf{B}, \mathsf{C}, \mathsf{D} \in \mathbb{R}^{2 \times 2}$ to make the filter $h_t := \mathsf{C}\mathsf{A}^t\mathsf{B} + \mathsf{D}\delta_{t-n}$ same as the filter defined in Section 2. Let us set $\mathsf{D} = 0$, and

$$
\mathsf{A} = \begin{bmatrix}
\cos\left(\frac{2\pi a_{1,1}}{U+1}\right) & -\sin\left(\frac{2\pi a_{1,1}}{U+1}\right) & 0 & 0 \\
\sin\left(\frac{2\pi a_{1,1}}{U+1}\right) & \cos\left(\frac{2\pi a_{1,1}}{U+1}\right) & 0 & 0 \\
0 & 0 & \cos\left(\frac{2\pi a_{1,2}}{U+1}\right) & -\sin\left(\frac{2\pi a_{1,2}}{U+1}\right) \\
0 & 0 & \sin\left(\frac{2\pi a_{1,2}}{U+1}\right) & \cos\left(\frac{2\pi a_{1,2}}{U+1}\right)
\end{bmatrix}.
$$

Then, we have

$$
\mathsf{A}^t = \begin{bmatrix}
\cos\left(\frac{2\pi a_{1,1}t}{U+1}\right) & -\sin\left(\frac{2\pi a_{1,1}t}{U+1}\right) & 0 & 0 \\
\sin\left(\frac{2\pi a_{1,1}t}{U+1}\right) & \cos\left(\frac{2\pi a_{1,1}t}{U+1}\right) & 0 & 0 \\
0 & 0 & \cos\left(\frac{2\pi a_{1,2}t}{U+1}\right) & -\sin\left(\frac{2\pi a_{1,2}t}{U+1}\right) \\
0 & 0 & \sin\left(\frac{2\pi a_{1,2}t}{U+1}\right) & \cos\left(\frac{2\pi a_{1,2}t}{U+1}\right)
\end{bmatrix}.
$$

Therefore, if we set

$$
\mathsf{B} = \begin{bmatrix}
1 & 0 & 0 & 0 \\
0 & 0 & 0 & 0 \\
0 & 0 & 1 & 0 \\
0 & 0 & 0 & 0
\end{bmatrix}, \quad
\mathsf{C} = \begin{bmatrix}
c_{1,1} & 0 & 0 & 0 \\
0 & 0 & 0 & 0 \\
0 & 0 & 0 & 0 \\
c_{1,2} & 0 & 0 & 0
\end{bmatrix},
$$

then we have

$$
h_t = \begin{bmatrix}
c_{1,1}\cos\left(\frac{2\pi a_{1,1}t}{U+1}\right) & 0 & 0 & 0 \\
0 & 0 & 0 & 0 \\
0 & 0 & 0 & 0 \\
c_{1,2}\sin\left(\frac{2\pi a_{1,2}t}{U+1}\right) & 0 & 0 & 0
\end{bmatrix}.
$$

Then, if we appropriately set $W^V$ and $W^Q$, this filter can realize the same output with our setting.

While we do not show the estimation ability for the filter above, we can easily extend our proof to derive the almost same estimation error bound for it.

## B  REPRODUCTION OF THE FINITE WINDOW SETTING

For mathematical simplicity, we assumed in Section 2 that the convolution of the SSM layers are performed within *finite windows*, which can be smaller than the sequence length. However, in practical applications, the window size is equal to the sequence length. In the theorem presented in Section 3, we consider the case where the window size matches the sequence length, and this aligns with realistic problem settings. On the other hand, in the nonparametric regression discussed in Section 4, since we consider infinitely long sequences, it is impossible to set the window size equal to the sequence length.

In the problem setting of Section 4, we can obtain the same output as when using a finite window size by performing some additional calculations with standard SSMs. Let $[u_t]_{t \le 0}$ be the input sequence, and $[x_t]_{t \le 0}, [y_t]_{t \le 0}$ be the sequence of states and outputs of standard SSMs, respectively. In other words, for any $t \le 0$, we have

$$
x_{t+1} = \mathsf{A}x_t + \mathsf{B}u_t,
$$
$$
y_t = \mathsf{C}x_t + \mathsf{D}u_t,
$$

and

$$x_t = \sum_{s \leq t} \mathsf{A}^{t-s} \mathsf{B} u_s.$$

Next, let $[x'_t]_{t \leq 0}, [y'_t]_{t \leq 0}$ be the sequence of states and outputs when the shifted input sequence $[u_{t-U-1}]_{t \leq 0}$ are fed into the same SSMs. Then, it holds

$$x'_{t+1} = \mathsf{A} x'_t + \mathsf{B} u_{t-U-1},$$
$$y'_t = \mathsf{C} x'_t + \mathsf{D} u_{t-U-1},$$

and

$$x'_t = \sum_{s \leq t} \mathsf{A}^{t-s} \mathsf{B} u_{s-U-1} = \sum_{s \leq t-U-1} \mathsf{A}^{t-s-U-1} \mathsf{B} u_s.$$

Therefore, we have

$$x_t - \mathsf{A}^{U+1} x'_t = \sum_{s=t-U}^{t} \mathsf{A}^{t-s} \mathsf{B} u_s.$$

Let $[y^\circ_t]_{t \leq 0}$ be the output sequence of SSMs with the finite window size of length $U + 1$. Then, we have

$$y^\circ_t = \sum_{s=t-U}^{t} \left( \mathsf{C} \mathsf{A}^{t-s} \mathsf{B} + \mathsf{D} \delta_{t-s} \right) u_s = \mathsf{C}(x_t - \mathsf{A}^{U+1} x'_t) + \mathsf{D} u_t.$$

Since $\mathsf{A}^{U+1}$ can be pre-computed, we can obtain the output of SSMs with the window by performing recurrent calculations for two SSMs.

## C   EMPIRICAL RESULTS ON THE SYNTHETIC TASKS

In order to empirically demonstrate the dynamic token selection ability of SSMs, we conducted experiments on the input copying, associative recall, and nonparametric regression.

We consider three types of models: (i) single-layer SSMs (SSM + FNN), (ii) two-layer SSMs (SSM + FNN + SSM + FNN), and (iii) Transformers. As we proved theoretically in Lemma 3.3, to exhibit dynamic token selection ability in SSMs, it is essential that SSMs are preceded and followed by FNN layers. Therefore, theoretically, (ii) two-layer SSMs are expected to perform similarly to (i) Transformers. Moreover, since (i) single-layer SSMs do not have the dynamic token selection ability, they are expected to perform worse than two-layer SSMs and Transformers.

To demonstrate the effectiveness of adding FNN layers to SSMs, we vary the dimension of the hidden states (i.e., dimension of the states in SSMs). Then, we observe the changes in the performance.

**Input Copying and Associative Recall**   We first conducted experiments on the input copying and associative recall tasks. Results are shown in Figure C.1. We can see that the performance of (ii) two-layer SSMs is better than (i) single-layer SSMs, particularly when the dimension of the hidden states is small and the FNN layers have sufficient expressive power. This means that the alternation of SSM layers and FNN layers is essential to exhibit SSMs' dynamic token selection ability. Moreover, we observe that the performance of (ii) two-layer SSMs is comparable to (iii) Transformers. This result empirically supports our theoretical analysis that SSMs combined with FNN layers can mimic the dynamic token selection ability of Transformers.

**Nonparametric Regression**   We also conducted experiments on the nonparametric regression task. In this experiment, we consider the following setting: let us consider the situation where inputs are given as sequences of words from the set $\mathcal{W}$. For each word $w \in \mathcal{W}$, we assign a value $r_w \in [0, 1]$. When the input sequence is $X = [w_1, w_2, \ldots, w_L]$, the output $y$ is given as

$$y = f(\max \{r_{w_1}, \ldots, r_{w_L}\}),$$

where $f$ is a non-linear function. The function to be estimated,

$$X \mapsto f(\max \{r_{w_1}, \ldots, r_{w_L}\}),$$

can be framed as a piecewise $\gamma$-smooth function. Indeed, the values assigned to the words can be seen as the importance of each token, and the input to the smooth function $f$ is the token with the highest importance.

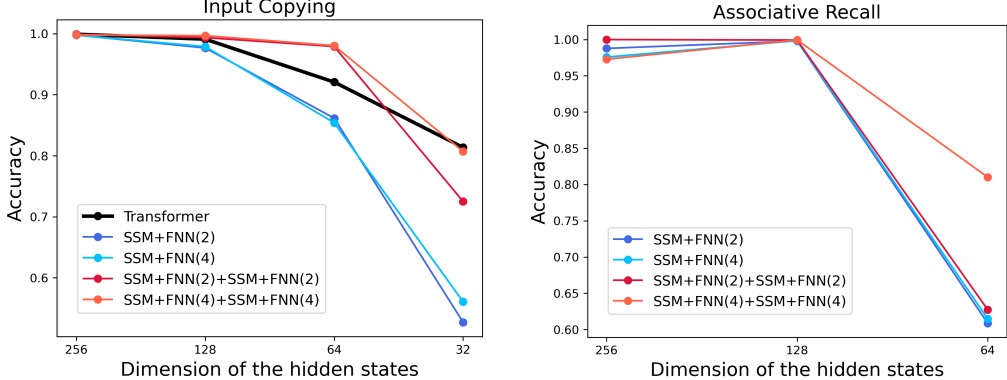

Figure C.1: Empirical results for input copying task (left) and associative recall task (right). We compare the performance of single-layer SSMs (SSM + FNN), two-layer SSMs (SSM + FNN + SSM + FNN), and Transformers. The number in parentheses following "FNN" indicates the depth of the FNN. We can see that two-layer SSMs with sufficiently expressive FNN layers exhibit performance comparable to Transformers, and outperform single-layer SSMs.

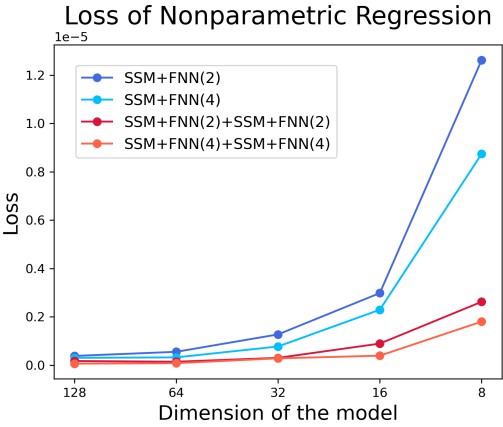

Figure C.2: Empirical results for nonparametric regression. We can see that two-layer SSMs (with FNNs) perform better than one-layer SSMs, similar to input copying and associative recall.

Results are shown in Figure C.2. For a single-layer SSM, the loss increases sharply as the dimension of the hidden state decreases. In contrast, when using a two-layer SSM, the rise in loss due to reducing the hidden state dimension is mitigated. This suggests that stacking two SSM layers helps improve performance in nonparametric regression. This observation aligns with our theoretical findings that having FNN layers both before and after the SSM is crucial.

# D    ADDITIONAL RESULTS ON SYNTHETIC TASKS

In this section, we provide additional results on the SSMs' ability to solve synthetic tasks. Specifically, we consider the two tasks: *induction heads* and *selective copying*.

## D.1    INDUCTION HEAD

The induction heads (Olsson et al., 2022) is a task to recall the word that appears immediately after a specific keyword. For example, if the keyword is x and the input sequence is "a c b d e c", the model have to output the word "b", which is the word that appears after "c".

To formalize the task, suppose that input sequences are given by the form "$x_1 \ x_2 \ \cdots \ x_V \ k$", where $x_1, \ldots, x_V, k \in \mathcal{W}$. Moreover, we assume that, for each sequence, there exists a unique $j \in [V-1]$ such that $x_j = k$. Then, the model is required to output $x_{j+1}$.

In this task, the position of the token to extract changes for each input sequence. Therefore, to solve this task, the model has to change which token to focus based on the input, i.e., the dynamic token selection ability is required.

For induction heads, we obtain the following result.

**Theorem D.1.** *There exists an SSM $\hat{F} \in \mathcal{S}(M, U, D, L, W, S, B)$ with*
$$M = 2, \quad U = V, \quad D, \ L, \ W, \ S, \ \log B \lesssim \log^5 V \log^8 |\mathcal{W}|,$$
*and decoding layer* Dec *with* $\|W_{\text{Dec}}\|_\infty \leq 1$ *such that, for any input sequences of induction heads task, the model generates the correct output.*

The proof is provided in Appendix H.

### D.2 SELECTIVE COPYING

The selective copying (Gu & Dao, 2023) is a variant of the input copying task, where there are some empty tokens between the tokens to copy. For example, if the input sequence is "⟨BOS⟩ a ⟨PAD⟩ ⟨PAD⟩ b ⟨PAD⟩ c ⟨PAD⟩ ⟨COPY⟩", the model have to generate the sequence "a b c" in an auto-regressive manner.

Similarly to the input copying task, since the models have to change the position of the token to copy, the dynamic token selection ability is required to solve this task. Moreover, since the model needs to avoid empty tokens at different positions for each sequence and copy only the necessary tokens. Therefore, it requires capturing the context of the sequence, making it a more challenging than input copying task.

To provide a formal definition of the task, suppose that the special tokens ⟨BOS⟩, ⟨PAD⟩, and ⟨COPY⟩ are included in the vocabulary $\mathcal{W}$. Then, let us consider the input sequence of the form "⟨BOS⟩ $x_1 \ x_2 \ \cdots \ x_V$ ⟨COPY⟩", where $x_1, \ldots, x_V \in \mathcal{W} \setminus \{$⟨BOS⟩, ⟨COPY⟩$\}$. For each $i \in [V]$, $x_i$ is a random variable that matches ⟨PAD⟩ with probability $\alpha \ (> 0)$. Otherwise, $x_i$ is generated from the uniform distribution over $\mathcal{W} \setminus \{$⟨BOS⟩, ⟨COPY⟩, ⟨PAD⟩$\}$. Let $i_1, \ldots, i_K \in [V]$ be the indices such that $x_{i_k} \neq$ ⟨PAD⟩ $(k \in [K])$. Then, the model is required to output the sequence "$x_{i_1} \cdots x_{i_K}$".

For the task described above, we obtain the following result.

**Theorem D.2.** *Let $\epsilon > 0$. Suppose that $|\mathcal{W}| \gtrsim \log^4(V/\epsilon)$. Then, there exists an SSM $\hat{F} \in \hat{\mathcal{S}}(M, U, D, L, W, S, B)$ with*
$$M = 2, \quad U = V, \quad D, \ L, \ W, \ S, \ \log B \lesssim \log^{84} V \log^{89} \epsilon^{-1} \log^8 |\mathcal{W}|,$$
*and decoding layer* Dec *with* $\|W_{\text{Dec}}\|_\infty \leq 1$ *such that, the model generates the correct sequence for selective copying task with probability $1 - \epsilon$.*

In this theorem, compared to the case of input copying, we additionally assume that $|\mathcal{W}| \gtrsim \log^4(V/\epsilon)$. This is mainly due to the existence of empty tokens in the input sequence, and is not due to the problems specific to the SSMs, i.e., the same problem would occur in the case of Transformers. More concretely, in the proof of the theorem, similarly to the proof of Theorem 3.1, we consider the $n$-gram immediately before the token, and construct a network that can find the same $n$-gram (excluding empty tokens) in the input sequence. Since there are empty tokens in the input sequence in selective copying, $n$-gram overlapping (excluding empty tokens) can easily occur compared to the case without empty tokens. In particular, when the vocabulary size $|\mathcal{W}|$ is small, $n$-gram overlapping much more likely to occur, thus it is difficult to copy the sequence correctly.

The proof is provided in Appendix H.

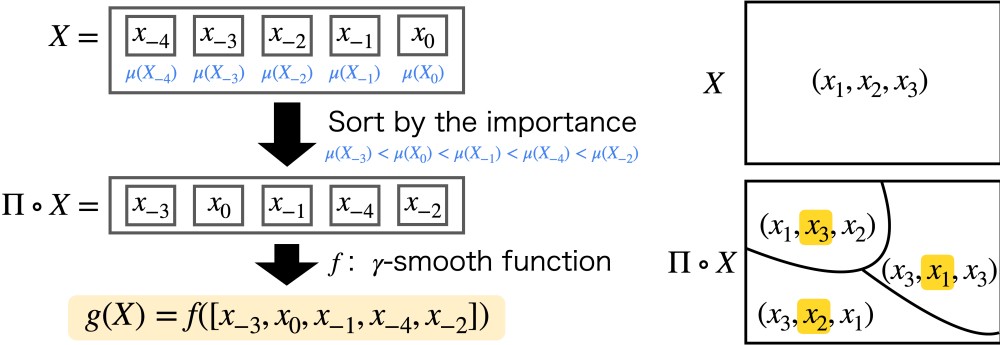

Figure E.1: Intuitive explanation of piecewise $\gamma$-smooth functions. Left: For simplicity, consider a finite-length input sequence $X = [x_{-4}, \ldots, x_{-1}, x_0]$. An importance function $\mu$ takes the sequence as input and determines the importance of the last token. Using the function $\mu$, the importance values of each token, $\mu(X_{-4}), \ldots, \mu(X_0)$, are determined. A permutation map $\Pi$ rearranges the tokens in ascending order of their importance. Finally, the rearranged tokens are fed into a $\gamma$-smooth function $f$. In the sorted sequence, tokens in the right have higher importance, and the function $f$ becomes less smooth for tokens positioned further to the right. Right: An intuitive explanation of how the smoothness of a function changes due to token reordering. As an example, consider a function with a 3-dimensional input vector $X = (x_1, x_2, x_3)$. Assume $f$ is only non-smooth in the direction of the second coordinate, while it is smooth in all other directions. If $X$ is directly fed into $f$, the second coordinate, $x_2$, is always the non-smooth direction. On the other hand, if the coordinates are rearranged by an input-dependent permutation map $\Pi$ before being passed to $f$, the smoothness of the function changes. For example, in the top-left region of the domain, the reordering might cause the second coordinate to correspond to $x_3$, making $x_3$ the non-smooth direction.

# E   THE INTUITION BEHIND PIECEWISE $\gamma$-SMOOTH FUNCTIONS

In this chapter, we provide an intuitive explanation of the definition of piecewise $\gamma$-smooth functions introduced in Section 4.2.

The goal of our study is to demonstrate that SSMs possess the ability to focus on important tokens in the input sequence, similar to Transformers. To achieve this, we formulate the importance of each token in terms of the smoothness of a function. Specifically, let us consider a function $f$ that takes a sequence of tokens $X = [\ldots, x_{-2}, x_{-1}, x_0]$ as input and outputs $y = f(X)$. If the function $f$ is smooth with respect to a token (coordinate) $x_i$, we regard that token as unimportant; conversely, if $f$ is not smooth with respect to $x_i$, we consider the token to be important. This is because, when $f$ is smooth with respect to $x_i$, the value of $f$ does not change significantly with variations in $x_i$, and vice versa.

To quantitatively handle the smoothness of a function, we first consider $\gamma$-smooth functions. As described in Section 4.2, $\gamma$-smooth functions form a class of functions that includes spaces such as mixed-Besov spaces and Sobolev spaces. While this class includes a wide variety of functions, once a specific function is fixed, its smoothness is also fixed. In other words, the locations of important tokens are independent of the input. Thus, even if we demonstrate the capability of SSMs to estimate functions in this class, it does not reveal whether SSMs possess the ability to dynamically adjust their focus based on the input, i.e., the dynamic token selection ability.

To reflect the dynamic token selection ability of Transformers and SSMs, we introduce piecewise $\gamma$-smooth functions. We provide an illustrative explanation in Figure E.1. To make the smoothness of a function dependent on the input, we consider rearranging the input tokens. If we rearrange the tokens using a permutation map $\Pi$ based on the input $X$ and apply the function $f$, the smoothness changes depending on the input, while the smoothness of the $\gamma$-smooth function $f$ itself is fixed. We define the composition of the permutation map $\Pi$ and the $\gamma$-smooth function $f$, i.e., $f \circ \Pi$, as a piecewise $\gamma$-smooth function.

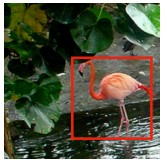 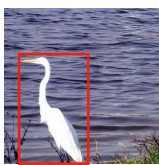

Bob was born in **New York**. After graduating from university, he became a **lawyer**.

What is Bob's **occupation**? ▶ Lawyer
In which **city** was Bob born? ▶ New York

Figure E.2: Real-world tasks where piecewise $\gamma$-smooth functions can be applied. Left: There are a different type of bird in each of the two images. The images are taken from ImageNet (Deng et al., 2009). When considering a task of classifying these two types of birds, the important region is only the part containing the bird, highlighted by the red box. By defining an importance function that assigns larger values to this region, the task can be framed within the framework of piecewise $\gamma$-smooth functions. Right: A passage and two related questions are given. Depending on the question, the important parts of the passage are different. Let us define the importance function that assigns larger values to the relevant parts of the passage based on the given question. Then, this problem setting can also be framed within the framework of piecewise $\gamma$-smooth functions.

To define the permutation map $\Pi$, we introduce an importance function $\mu$. The function $\mu$ takes a sequence of tokens as input and returns the importance of the last token as a real number. Given a sequence $X = [\ldots, x_{-2}, x_{-1}, x_0]$, let $X_{-i} = [\ldots, x_{-i-2}, x_{-i-1}, x_{-i}]$. The importance of a token $x_{-i}$ is then computed as $\mu(X_{-i})$, as shown in Figure E.1. The map $\Pi$ rearranges the tokens in ascending order of their importance scores $\mu(X_{-i})$.

As assumed in Assumption 4.3, the function $f$ becomes smoother with respect to tokens located farther from position 0. Thus, tokens with higher importance as defined by $\mu$ are rearranged to positions closer to position 0 after the permutation. Consequently, these tokens are considered more critical for the function $f$.

Thus, the piecewise $\gamma$-smooth function $g = f \circ \Pi$ is defined. The following two points are particularly important:

- While $\gamma$-smooth functions have fixed smoothness, piecewise $\gamma$-smooth functions have smoothness depending on the input. This is because the tokens are sorted by the order of importance.

- The importance of tokens are determined by the importance function $\mu$. If a token has high importance, it is a significant token for the function $f$.

In Figure E.2, we present concrete examples of real-world problems where piecewise $\gamma$-smooth functions are applicable.

In the example on the left, there are two images with birds. We consider the task of predicting the species of the bird in each image. For this task, only the regions containing the bird are relevant, while the other parts of the images are not essential. Since the locations of the birds differ between the two images, the important regions vary depending on the input. By defining an importance function that assigns higher importance to tokens corresponding to the regions containing the bird, this problem can be framed within the framework of piecewise $\gamma$-smooth functions.

In the example on the right, a passage and related questions are provided. We consider the task of inferring appropriate answers to the questions based on the passage. For the first question, which asks about Bob's profession, the focus should be on the blue-highlighted part of the passage. For the second question, which asks about Bob's hometown, the focus shifts to the green-highlighted part. We can define an importance function that takes the passage and the question as input and assigns higher values to tokens corresponding to the relevant parts of the passage (e.g., the blue part for the first question and the green part for the second question). Then, we can interpret the problem within the framework of piecewise $\gamma$-smooth functions.

# F  AUXILIARY LEMMAS

In the following discussion, to simplify the notation, we define the function class $\Psi'(D, B)$ by

$$\Psi'(D, B) := \left\{ t \mapsto [c_{1,k} \cos{(2\pi a_{1,k} t)} + c_{2,k} \sin{(2\pi a_{2,k} t)}]_{k=1}^{D} \;\middle|\; \|c\|_{\infty} \leq B, \|a\|_{\infty} \leq B. \right\}.$$

First, we prove the following lemma, which states the properties of the $\mathrm{Softmax}$ and multi-variate Swish function.

**Lemma F.1** (Properties of $\mathrm{Softmax}$ and Multi-variate Swish function). *Fix $\theta \in \mathbb{R}^d$. Assume that there exists an index $i^* \in [d]$ and $\delta > 0$ such that $\theta_{i^*} > \theta_i + \delta$ for all $i \neq i^*$. Then, the following two statements hold:*

1. *(Lemma C.1 of Takakura & Suzuki (2023)) It holds*

$$\sum_{i=1}^{d} |\mathrm{Softmax}(\theta)_i - \delta_{i,i^*}| \leq 2d \exp(-\delta).$$

2. *For any $x \in [0, 1]^d$, it holds*

$$\left| \sum_{i=1}^{d} \mathrm{Softmax}(\theta)_i \cdot x_i - x_{i^*} \right| \leq 2d^2 \exp(-\delta).$$

*Proof.* We prove the second one. Using the first argument, we have

$$\left| \sum_{i=1}^{d} \mathrm{Softmax}(\theta)_i \cdot x_i - x_{i^*} \right|$$

$$\leq \left| \sum_{i \neq i^*} \mathrm{Softmax}(\theta)_i \cdot x_i + (\mathrm{Softmax}(\theta)_{i^*} \cdot x_{i^*} - x_{i^*}) \right|$$

$$= \left| \sum_{i \neq i^*} \mathrm{Softmax}(\theta)_i \cdot x_i + (\mathrm{Softmax}(\theta)_{i^*} \cdot x_{i^*} - \delta_{i^*,i^*} x_{i^*}) \right|$$

$$\leq \sum_{i \neq i^*} |\mathrm{Softmax}(\theta)_i - \delta_{i,i^*}| \cdot x_i + |\mathrm{Softmax}(\theta)_{i^*} - \delta_{i^*,i^*}| \cdot x_{i^*}$$

$$\leq \sum_{i=1}^{d} |\mathrm{Softmax}(\theta)_i - \delta_{i,i^*}| \cdot x_i$$

$$\leq 2d^2 \exp(-\delta),$$

which completes the proof. $\qquad \square$

The following is a famous fact that there exists a neural network that realize the clipping function.

**Lemma F.2.** *Let $a, b \in \mathbb{R}$. There exists a neural neural network $f_{\mathrm{clip}} \in \Psi(L, W, S, B)$ with*

$$L \lesssim 1, \quad W \lesssim 1, \quad S \lesssim 1, \quad B \lesssim |a| + |b|,$$

*such that, for any $x \in \mathbb{R}$, it holds*

$$f_{\mathrm{clip}}(x) = \begin{cases} a & \text{if } x \leq a, \\ x & \text{if } a \leq x \leq b, \\ b & \text{if } b \leq x. \end{cases}$$

The following lemma shows the approximation ability of FNN for some elementary functions.

**Lemma F.3** (Lemma F.6, Lemma F.7, Lemma F.12 of Oko et al. (2023), Corollary 4.2 of Perekrestenko et al. (2018)). *The following statements hold:*

(mult) *Let $d \geq 2, C \geq 1, \epsilon_{\mathrm{error}} \in (0, 1]$. For any $\epsilon > 0$, there exists a neural network $f_{\mathrm{mult}} \in \Psi(L, W, S, B)$ with*

$$L \lesssim (\log \epsilon^{-1} + d \log C) \cdot \log d, \quad W \lesssim d, \quad S \lesssim d \log \epsilon^{-1} + d \log C, \quad \log B \lesssim d \log C,$$

*such that, for any $x \in [0, C]^d$ and $x \in \mathbb{R}^d$ with $\|x - x'\|_\infty \le \epsilon_{\mathrm{error}}$, it holds*

$$\left| f_{\mathrm{mult}}(x') - \prod_{i=1}^{d} x_i \right| \le \epsilon + d \cdot C^d \cdot \epsilon_{\mathrm{error}}.$$

*(rec) For any $\epsilon \in (0, 1)$, there exists $f_{\mathrm{rec}} \in \Psi(L, W, S, B)$ with*
$$L \lesssim \log^2 \epsilon^{-1}, \quad W \lesssim \log^3 \epsilon^{-1}, \quad S \lesssim \log^4 \epsilon^{-1}, \quad \log B \lesssim \log \epsilon^{-1},$$
*such that, for any $x \in [\epsilon, \epsilon^{-1}]$ and $x' \in \mathbb{R}$, it holds*
$$\left| f_{\mathrm{rec}}(x') - \frac{1}{x} \right| \le \epsilon + \frac{|x' - x|}{\epsilon^2}.$$

*(exp) For any $\epsilon > 0$, there exists $f_{\exp} \in \Psi(L, W, S, B)$ with*
$$L \lesssim \log^2 \epsilon^{-1}, \quad W \lesssim \log \epsilon^{-1}, \quad S \lesssim \log^2 \epsilon^{-1}, \quad \log B \lesssim \log^2 \epsilon^{-1},$$
*such that, for any $x, x' \ge 0$, it holds*
$$|f_{\exp}(x') - \exp(x)| \le \epsilon + |x' - x|.$$

*(cos) For any $\epsilon > 0, a > 0, b \in \mathbb{R}, C \ge 1$, there exists $f_{\cos} \in \Psi(L, W, S, B)$ with*
$$L \lesssim \log^2 \epsilon^{-1} + \log(aD + b), \quad W \lesssim 1,$$
$$S \lesssim \log^2 \epsilon^{-1} + \log(aD + b), \quad \log B \lesssim \max\{1, \log|b/a|\},$$
*such that, for any $x \in [-D, D]$, it holds*
$$|f_{\cos}(x) - \cos(ax + b)| \le \epsilon.$$

We also use the following lemma, which gives the approximation error of $(x, y) \mapsto y/x$.

**Lemma F.4.** *For any $\epsilon \in (0, 1]$, there exists a neural network $\phi \in \Psi(L, W, S, B)$ with*
$$L \lesssim \log^2 \epsilon^{-1}, \quad W \lesssim \log^3 \epsilon^{-1}, \quad S \lesssim \log^4 \epsilon^{-1}, \quad \log B \lesssim \log \epsilon^{-1},$$
*such that, for any $x, y, x', y' \in \mathbb{R}$ with $x \in [\epsilon, \epsilon^{-1}], y \in [0, \epsilon^{-1}]$, it holds*
$$\left| \phi(x', y') - \frac{y}{x} \right| \le \epsilon + \frac{|x - x'|}{\epsilon^8} + \frac{|y - y'|}{\epsilon^2}.$$

*Proof.* From (rec) of Lemma F.3, there exists a neural network $\phi_1 \in \Psi(L, W, S, B)$ with
$$L \lesssim \log^2 \epsilon^{-1}, \quad W \lesssim \log^3 \epsilon^{-1}, \quad S \lesssim \log^4 \epsilon^{-1}, \quad \log B \lesssim \log \epsilon^{-1},$$
such that, for any $x \in [\epsilon, \epsilon^{-1}] \subseteq [\epsilon^3, \epsilon^{-3}]$ and $x' \in \mathbb{R}$, it holds
$$\left| \phi_1(x') - \frac{1}{x} \right| \le \epsilon^3 + \frac{|x - x'|}{\epsilon^6}.$$
Next, using (mult) of Lemma F.3, there exists a neural network $\phi_2 \in \Psi(L, W, S, B)$ with
$$L \lesssim \log \epsilon^{-1}, \quad W \lesssim 1, \quad S \lesssim \log \epsilon^{-1}, \quad \log B \lesssim \log \epsilon^{-1},$$
such that, for any $y, z \in [0, \epsilon^{-1}]$ and $y', z' \in \mathbb{R}$, it holds
$$|\phi_2(z', y') - zy| \lesssim \epsilon + \frac{|z - z'| + |y - y'|}{\epsilon^2}.$$
Therefore, we have
$$\left| \phi_2(\phi_1(x'), y') - \frac{y}{x} \right| \lesssim \epsilon + \frac{1}{\epsilon^2}\left( \left| \phi_1(x') - \frac{1}{x} \right| + |y - y'| \right)$$
$$\lesssim \epsilon + \frac{|x - x'|}{\epsilon^8} + \frac{|y - y'|}{\epsilon^2}.$$

$\square$

Lastly, we state the following lemma, which shows that the Gaussian kernel can be approximated expressed as the sum of the product of neural networks.

**Lemma F.5.** *There exists* $N \in \mathbb{N}$ *and FNNs* $\phi_n, \phi_n', \phi_n'' \in \Psi_{1,1}(L, W, S, B)$, $\psi_n, \psi_n' \in \Psi'(1, B)$ $(n = 1, \dots, N)$ *with*

$$N \lesssim \log^2 \epsilon^{-1},$$
$$L \lesssim \log^4 \epsilon^{-1} \log^2 \kappa, \quad W \lesssim 1, \quad S \lesssim \log^4 \epsilon^{-1} \log^2 \kappa, \quad \log B \lesssim \log^2 \epsilon^{-1} \log \kappa,$$
$$L' = 1, \quad W' \lesssim \log^2 \epsilon^{-1}, \quad S' \lesssim \log^2 \epsilon^{-1},$$

*such that,*

- *for any $t, x \in [-1, 1]$, it holds*

$$\left| \exp\left(-\kappa \cdot \sin^2\left(\frac{\pi}{2}(t - x)\right)\right) - \sum_{n=1}^{N} \psi_n(t)\phi_n(x) \right| \lesssim \epsilon,$$

- *for any $x, y \in [-1, 1]$, it holds*

$$\left| \exp\left(-\kappa \cdot \sin^2\left(\frac{\pi}{2}(x - y)\right)\right) - \sum_{n=1}^{N} \phi_n'(x)\phi_n''(y) \right| \lesssim \epsilon,$$

- *for any $t \in [-1, 1]$, it holds*

$$\left| \exp\left(-\kappa \cdot \sin^2\left(\frac{\pi t}{2}\right)\right) - \sum_{n=1}^{N} \psi_n'(t) \right| \lesssim \epsilon.$$

*Proof.* The first part of the proof is inspired by Lemma F.12 of Oko et al. (2023). Let us set $A = \log 3\epsilon^{-1}$. The Taylor expansion of $\exp$ shows that, for any $x \in [0, A]$, it holds

$$\left| \exp(-x) - \sum_{n=0}^{N-1} \frac{(-1)^n}{n!} x^n \right| \leq \frac{A^N}{N!}.$$

Additionally, we can evaluate the right-hand side as $A^k/k! \leq (eA/k)^k$. Therefore, if we set $N = \max\left\{ 2eA, \lceil \log_2 3\epsilon^{-1} \rceil \right\}$, the error can be bounded by $\epsilon/3$. Moreover, for $x > A$, we have

$$\left| \exp(-x) - \sum_{n=0}^{N-1} \frac{(-1)^n}{n!} x^n \right| \leq |\exp(-x) - \exp(-A)| + \left| \exp(-A) - \sum_{n=0}^{N-1} \frac{(-1)^n}{n!} x^n \right|$$

$$\leq \frac{\epsilon}{3} + \frac{2\epsilon}{3} = \epsilon.$$

Next, let us approximate $\sum_{n=0}^{N-1} \frac{(-\kappa)^n}{n!} \sin^{2n}\left(\frac{\pi}{2}(t - x)\right)$. We use the fact that

$$\sin^{2n}(x) = \left(\frac{e^{ix} - e^{-ix}}{2}\right)^{2n} = \frac{1}{2^{2n}} \sum_{k=0}^{2n} \binom{2n}{k} (-1)^k e^{i(2k - 2n)x}$$

$$= \frac{(-1)^n}{2^{2n}} \binom{2n}{n} + \sum_{k \geq n+1} \frac{(-1)^k}{2^{2n-1}} \binom{2n}{k} \cos\left((2k - 2n)x\right),$$

where $c_n = 1$ if $n$ is even and $c_n = 0$ if $n$ is odd. Thus, we have

$$\sum_{n=0}^{N-1} \frac{(-\kappa)^n}{n!} \sin^{2n}\left(\frac{\pi}{2}(t - x)\right)$$

$$= \sum_{n=0}^{N-1} \frac{\kappa^n}{n! 2^{2n}} \binom{2n}{n} + \sum_{n=0}^{N-1} \sum_{k \geq n+1} \frac{(-\kappa)^n}{n!} \frac{1}{2^{2n-1}} \binom{2n}{k} \cos\left(\pi(k - n)(t - x)\right)$$

$$= \sum_{n=0}^{N-1} \frac{\kappa^n}{n! 2^{2n}} \binom{2n}{n} + \sum_{n=0}^{N-1} \sum_{k \geq n+1} \frac{(-\kappa)^n}{n!} \frac{1}{2^{2n-1}} \binom{2n}{k} \Bigg( \cos\left(\pi(k - n)t\right) \cos\left(\pi(k - n)x\right)$$

$$+ \sin\left(\pi(k - n)t\right) \sin\left(\pi(k - n)x\right) \Bigg),$$

which is decomposed into the sum of products of functions of $t$ and $x$. Since

$$\left|\frac{(-\kappa)^n}{n!}\frac{1}{2^{2n-1}}\binom{2n}{k}\right| \le \frac{\kappa^n}{n!2^n}\frac{(2n)!}{k!(2n-k)!} \le \frac{\kappa^n}{n!2^n}\frac{2^n(n!)^2}{(\max(k,2n-k))!} = \frac{\kappa^n}{n!2^n}\frac{2^n(n!)^2}{n!} \le \kappa^N,$$

we can see that, there exists $C_0, C_{n,k}$ $(n = 0, \dots, N-1; k = 0, \dots, N-1)$ with

$$C_0 \le \kappa^N, \quad C_{n,k} \le \kappa^N,$$

such that

$$\sum_{n=0}^{N-1}\frac{(-\kappa)^n}{n!}\sin^{2n}\left(\frac{\pi}{2}(t-x)\right) = C_0 + \sum_{n=0}^{N-1}\sum_{k=0}^{N-1}C_{n,k}\big(\cos\left(\pi(k-n)t\right)\cos\left(\pi(k-n)x\right)$$
$$+ \sin\left(\pi(k-n)t\right)\sin\left(\pi(k-n)x\right)\big).$$

The second item to be proved is already obtained setting $x = 0$.

To prove the first item, we approximate each term using neural networks. Lemma F.3 implies that, for any $n, k$ and $\epsilon > 0$, there exists a neural network $\phi_{1,n,k}, \phi_{2,n,k} \in \Psi_{1,1}(L, W, S, B)$ with

$$L \lesssim N^2\log^2\kappa + \log^2\epsilon^{-1}, \quad W \lesssim 1, \quad S \lesssim N^2\log^2\kappa + \log^2\epsilon^{-1}, \quad \log B \lesssim 1,$$

such that

$$|\cos(\pi(k-n)x) - \phi_{1,n,k}(x)| \le \epsilon/(N^2\kappa^N), \quad |\sin(\pi(k-n)x) - \phi_{2,n,k}(x)| \le \epsilon/(N^2\kappa^N).$$

Then, if we approximate $\exp(-\kappa \cdot \cos(2\pi(t-x)))$ by

$$C_0 + \sum_{n=0}^{N-1}\sum_{k=0}^{N-1}C_{n,k}(\cos\left(\pi(k-n)t\right)\phi_{1,n,k}(x) + \sin\left(\pi(k-n)t\right)\phi_{2,n,k}(x)),$$

the error can be bounded by

$$\epsilon + \sum_{n=0}^{N-1}\sum_{k=0}^{N-1}C_{n,k}\cdot\frac{2\epsilon}{N^2\kappa^N} \le \epsilon + N^2\kappa^N\cdot\frac{2\epsilon}{N^2\kappa^N} \le 3\epsilon,$$

which gives the desired result.

For the third item, if we utilize $\phi_{1,n,k}$ and $\phi_{2,n,k}$ to approximate $\cos(\pi(k-n)y)$ and $\sin(\pi(k-n)y)$, respectively, we can obtain the desired result. $\qquad\square$

The following lemma is the multi-dimensional version of Lemma F.5.

**Lemma F.6.** *There exists $N \in \mathbb{N}$ and FNNs $\phi_{n,i}, \phi'_{n,i} \in \Psi_{d,1}(L, W, S, B)$ with*

$$N \lesssim \log^2\epsilon^{-1} + \log^2 d,$$
$$L \lesssim d^4\log^4\epsilon^{-1}\log^2\kappa, \quad W \lesssim d, \quad S \lesssim d^4\log^4\epsilon^{-1}\log^2\kappa, \quad \log B \lesssim d\log^2\epsilon^{-1}\log\kappa,$$

*such that, for any $x, y \in [-1,1]^d$, it holds*

$$\left|\exp\left(-\kappa\cdot\sum_{i=1}^n\sin^2\left(\frac{\pi}{2}(x_i - y_i)\right)\right) - \sum_{n=1}^N\phi_n(x)\phi'_n(y)\right| \lesssim \epsilon.$$

*Proof.* The proof of Lemma F.5 shows that, for $N \sim \log^2\epsilon^{-1} + \log^2 d$, there exists $\phi_1^*, \dots, \phi_N^*$ and $C_1, \cdots, C_N$ with $|C_n| \le \kappa^N$ such that

$$\left|\exp\left(-\kappa\cdot\sin^2\left(\frac{\pi}{2}(x_i - y_i)\right)\right) - \sum_{n=1}^N C_n\phi_n^*(x_i)\phi_n^*(y_i)\right| \lesssim \frac{\epsilon}{d},$$

where $\phi_n^*$ is a function represented as $\sin(a_n x + b_n)$ with some $a_n, b_n \in \mathbb{R}$. Therefore,

$$\prod_{i=1}^d\left(\sum_{n=1}^N C_n\phi_n^*(x_i)\phi_n^*(y_i)\right) = \sum_{n_1,\dots,n_d}C_{n_1}\cdots C_{n_d}\phi_{n_1}^*(x_1)\cdots\phi_{n_d}^*(x_d)\phi_{n_1}^*(y_1)\cdots\phi_{n_d}^*(y_d),$$

which have $N^d$ terms, is an approximation of $\exp\left(-\kappa \cdot \sum_{i=1}^{n} \sin^2\left(\frac{\pi}{2}(x_i - y_i)\right)\right)$, and the error is bounded as

$$\left| \exp\left(-\kappa \cdot \sum_{i=1}^{n} \sin^2\left(\frac{\pi}{2}(x_i - y_i)\right)\right) - \prod_{i=1}^{d}\left(\sum_{n=1}^{N} C_n \phi_n^*(x_i)\phi_n^*(y_i)\right) \right| \leq d \cdot \frac{\epsilon}{d} = \epsilon.$$

Using (cos) of Lemma F.3, we can see that, for any $n \in [N]$ and $\epsilon > 0$, there exists a neural network $\psi_{1,n} \in \Psi_{d,1}(L, W, S, B)$ with

$$L \lesssim d^4 \log^4 \epsilon^{-1} \log \kappa, \quad W \lesssim 1, \quad S \lesssim d^4 \log^4 \epsilon^{-1} \log \kappa, \quad \log B \lesssim \log^2 \epsilon^{-1} \log \kappa,$$

such that

$$|\phi_n^*(x) - \psi_{1,n}(x)| \leq \frac{\epsilon}{d\kappa^{dN} N^d}.$$

Moreover, using (mult) of Lemma F.3, we can see that there exists a neural network $\psi_{2,n_1,\ldots,n_d}, \psi_3 \in \Psi(L, W, S, B)$ with

$$L \lesssim d^2 \log^2 \epsilon^{-1} \log^2 \kappa, \quad W \lesssim d, \quad S \lesssim d^3 \log^2 \epsilon^{-1} \log^2 \kappa, \quad \log B \lesssim d,$$

such that

$$|\psi_2(x_1, \ldots, x_d) - C_{n_1} \cdots C_{n_d} x_1 x_2 \cdots x_d| \lesssim \frac{\epsilon}{\kappa^{dN}}$$

$$|\psi_3(x_1, \ldots, x_d) - x_1 x_2 \cdots x_d| \lesssim \frac{\epsilon}{\kappa^{dN}}.$$

for any $x \in \mathbb{R}$ with $|x| \leq 1$. for any $x_1, \ldots, x_d \in \mathbb{R}$ with $|x_i| \leq 1$. Then, we have

$$\left| \sum_{n_1,\ldots,n_d} \psi_{2,n_1,\ldots,n_d}(\psi_{1,n_1}(x_1), \ldots, \psi_{1,n_d}(x_d))\psi_3(\psi_{1,n_1}(y_1), \ldots, \psi_{1,n_d}(y_d)) \right.$$
$$\left. - \exp\left(-\kappa \cdot \sum_{i=1}^{n} \sin^2\left(\frac{\pi}{2}(x_i - y_i)\right)\right) \right|$$

$$\leq \frac{\epsilon}{\kappa^{dN}} \cdot \kappa^{dN} + \left| \sum_{n_1,\ldots,n_d} C_{n_1} \cdots C_{n_d} \cdot \psi_{1,n_1}(x_1) \cdots \psi_{1,n_d}(x_d) \cdot \psi_{1,n_1}(y_1) \cdots \psi_{1,n_d}(y_d) \right.$$
$$\left. - \exp\left(-\kappa \cdot \sum_{i=1}^{n} \sin^2\left(\frac{\pi}{2}(x_i - y_i)\right)\right) \right|$$

$$\leq \frac{\epsilon}{\kappa^{dN}} \cdot \kappa^{dN} + d\kappa^{dN} N^d \cdot \frac{\epsilon}{d\kappa^{dN} N^d}$$
$$+ \left| \sum_{n_1,\ldots,n_d} C_{n_1} \cdots C_{n_d} \cdot \phi_1^*(x_1) \cdots \psi_d^*(x_d) \cdot \phi_1^*(y_1) \cdots \psi_d^*(y_d) \right.$$
$$\left. - \exp\left(-\kappa \cdot \sum_{i=1}^{n} \sin^2\left(\frac{\pi}{2}(x_i - y_i)\right)\right) \right|$$

$$\lesssim \epsilon,$$

which completes the proof. $\qquad\square$

## G    PROOF OF LEMMA 3.3

First, for any $j \neq j^*$, it holds

$$\left(\frac{1}{2}\|q\|^2 + \frac{1}{2}\|k_j\|^2 - \frac{1}{2}\|q - k_j\|^2\right) - \left(\frac{1}{2}\|q\|^2 + \frac{1}{2}\|k_{j^*}\|^2 - \frac{1}{2}\|q - k_{j^*}\|^2\right) = q^\top k_j - q^\top k_{j^*} \leq -\delta.$$

Now, since it is hold that

$$u^2 - \frac{u^4}{3} \leq \sin^2(u) = \frac{1 - \cos 2u}{2} \leq u^2,$$

for $u \in [0, \pi/2]$, for $A > 0, j \in [-V : 0], i \in [d']$, we have

$$\left| \left( A \sin \left( \frac{\pi}{2A}(k_{ji} - q_j) \right) \right)^2 - \left( \frac{\pi}{2}(k_{ji} - q_j) \right)^2 \right|$$

$$= \left| A^2 \sin^2 \left( \frac{\pi}{2A}(k_{ji} - q_j) \right) - A^2 \left( \frac{\pi}{2A}(k_{ji} - q_i) \right)^2 \right|$$

$$\leq A^2 \cdot \left( \frac{\pi}{2A}(k_{ji} - q_i) \right)^4$$

$$\leq \frac{\pi^4}{A^2}.$$

Therefore, if we set $A = \sqrt{\frac{16\pi^2 d'}{\delta}}$, it holds

$$\left( \frac{1}{2}\|q\|^2 + \frac{1}{2}\|k_j\|^2 - \frac{1}{2} \cdot \frac{4}{\pi^2} \sum_{i=1}^{d'} \left( A \sin \left( \frac{\pi}{2A}(k_{ji} - q_i) \right) \right)^2 \right)$$

$$- \left( \frac{1}{2}\|q\|^2 + \frac{1}{2}\|k_{j^*}\|^2 - \frac{1}{2} \cdot \frac{4}{\pi^2} \sum_{i=1}^{d'} \left( A \sin \left( \frac{\pi}{2A}(k_{j^*} - q_i) \right) \right)^2 \right)$$

$$\leq \left( \frac{1}{2}\|q\|^2 + \frac{1}{2}\|k_j\|^2 - \frac{1}{2}\|q - k_j\|^2 \right) - \left( \frac{1}{2}\|q\|^2 + \frac{1}{2}\|k_{j^*}\|^2 - \frac{1}{2}\|q - k_{j^*}\|^2 \right)$$

$$+ \frac{4d'}{\pi^2} \left| \left( A \sin \left( \frac{\pi}{2A}(k_{ji} - q_i) \right) \right)^2 - \left( \frac{\pi}{2}(k_{ji} - q_i) \right)^2 \right|$$

$$+ \frac{4d'}{\pi^2} \left| \left( A \sin \left( \frac{\pi}{2A}(k_{j^*i} - q_i) \right) \right)^2 - \left( \frac{\pi}{2}(k_{j^*i} - q_i) \right)^2 \right|$$

$$\leq -\delta + \frac{4d'\pi^2}{A^2} + \frac{4d'\pi^2}{A^2} \leq -\delta + \frac{\delta}{4} + \frac{\delta}{4} = -\frac{\delta}{2}.$$

In the following, we denote

$$\mu'_j := \frac{1}{2}\|q\|^2 + \frac{1}{2}\|k_j\|^2 - \frac{1}{2} \cdot \frac{4}{\pi^2} \sum_{i=1}^{d'} \left( A \sin \left( \frac{\pi}{2A}(k_{ji} - q_i) \right) \right)^2.$$

Using Lemma F.1, we have

$$\left\| \frac{\sum_{j=-V}^{0} \exp(\kappa \mu'_j) \cdot v_j}{\sum_{j=-V}^{0} \exp(\kappa \mu'_j)} - v_{j^*} \right\|_\infty \leq 2(V + 1)^2 \exp\left( -\frac{\delta\kappa}{2} \right).$$

for any $\kappa > 0$. Therefore, if we set $\kappa = \Omega\left( \frac{\log \epsilon^{-1} + \log V}{\delta} \right)$, the right-hand side is less than $\epsilon$.

Next, let us consider approximating

$$\exp(\kappa \mu'_j) \cdot v_j = v_j \cdot \exp\left( \kappa \cdot \left( \frac{1}{2}\|q\|^2 + \frac{1}{2}\|k_j\|^2 - \frac{1}{2} \cdot \frac{4}{\pi^2} \sum_{i=1}^{d'} \left( A \sin \left( \frac{\pi}{2A}(k_{ji} - q_i) \right) \right)^2 \right) \right)$$

$$= v_j \cdot \exp\left( \frac{\kappa}{2}\|q\|^2 \right) \exp\left( \frac{\kappa}{2}\|k_j\|^2 \right) \exp\left( -\frac{32\kappa d'}{\delta} \sum_{i=1}^{d'} \sin^2 \left( \frac{\pi}{2} \frac{k_{ji} - q_i}{A} \right) \right).$$

Combining (mult) and (exp) in Lemma F.3, we can see that there exists a neural network $\phi_1 \in \Psi(L, W, S, B)$ with

$$L \lesssim \frac{d'^2}{\delta^2} \left( \log^2 \epsilon^{-1} + \log^2 V \right), \quad W \lesssim \frac{d'}{\delta} \left( \log \epsilon^{-1} + \log V \right),$$

$$S \lesssim \frac{d'^2}{\delta^2} \left( \log^2 \epsilon^{-1} + \log^2 V \right), \quad \log B \lesssim \frac{d'^2}{\delta^2} \left( \log^2 \epsilon^{-1} + \log^2 V \right),$$

such that

$$\left| \phi_1(x) - \exp\left( \frac{\kappa}{2}\|x\|^2 \right) \right| \lesssim \frac{1}{(V + 1)^2} \epsilon^9 \exp(-9\kappa d' - 4\delta)$$

for any $x \in \mathbb{R}$ with $\|x\|_\infty \leq 1$. Moreover, using Lemma F.6, we can see that there exists neural networks $\phi_{2,n}, \phi_{3,n} \in \Psi(L, W, S, B)$ $(n = 1, \ldots, N)$ with

$$N \lesssim \frac{d'^2}{\delta^2} \left(\log^2 \epsilon^{-1} + \log^2 V\right), \quad L \lesssim \frac{d'^8}{\delta^5} \left(\log^5 \epsilon^{-1} + \log^5 V\right), \quad W \lesssim d',$$

$$S \lesssim \frac{d'^8}{\delta^5} \left(\log^5 \epsilon^{-1} + \log^5 V\right), \quad \log B \lesssim \frac{d'^3}{\delta^2} \left(\log^3 \epsilon^{-1} + \log^3 V\right),$$

such that, for any $x, y \in [-1, 1]^d$,

$$\left| \sum_{n=1}^{N} \phi_{2,n}(x) \phi_{3,n}(y) - \exp\left( -\frac{32\kappa d'}{\delta} \sum_{i=1}^{d'} \sin^2\left(\frac{\pi}{2}(x-y)\right) \right) \right| \lesssim \frac{1}{(V+1)^2} \epsilon^9 \exp(-9\kappa d' - 4\delta).$$

Note that, by clipping the output appropriately, we can ensure that

$$\|\phi_1\|_\infty \leq \exp\left(\frac{\kappa d'}{2}\right), \quad \|\phi_{2,n}\|_\infty \leq \left(\frac{32\kappa d'}{\delta}\right)^{d'N}, \quad \|\phi_{3,n}\|_\infty \leq \left(\frac{32\kappa d'}{\delta}\right)^{d'N},$$

without changing the approximation error. Therefore, we have

$$\left\| v_j \cdot \phi_1(q)\phi_1(k_j) \sum_{n=1}^{N} \phi_{2,n}\left(\frac{q}{A}\right)\phi_{3,n}\left(\frac{k_j}{A}\right) - \exp(\kappa\mu_j') \cdot v_j \right\|_\infty$$

$$= \left\| v_j \cdot \phi_1(q)\phi_1(k_j) \sum_{n=1}^{N} \phi_{2,n}\left(\frac{q}{A}\right)\phi_{3,n}\left(\frac{k_j}{A}\right) \right.$$

$$\left. - v_j \cdot \exp\left(\frac{\kappa}{2}\|q\|^2\right) \exp\left(\frac{\kappa}{2}\|k_j\|^2\right) \exp\left( -\frac{32\kappa d'}{\delta} \sum_{i=1}^{d'} \sin^2\left(\frac{\pi}{2} \frac{k_{ji} - q_i}{A}\right) \right) \right\|_\infty$$

$$\lesssim \exp(\kappa d') \cdot \frac{1}{(V+1)^2} \epsilon^9 \exp(-9\kappa d' - 4\delta) = \frac{1}{(V+1)^2} \epsilon^9 \exp(-8(\kappa d' + \delta/2)).$$

Using (mult) of Lemma F.3, we can see that there exists a neural network $\phi_4 \in \Psi(L, W, S, B)$ with

$$L \lesssim \frac{d'^3}{\delta^3} \left(\log^3 \epsilon^{-1} + \log^3 V\right), \quad W \lesssim 1, \quad S \lesssim \frac{d'^3}{\delta^3} \left(\log^3 \epsilon^{-1} + \log^3 V\right), \quad \log B \lesssim 1,$$

such that $|\phi_4(x, y) - xy| \lesssim \epsilon^9 \cdot \frac{1}{N(V+1)^2} \left(\frac{\delta}{32\kappa d'}\right)^{d'N} \exp(-(17\kappa d' + \delta)/2)$ for any $x, y \in \mathbb{R}$ with $|x| \leq \exp\left(\frac{\kappa d'}{2}\right), |y| \leq \left(\frac{32\kappa d'}{\delta}\right)^{d'N}$. Additionally, there exists a neural network $\phi_5 \in \Psi(L, W, S, B)$ with

$$L \lesssim \frac{d'^3}{\delta^3} \left(\log^3 \epsilon^{-1} + \log^3 V\right), \quad W \lesssim 1, \quad S \lesssim \frac{d'^3}{\delta^3} \left(\log^3 \epsilon^{-1} + \log^3 V\right), \quad \log B \lesssim 1,$$

such that $|\phi_5(v, x, y) - v \cdot xy| \lesssim \epsilon^9 \cdot \frac{1}{N(V+1)^2} \left(\frac{\delta}{32\kappa d'}\right)^{d'N} \exp(-(17\kappa d' + \delta)/2)$ for any $v \in \mathbb{R}^{d'} x, y \in \mathbb{R}$ with $\|v\|_\infty \leq 1$ and $|x| \leq \exp\left(\frac{\kappa d'}{2}\right), |y| \leq \left(\frac{32\kappa d'}{\delta}\right)^{d'N}$. Using these networks, we have the following approximation:

$$\left| \phi_4\left(\phi_1(q), \phi_{2,n}\left(\frac{q}{A}\right)\right) - \phi_1(q)\phi_{2,n}\left(\frac{q}{A}\right) \right|$$

$$\lesssim \epsilon^9 \cdot \frac{1}{N(V+1)^2} \left(\frac{\delta}{32\kappa d'}\right)^{d'N} \exp(-(17\kappa d' + \delta)/2),$$

$$\left\| \phi_5\left(v_j, \phi_1(k_j), \phi_{3,n}\left(\frac{k_j}{A}\right)\right) - v_j \phi_1(k_j)\phi_{3,n}\left(\frac{k_j}{A}\right) \right\|_\infty$$

$$\lesssim \epsilon^9 \cdot \frac{1}{N(V+1)^2} \left(\frac{\delta}{32\kappa d'}\right)^{d'N} \exp(-(17\kappa d' + \delta)/2).$$

This implies that there exist neural networks $\Phi_1, \psi_1 \in \Psi(L, W, S, B)$ with

$$L \lesssim \frac{d'^8}{\delta^5} \left(\log^5 \epsilon^{-1} + \log^5 V\right), \quad W \lesssim \frac{d'}{\delta} \left(\log \epsilon^{-1} + \log V\right),$$

$$S \lesssim \frac{d'^8}{\delta^5}\big(\log^5 \epsilon^{-1} + \log^5 V\big), \quad \log B \lesssim \frac{d'^3}{\delta^2}\big(\log^3 \epsilon^{-1} + \log^3 V\big),$$

such that

$$\big\|\Phi_1(k_j, v_j)\psi(q) - \exp\big(\kappa \mu_j'\big) \cdot v_j\big\|_\infty$$

$$= \left\| \sum_{n=1}^N \phi_4\Big(\phi_1(q), \phi_{2,n}\Big(\frac{q}{A}\Big)\Big)\phi_5\Big(v_j, \phi_1(k_j), \phi_{3,n}\Big(\frac{k_j}{A}\Big)\Big) - v_j \cdot \exp\big(\kappa\mu_j'\big) \right\|_\infty$$

$$\leq \left\| \sum_{n=1}^N \phi_4\Big(\phi_1(q), \phi_{2,n}\Big(\frac{q}{A}\Big)\Big)\phi_5\Big(v_j, \phi_1(k_j), \phi_{3,n}\Big(\frac{k_j}{A}\Big)\Big) \right.$$

$$\left. - v_j \cdot \phi_1(q)\phi_1(k_j)\sum_{n=1}^N \phi_{2,n}\Big(\frac{q}{A}\Big)\phi_{3,n}\Big(\frac{k_j}{A}\Big) \right\|_\infty$$

$$+ \left\| v_j \cdot \phi_1(q)\phi_1(k_j)\sum_{n=1}^N \phi_{2,n}\Big(\frac{q}{A}\Big)\phi_{3,n}\Big(\frac{k_j}{A}\Big) - v_j \cdot \exp\big(\kappa\mu_j'\big) \right\|$$

$$\lesssim \frac{1}{(V+1)^2}\epsilon^9 \exp(-8(\kappa d' + \delta/2)),$$

where $\Phi_1(k_j, v_j) \in \mathbb{R}^{d' \times N}$ and $\psi(q) \in \mathbb{R}^N$. Summing up the error for $j = -V, \ldots, 0$, we have

$$\left\| \Big(\sum_{j=-V}^0 \Phi_1(k_j, v_j)\Big)\psi_1(q) - \sum_{j=-V}^0 \exp\big(\kappa\mu_j'\big)\cdot v_j \right\|_\infty \lesssim \frac{1}{V+1}\epsilon^9 \exp(-8(\kappa d' + \delta/2)).$$

Similarly, there exist neural networks $\psi_2, \psi_3 \in \Psi(L, W, S, B)$ with

$$L \lesssim \frac{d'^8}{\delta^5}\big(\log^5 \epsilon^{-1} + \log^5 V\big), \quad W \lesssim \frac{d'}{\delta}\big(\log \epsilon^{-1} + \log V\big),$$

$$S \lesssim \frac{d'^8}{\delta^5}\big(\log^5 \epsilon^{-1} + \log^5 V\big), \quad \log B \lesssim \frac{d'^3}{\delta^2}\big(\log^3 \epsilon^{-1} + \log^3 V\big),$$

such that

$$\left| \psi_3(q)^\top \sum_{j=-V}^0 \psi_2(k_j) - \sum_{j=-V}^0 \exp\big(\kappa\mu_j'\big)\cdot v_j \right| \lesssim \frac{1}{V+1}\epsilon^9 \exp(-8(\kappa d' + \delta/2)),$$

where $\psi_2(k_j), \psi_3(q) \in \mathbb{R}^N$. Note that $\Phi_1(k_j, v_j) \in \mathbb{R}^{d' \times N}$ and $\psi_2(k_j) \in \mathbb{R}^N$ are the output of the convolution layer.

From (mult) in Lemma F.3, we can see that there exists a neural network $\phi_6 \in \Psi(L, W, S, B)$ with

$$L \lesssim \frac{d'}{\delta}\big(\log \epsilon^{-1} + \log V\big), \quad W \lesssim 1,$$

$$S \lesssim \frac{d'}{\delta}\big(\log \epsilon^{-1} + \log V\big), \quad \log B \lesssim 1,$$

such that $|\phi_6(x, y) - Xy| \lesssim \frac{1}{V+1}\epsilon^9 \exp(-8(\kappa d' + \delta/2))$ for any $X \in \mathbb{R}^{d' \times N}, y \in \mathbb{R}^N$ with $\|X\|_\infty, \|y\|_\infty \leq (V+1)\Big(\frac{32\kappa d'}{\delta}\Big)^{d'N}\exp\Big(\frac{\kappa d'}{2}\Big)$. Similarly, we have the network $\phi_7$ that approximates $x^\top y$ for any $x, y \in \mathbb{R}^N$ with $\|x\|_\infty, \|y\|_\infty \leq (V+1)\Big(\frac{32\kappa d'}{\delta}\Big)^{d'N}\exp\Big(\frac{\kappa d'}{2}\Big)$. Then, we have

$$\left\| \phi_6\Big(\sum_{j=-V}^0 \Phi_1(k_j, v_j), \psi_1(q)\Big) - \Big(\sum_{j=-V}^0 \Phi_1(k_j, v_j)\Big)\psi_1(q) \right\|_\infty \lesssim \frac{1}{V+1}\epsilon^9 \exp(-8(\kappa d' + \delta/2)),$$

$$\left| \phi_7\Big(\psi_3(q), \sum_{j=-V}^0 \psi_2(k_j)\Big) - \psi_3(q)^\top \sum_{j=-V}^0 \psi_2(k_j) \right| \lesssim \frac{1}{V+1}\epsilon^9 \exp(-8(\kappa d' + \delta/2)).$$

Now, from Lemma F.4, there exists a neural network $\phi_8 \in \Psi(L, W, S, B)$ with

$$L \lesssim \frac{d'^2}{\delta^2}\left(\log^2 \epsilon^{-1} + \log^2 V\right), \quad W \lesssim \frac{d'^3}{\delta^3}\left(\log^3 \epsilon^{-1} + \log^3 V\right),$$

$$S \lesssim \frac{d'^4}{\delta^4}\left(\log^4 \epsilon^{-1} + \log^4 V\right), \quad \log B \lesssim \frac{d'}{\delta}\left(\log \epsilon^{-1} + \log V\right),$$

such that

$$\left|\phi_8(x', y') - \frac{y'}{x'}\right| \lesssim \epsilon + \frac{|x - x'|}{\tau^8} + \frac{|y - y'|}{\tau^2}$$

for any $x, y, x', y' \in \mathbb{R}$ with $x \in [\tau, \tau^{-1}]$ and $y \in [0, \tau^{-1}]$, where

$$\tau := \min\left\{\epsilon, \exp(-(\kappa d' + \delta/2))/(V + 1)\right\}$$

Since it holds

$$\sum_{j=-V}^{0} \exp(\kappa \mu_j) \cdot v_{ji} \leq \sum_{j=-V}^{0} \exp\left(\kappa \mu_j'\right) \leq (V + 1) \exp(\kappa d' + \delta/2),$$

for any $i \in [d']$, and

$$\sum_{j=-V}^{0} \exp(\kappa \mu_j) \geq \exp(\kappa \mu_{j*}) \geq \exp(-\kappa d' - \delta/2),$$

we have

$$\left\|\phi_8\left(\phi_6\left(\sum_{j=-V}^{0} \Phi_1(k_j, v_j), \psi_1(q)\right), \phi_7\left(\psi_3(q), \sum_{j=-V}^{0} \psi_2(k_j)\right)\right) - \frac{\sum_{j=-V}^{0} \exp\left(\kappa \mu_j'\right) \cdot v_j}{\sum_{j=-V}^{0} \exp\left(\kappa \mu_j'\right)}\right\|_\infty$$

$$\lesssim \epsilon + \frac{\epsilon^9 \exp(-8(\kappa d' + \delta/2))/(V + 1)}{\tau^8} + \frac{\epsilon^9 \exp(-8(\kappa d' + \delta/2))/(V + 1)}{\tau^2} \lesssim \epsilon.$$

which completes the proof.

# H  PROOF OF THEOREM 3.1, 3.2, D.1 AND D.2

## H.1  CONSTRUCTING THE EMBEDDINGS

To construct the embeddings, we use the following lemma.

**Lemma H.1.** *Let $S$ be an arbitrary set of $n$ points in $\mathbb{R}^d$, and $m \geq 1$ be a integer. Suppose that, for any $x \in S$, it holds $\|x\|_2 \leq 1$. Then, there exists a matrix $R \in \mathbb{R}^{k \times d}$ with $k \leq 512m^4 \log n + 1$ satisfying the following:*

- *Any elements of $R$ is $+1/\sqrt{k}$ or $-1/\sqrt{k}$.*

- *For any $x, y \in S$, it holds $\left|(Rx)^\top(Ry) - x^\top y\right| \leq \frac{1}{8m^2}$.*

To prove Lemma H.1, we use the following proposition.

**Proposition H.2** (Theorem 1.1 in Achlioptas (2003))**.** *Let $P$ be an arbitrary set of $n$ points in $\mathbb{R}^d$. Given $\epsilon, \beta > 0$, let $k$ be an integer such that $k \geq \frac{4+2\beta}{\epsilon^2-\epsilon^3} \log n$. Let $R$ be a $k \times d$ matrix whose entries are independent random variables drawn from the uniform distribution on $\left\{1/\sqrt{k}, -1/\sqrt{k}\right\}$. Then, with probability at least $1 - n^{-\beta}$, for any $x, y \in P$,*

$$(1 - \epsilon)\|x - y\|_2^2 \leq \|Rx - Ry\|_2^2 \leq (1 + \epsilon)\|x - y\|_2^2.$$

*Proof of Lemma H.1.* We apply Proposition H.2 with

$$P = S \cup \{-s \mid s \in S\}, \quad \epsilon = \frac{1}{8m^2}, \quad \beta = 1.$$

Let $k$ be an integer such that

$$k \geq 512m^4 \log n \left(\geq \frac{4 + 2\beta}{\epsilon^2 - \epsilon^3} \log n\right),$$

and $R$ be the random matrix defined in Proposition H.2. With probability at least $1 - \frac{1}{2n}$, for any $x, y \in P$,

$$\left| \|Rx - Ry\|_2^2 - \|x - y\|_2^2 \right| \leq \frac{1}{8m^2} \|x - y\|_2^2, \tag{H.1}$$

$$\left| \|Rx + Ry\|_2^2 - \|x + y\|_2^2 \right| \leq \frac{1}{8m^2} \|x + y\|_2^2. \tag{H.2}$$

Since $1 - \frac{1}{2n} \geq \frac{1}{2^{kd}}$ for any positive integer $n, k, d$, we can choose $R$ such that (H.1) holds. For such $R$, it holds

$$
\begin{aligned}
(Rx)^\top (Ry) - x^\top y &= \frac{1}{4} \left( \|Rx + Ry\|_2^2 - \|Rx - Ry\|_2^2 - \|x + y\|_2^2 + \|x + y\|_2^2 \right) \\
&\leq \frac{1}{4} \left( \left| \|Rx + Ry\|_2^2 - \|x + y\|_2^2 \right| + \left| \|Rx - Ry\|_2^2 - \|x - y\|_2^2 \right| \right) \\
&\leq \frac{1}{4} \left( \frac{1}{4m^2} \|x + y\|_2^2 + \frac{1}{4m^2} \|x - y\|_2^2 \right) \\
&= \frac{1}{16m^2} \|x\|_2^2 + \frac{1}{16m^2} \|y\|_2^2 \\
&\leq \frac{1}{8m^2},
\end{aligned}
$$

which completes the proof. $\qquad \square$

## H.2 PROOF OF THE THEOREMS

The following is the essential lemma to prove the theorems.

**Lemma H.3.** *Let $m \in \mathbb{N}_{>0}$ and $Z = [z_{-V}, \ldots, z_0] \in [-1, 1]^{|\mathcal{W}| \times [-V : 0]}$ be a sequence of one-hot vector representing the alphabets in set $S$. Suppose that there uniquely exists $j^* \in [-V : -1]$ such that*

$$[z_{j^*-m}, z_{j^*-m+1}, \ldots, z_{j^*-1}] = [z_{-m+1}, z_{-m+2}, \ldots, z_0],$$

*where $z_j = 0$ for $j \notin [-m+1 : 0]$. Then, there exists $F \in \mathcal{S}(M, U, D, L, W, S, B)$ and $W \in \mathbb{R}^{|\mathcal{W}| \times D}$ with*

$$M = 2, \quad U = V, \quad D = m^{14} \log^2 V \log^3 |\mathcal{W}|, \quad L \lesssim m^{37} \log^5 V \log^8 |\mathcal{W}|,$$

$$W \lesssim m^{15} \log^3 V \log^3 |\mathcal{W}|, \quad S \lesssim m^{37} \log^5 V \log^8 |\mathcal{W}|, \quad \log B \lesssim m^{10} \log^2 V \log^2 |\mathcal{W}|,$$

*such that $j^* = \arg\max_{j=-V,\ldots,0} (W \cdot F(Z)_0)_j$.*

*Proof.* Set $\kappa \sim V^2(\log m + \log V + \log\log |\mathcal{W}|)$. Additionally, let us set embedding $E_1 = R \in \mathbb{R}^{D \times |\mathcal{W}|}$ as in Lemma H.1, and set $E_2 = 0$. We define $x_j = Rz_j$ for $j = -V, \ldots, 0$. The third item of Lemma F.5, we can see that there exists $\psi_n \in \Psi'(D, B)$ with

$$N \lesssim \log^2 m + \log^2 V + \log^2 \log |\mathcal{W}|, \quad D = 1, \quad \log B \lesssim \log^2 m + \log^2 V + \log^2 \log |\mathcal{W}|,$$

such that

$$\left\| \exp\left( -\kappa \cdot \sin^2\left( \frac{\pi \cdot (j - k)}{2(V + 1)} \right) \right) - \sum_{n=1}^{N} \psi_n\left( \frac{j}{V + 1} \right) \right\|_\infty \leq \frac{1}{8m^2 d(V + 1)}.$$

for any $j = -V, \ldots, 0$. Since

$$\left| \exp\left( -\kappa \cdot \sin^2\left( \frac{\pi \cdot (k - j)}{2(V + 1)} \right) \right) - \delta_{j,k} \right| \leq \frac{1}{8m^2 d(V + 1)},$$

it holds

$$\left\| \sum_{j=-V}^{0} \sum_{n=1}^{N} \psi_n\left( \frac{j}{V + 1} \right) \cdot x_j - x_k \right\|_\infty \leq \frac{1}{4m^2 d}.$$

Therefore, there exists $g_1 \in \mathcal{C}(U, D, B)$ with

$$U = V, \quad D = m, \quad \log B \lesssim m^2 \log V,$$

such that

$$g(X) = \begin{bmatrix} q'_{-V} & q'_{-V+1} & \cdots & q'_0 \\ k'_{-V} & k'_{-V+1} & \cdots & k'_0 \\ v'_{-V} & v'_{-V+1} & \cdots & v'_0 \end{bmatrix},$$

$$\left\| \begin{bmatrix} q'_{-V} & q'_{-V+1} & \cdots & q'_0 \\ k'_{-V} & k'_{-V+1} & \cdots & k'_0 \\ v'_{-V} & v'_{-V+1} & \cdots & v'_0 \end{bmatrix} - \begin{bmatrix} q_{-V} & q_{-V+1} & \cdots & q_0 \\ k_{-V} & k_{-V+1} & \cdots & k_0 \\ v_{-V} & v_{-V+1} & \cdots & v_0 \end{bmatrix} \right\|_\infty \le \frac{1}{4m^2 d},$$

where

$$q_j = \left[ x_{j-m+1}^\top, x_{j-m+2}^\top, \ldots, x_j^\top \right]^\top,$$
$$k_j = \left[ x_{j-m+1}^\top, x_{j-m+2}^\top, \ldots, x_j^\top \right]^\top,$$
$$v_j = x_j,$$

and $x_j = 0$ for $j \notin [-m+1 : 0]$. Then, it holds,

$$q'_0{}^\top k'_{j^*} \ge q_0^\top k_{j^*} - md \cdot \frac{1}{4m^2 d} \ge 1 - \frac{1}{8m^2} \cdot m - \frac{1}{4m} = 1 - \frac{3}{8m},$$

and, for any $j \in [-V : 0] \setminus \{j^*\}$,

$$q'_0{}^\top k'_j \le q_0^\top k_j + md \cdot \frac{1}{4m^2 d} \le \frac{m-1}{m} + \frac{1}{8m^2} \cdot m + \frac{1}{4m} = 1 - \frac{5}{8m}.$$

Therefore, due to Lemma 3.3, there exists $f_1, f_2 \in \Psi(L, W, S, B)$ and $g_2 \in \mathcal{C}(U, D, B)$ with

$$U = V, \quad D = m^{14} \log^2 V \log^3 |\mathcal{W}|, \quad L \lesssim m^{37} \log^5 V \log^8 |\mathcal{W}|,$$

$$W \lesssim m^{15} \log^3 V \log^3 |\mathcal{W}|, \quad S \lesssim m^{37} \log^5 V \log^8 |\mathcal{W}|, \quad \log B \lesssim m^{14} \log^2 V \log^2 |\mathcal{W}|,$$

such that

$$\| f_2 \circ g_2 \circ f_1 \circ g_1(X) - x_{j^*} \|_\infty \le \frac{1}{4}.$$

Therefore, $\left\| R^\top (F(Z)_0) - z_{j^*} \right\|_\infty \le \frac{1}{4} + \frac{1}{8m^2}$, which completes the proof. □

Now, we prove Theorem 3.1, Theorem 3.2 and Theorem D.1.

*Proof of Theorem 3.1.* Due to Lemma 2.4 of Jelassi et al. (2024), if we set $m \lesssim \log(V/\epsilon)/\log|\mathcal{W}|$ in Lemma H.3, we can achieve $\mathrm{err}_V \le \epsilon$. Therefore, the result follows. □

*Proof of Theorem 3.2.* Applying $m = 1$ and $V \le |\mathcal{W}|$ directly gives the result. □

*Proof of Theorem D.1.* The proof is completely the same as the proof of Theorem 3.2. Note that associative recall is the special case of induction heads where the set keys and the set queries are completely split. □

Finally, we prove Theorem D.2.

*Proof of Theorem D.2.* The probability of having $M$ or more consecutive $\langle \mathrm{PAD} \rangle$ tokens in the input sequence is at most $V \cdot \alpha^M$. Therefore, if $M \sim \log V + \log \epsilon^{-1}$, this probability becomes less than $\epsilon/2$. Hence, in the following discussion, we consider situations where $\langle \mathrm{PAD} \rangle$ does not appear consecutively $M$ times or more.

Let "$s_0, s_1, \ldots, s_V$" be the input sequence, and take an arbitrary index $i \in [V]$. Fix a positive integer $K$. Due to the definition of $M$, if we set $m = KM$, there are at least $K$ tokens that are not $\langle \mathrm{PAD} \rangle$ in the sequence $[s_{i-m+1}, s_{i-m+2}, \ldots, s_i]$.

Now, let us upper bound the probability that there exists $i \ne j$ $(i, j \in [V])$ such that two sequences $[s_{i-m+1}, s_{i-m+2}, \ldots, s_i]$ and $[s_{j-m+1}, s_{j-m+2}, \ldots, s_j]$ have common $K$ elements (including duplicates) that are not $\langle \mathrm{PAD} \rangle$. First, we have $|\mathcal{W}|^K$ choices for the common $K$ elements. Then, we have $(m!/(m-k)!)^2$ choices for the positions of the common $K$ elements in the sequence of length

$m$. Each choice of the positions occurs with probability at most $1/|\mathcal{W}|^{2K}$. Moreover, there are at most $V^2$ choices for the indices $i$ and $j$. Therefore, the probability can be upper bounded by

$$
\begin{aligned}
V^2 \cdot |\mathcal{W}|^K \cdot \frac{m!^2}{(m-K)!^2} \cdot \frac{1}{|\mathcal{W}|^{2K}} &\leq \frac{V^2}{|\mathcal{W}|^K} \cdot \frac{m!^2}{(m-K)!^2} \\
&\leq \frac{V^2}{|\mathcal{W}|^K} \cdot \frac{e^2(m+1/e)^{2(m+1)}}{e^2((m-K)/e)^{2(m-K)}} \\
&\leq \frac{V^2}{|\mathcal{W}|^K} \cdot \frac{(m+1)^{2(m+1)}}{e^{2(m+1)}} \cdot \frac{e^{2(m-K)}}{(m-K)^{2(m-K)}} \\
&\leq \frac{V^2}{e^2} \cdot \left( \frac{(MK-K)^2}{e^2|\mathcal{W}|} \right)^K \cdot \frac{(m+1)^{2(m+1)}}{(m-K)^{2m}} \\
&\leq \frac{V^2(KM+1)^2}{e^2} \cdot \left( \frac{K^2(M-1)^2}{e^2|\mathcal{W}|} \right)^K \cdot \left( \frac{KM+1}{KM-K} \right)^{2KM} \\
&\leq \frac{V^2(KM+1)^2}{e^2} \cdot \left( \frac{K^2(M-1)^2}{e^2|\mathcal{W}|} \cdot \left( \frac{M+1}{M-1} \right)^M \right)^K \\
&\leq \frac{V^2(KM+1)^2}{e^2} \cdot \left( \frac{C \cdot K^2(M-1)^2}{|\mathcal{W}|} \right)^K,
\end{aligned}
$$

where $C > 0$ is a universal constant. Therefore, if $|\mathcal{W}| \geq 2C \cdot K^2(M-1)^2$ and $K \sim \log \epsilon^{-1} + \log V$, the probability is less than $\epsilon/2$.

Let us consider the situation where the input "$s_0, s_1, \ldots, s_L$" is fed into the model. Let us set embedding $E_1 = R \in \mathbb{R}^{D \times |\mathcal{W}|}$ as in Lemma H.1, and set $E_2 = 0$. Additionally, let $x_j$ be the embedding of $s_j$ for $j = 0, \ldots, L$. Let us consider the model that finds $s_I$ with the index $I \in [L]$ such that the partial sequence $[s_{I-m+1}, s_{I-m+2}, \ldots, s_I]$ and $[s_{L-m+1}, s_{L-m+2}, \ldots, s_L]$ have the same $K$ elements that are not $\langle \text{PAD} \rangle$.

The similar discussion as in the proof of Lemma H.3 reveals that there exists $\psi_n \in \Psi'(D, B)$ with

$$
N \lesssim \log^2 m + \log^2 V, \quad D = 1, \quad \log B \lesssim \log^2 m + \log^2 V,
$$

such that

$$
\left\| \sum_{j=0}^{L} \sum_{n=1}^{N} \psi_n \left( \frac{j}{V+1} \right) \cdot x_j - \frac{1}{m} \sum_{j=L-m+1}^{L} x_j \right\|_\infty \leq \frac{1}{4m^2}.
$$

Therefore, there exists $g_1 \in \mathcal{C}(U, D, B)$ with

$$
U = L, \quad D \lesssim \log^2 m + \log^2 V, \quad \log B \lesssim \log^2 m + \log^2 V,
$$

such that

$$
g_1(X) = \begin{bmatrix} q'_0 & q'_1 & \cdots & q'_L \\ k'_0 & k'_1 & \cdots & k'_L \\ v'_0 & v'_1 & \cdots & v'_L \end{bmatrix},
$$

$$
\left\| \begin{bmatrix} q'_0 & q'_1 & \cdots & q'_L \\ k'_0 & k'_1 & \cdots & k'_L \\ v'_0 & v'_1 & \cdots & v'_L \end{bmatrix} - \begin{bmatrix} q_0 & q_1 & \cdots & q_L \\ k_0 & k_1 & \cdots & k_L \\ v_0 & v_1 & \cdots & v_L \end{bmatrix} \right\|_\infty \leq \frac{1}{4m^2},
$$

where

$$
q_t = \frac{1}{m} \sum_{j=t-m+1}^{t} x_j, \quad k_t = \frac{1}{m} \sum_{j=t-m+1}^{t} x_j, \quad v_t = x_t,
$$

Then, if $[s_{j-m+1}, s_{j-m+2}, \ldots, s_j]$ and $[s_{L-m+1}, s_{L-m+2}, \ldots, s_L]$ have the common $K$ elements, it holds

$$
q'^\top_L k'_j \geq \frac{K}{m^2} - \frac{1}{4m^2}.
$$

Moreover, if the number of common elements in $[s_{j-m+1}, s_{j-m+2}, \ldots, s_j]$ and $[s_{L-m+1}, s_{L-m+2}, \ldots, s_L]$ is less than $K$, it holds

$$
q'^\top_L k'_j \leq \frac{K-1}{m^2} + \frac{1}{4m^2} = \frac{K}{m^2} - \frac{3}{4m^2}.
$$

Therefore, due to Lemma 3.3, there exists $f_1, f_2 \in \Psi(L, W, S, B)$ and $g_2 \in \mathcal{C}(U, D, B)$ with

$$U = V, \quad D = m^{16} \log^2 V \log^3 |\mathcal{W}|, \quad L \lesssim m^{42} \log^5 V \log^8 |\mathcal{W}|,$$

$$W \lesssim m^{18} \log^3 V \log^3 |\mathcal{W}|, \quad S \lesssim m^{42} \log^5 V \log^8 |\mathcal{W}|, \quad \log B \lesssim m^{16} \log^2 V \log^2 |\mathcal{W}|,$$

such that

$$\|f_2 \circ g_2 \circ f_1 \circ g_1(X) - x_I\|_\infty \le \frac{1}{4}.$$

Therefore, $\left\|R^\top (F(Z)_0) - x_I\right\|_\infty \le \frac{1}{4} + \frac{1}{8m^2}$, which completes the proof. $\qquad\square$

# I  PROOF OF THEOREM 4.5

## I.1  PREPARATION: APPROXIMATION OF $\gamma$-SMOOTH FUNCTIONS

Before proving Theorem 4.5, we prove the following theorem Theorem I.2 under Assumption I.1 about the approximation of $\gamma$-smooth functions.

**Assumption I.1.** *The true function $F^\circ$ satisfies $F_0^\circ \in \mathcal{F}_{p,\theta}^\gamma$, where $\gamma$ is mixed or anisotropic smoothness. Suppose that it holds $\|F\|_{\mathcal{F}_{p,\theta}^\gamma} \le 1$ and $\|F_0^\circ\|_\infty \le R$, where $R > 0$ is a constant. Additionally, we assume the smoothness parameter $a$ satisfies $\|a\|_{wl^\alpha} \le 1$ for some $0 < \alpha < \infty$ and $a_{ij} = \Omega(\log(|j| + 1))$. Moreover, if $\gamma$ is mixed smoothness, we assume $\bar{a}_1 < \bar{a}_2$.*

**Theorem I.2.** *Suppose that target function $F^\circ$ satisfies Assumption I.1. Then, for any $T > 0$, there exists an SSM $F \in \mathcal{S}(M, U, D, L, W, S, B)$ with*

$$M = 1, \quad \log U \sim T, \quad D \sim T^{1/\alpha}, \quad L \sim T, \quad W \sim T^{1/\alpha},$$

$$W' \sim T^{1/\alpha} 2^{T/a^\dagger}, \quad S \sim T^{2/\alpha} \max\left\{T^{2/\alpha}, T^2\right\} 2^{T/a^\dagger}, \quad \log B \sim T^{1/\alpha}, \tag{I.1}$$

*such that $\|F - F^\circ\|_{2, P_X} \lesssim 2^{-T}$.*

Given a smoothness function $\gamma\colon \mathbb{N}_0^{d \times \infty} \to \mathbb{R}$, we define

$$I(T, \gamma) \coloneqq \{(i, j) \mid \exists s \in \mathbb{N}_0^{d \times \infty} \text{ such that } s_{ij} \neq 0, \gamma(s) < T\},$$

$$d_{\max} \coloneqq |I(T, \gamma)|.$$

The *feature extraction map* $\Gamma\colon \mathbb{R}^{d \times \infty} \to \mathbb{R}^{d_{\max}}$ is defined as

$$\Gamma(X) = [X_{i_1, j_1}, \ldots, X_{i_{d_{\max}}, j_{d_{\max}}}].$$

The following lemma shows that, if FNN receives finite number of "important" features, it can approximate $\gamma$-smooth functions and piecewise $\gamma$-smooth functions. This is mainly due to the condition $\|a\|_{wl^\alpha} \le 1$, which induces sparsity of important features.

**Lemma I.3** (Theorem D.3 in Takakura & Suzuki (2023)). *Suppose that the target functions $f \in \mathcal{F}_{p,\theta}^\gamma$ and $g \in \mathcal{P}_{p,\theta}^\gamma$ satisfy $\|f\|_\infty \le R$ and $\|g\|_\infty \le R$, where $R > 0$ and $\gamma$ is the mixed or anisotropic smoothness and the smoothness parameter $a$ satisfies $\|a\|_{wl^\alpha} \le 1$. For any $T > 0$, there exist FNNs $\hat{f}_T, \hat{g}_T \in \Psi(L, W, S, B)$ such that*

$$\left\|\hat{f}_T \circ \Gamma - f\right\|_{2, P_X} \lesssim 2^{-T},$$

$$\|\hat{g}_T \circ \Gamma \circ \Pi - g\|_{2, P_X} \lesssim 2^{-T},$$

*where*

$$L \sim \max\left\{T^{2/\alpha}, T^2\right\}, W \sim T^{1/\alpha} 2^{T/a^\dagger},$$

$$S \sim T^{2/\alpha} \max\left\{T^{2/\alpha}, T^2\right\} 2^{T/a^\dagger}, \log B \sim T^{1/\alpha}.$$

From this lemma, we can see that, if the the convolution layer can approximate $\Gamma$, the SSM can give important features to the FNN, and the FNN can approximate the target function.

Now, we prove Theorem I.2.

*Proof of Theorem I.2.* Firstly, we construct the embedding layer $\mathrm{Emb}\colon \mathbb{R}^{d\times\infty} \to \mathbb{R}^{D\times\infty}$. Set the embedding dimension $D$ as $\max\{d, d_{\max}\} + 1$. We set $E_1 \in \mathbb{R}^{D\times d}$ to satisfy

$$E_1 x = [x_1, \ldots, x_d, 0, \underbrace{0, \ldots, 0}_{D-d-1 \text{ elements}}]^\top.$$

for $x = [x_1, \ldots, x_d] \in \mathbb{R}^d$. Additionally, we set $E_2 \in \mathbb{R}^D$ to satisfy

$$E_2 = [\underbrace{0, \ldots, 0}_{d \text{ elements}}, 1, \underbrace{0, \ldots, 0}_{D-d-1 \text{ elements}}]^\top.$$

Note that $\|E_1\|_\infty = \|E_2\|_\infty = 1$. Then, the constructed embedding layer $\mathrm{Emb}$ is represented as follows:

$$\mathrm{Emb}(X) = \begin{bmatrix} \cdots & x_t & \cdots \\ \cdots & 1 & \cdots \\ \cdots & 0 & \cdots \\ \vdots & \vdots & \vdots \\ \cdots & 0 & \cdots \end{bmatrix} \in \mathbb{R}^{D\times\infty}.$$

Secondly, we construct the convolution layer. The role of this layer is to approximate the feature extractor $\Gamma$. The weight matrix $W^V \in \mathbb{R}^{D\times|X|}$ is set to extract the important "dimensions" $(i_1, \ldots, i_{d_{\max}})$. More precisely, we set $W_V$ to satisfy

$$W^V y = [y_{i_1}, \ldots, y_{i_{d_{\max}}}, \underbrace{0, \ldots, 0}_{D-d_{\max} \text{ elements}}] \in \mathbb{R}^D$$

for $y = [y_1, \ldots, y_D] \in \mathbb{R}^D$. Then, the resulted projection is represented as follows:

$$W^V(\mathrm{Emb}(X)) = \begin{bmatrix} \cdots & X_{t,i_1} & \cdots \\ \vdots & \vdots & \vdots \\ \cdots & X_{t,i_{d_{\max}}} & \cdots \\ \cdots & 0 & \cdots \\ \vdots & \vdots & \cdots \\ \cdots & 0 & \cdots \end{bmatrix} \in \mathbb{R}^{D\times\infty}.$$

Next, we construct the convolution filter. From the assumption $a_{ij} = \Omega(\log(|j|+1))$, we can choose the window size $U \in \mathbb{N}$ such that

$$\log U \sim T \quad \text{and} \quad a_{ij} \leq T \implies j \leq U.$$

Lemma F.5 shows that, for each $j_m$ ($m = 1, \ldots, d_{\max}$), for any $\epsilon > 0, \kappa > 0$, there exists $k_m \in \Psi'(W', B)$ with

$$W' \lesssim \log^2 \epsilon^{-1}, \quad B \lesssim \log \epsilon^{-1} \log \kappa$$

such that

$$\max_{j=0,\ldots,U} \left| k_m\left(\frac{j}{U}\right) - \exp\left(-\kappa \cdot \sin^2\left(\frac{\pi}{2}\left(\frac{j}{U} - \frac{j_m}{U}\right)\right)\right) \right| \lesssim \epsilon.$$

Now, if $|j - j_m| \geq 1$, it holds

$$\exp\left(-\kappa \cdot \sin^2\left(\frac{\pi}{2}\left(\frac{j}{U} - \frac{j_m}{U}\right)\right)\right) \leq \exp\left(-\kappa \cdot \left(\frac{2}{\pi} \cdot \frac{\pi}{2} \cdot \frac{1}{U}\right)^2\right) = \exp\left(-\frac{\kappa}{U^2}\right),$$

and, if $j = j_m$, it holds

$$\exp\left(-\kappa \cdot \sin^2\left(\frac{\pi}{2}\left(\frac{j}{U} - \frac{j_m}{U}\right)\right)\right) = 1.$$

Therefore, if we set $\kappa = U^2 \log \epsilon^{-1}$, we have

$$\max_{j=0,\ldots,U} \left| k_m\left(\frac{j}{U}\right) - \delta_{j_m}(j) \right| \lesssim 2\epsilon,$$

where $\delta_{j'}$ is the function defined by

$$\delta_{j'}(j) = \begin{cases} 1 & \text{if } j = j', \\ 0 & \text{otherwise.} \end{cases}$$

This inequality show that the filter $k$ can approximately extract the important tokens.

Finally, we set the weight matrix $W^Q$ by

$$W_{i,j}^Q = \begin{cases} 1 & \text{if } j = d+1 \\ 0 & \text{otherwise,} \end{cases}$$

which results in $W^Q(\text{Emb}(X)) = [1, \ldots, 1]^\top$ and

$$g_1 \circ \text{Emb}(X) = W^Q(\text{Emb}(X)) \odot (\beta^{(1)} * W^{(0)}(\text{Emb}(X)))$$

$$= \beta^{(1)} * W^{(0)}(\text{Emb}(X))$$

$$= [z_t]_{t=-\infty}^0 \in \mathbb{R}^{D \times \infty},$$

$$z_t = \sum_{s=0}^{U-1} k(s) \sum_{i=1}^{D} W_{i,t-s}^{(0)} X_{i,t-s}$$

$$= \begin{bmatrix} \sum_{s=0}^{U-1} (k(s))_1 X_{i_1, t-s} \\ \vdots \\ \sum_{s=0}^{U-1} (k(s))_{d_{\max}} X_{i_{d_{\max}}, t-s} \\ 0 \\ \vdots \\ 0 \end{bmatrix}.$$

Thirdly, we construct the FNN layer. From Lemma I.3, there exists an FNN $\hat{f} \in \Psi(L, W, S, B)$ such that

$$\left\| \hat{f} \circ \Gamma - F^\circ \right\|_{2, P_X} \lesssim 2^{-T}, \tag{I.2}$$

where

$$L \sim \max\left\{ T^{2/\alpha}, T^2 \right\}, W \sim T^{1/\alpha} 2^{T/a^\dagger},$$
$$S \sim T^{2/\alpha} \max\left\{ T^{2/\alpha}, T^2 \right\} 2^{T/a^\dagger}, \log B \sim T^{1/\alpha}. \tag{I.3}$$

Let $C \colon \mathbb{R}^D \to \mathbb{R}^d$ be a linear map such that

$$Cy = [y_1, \ldots, y_{d_{\max}}]^\top$$

for $y = [y_1, \ldots, y_D]^\top \in \mathbb{R}^D$, and we set $f_1 := \hat{f} \circ C$. Note that $f_1 \in \Psi(L, W, S, B)$ for $L, W, S, B$ defined in (I.3). The constructed SSM $\hat{F}$ is represented as follows:

$$\hat{F}(X) = f_1(z_0) = \hat{f} \circ C(z_0) = \hat{f} \circ \hat{\Gamma}(X)_0,$$

where

$$\hat{\Gamma}(X) = \left[ \sum_{s=0}^{U-1} (k(s))_m X_{i_m, -s} \right]_{m=1}^{d_{\max}} \in \mathbb{R}^{d_{\max}}.$$

Now, we evaluate the error between the target function $F^\circ$ and the constructed model $\hat{F}$. We evaluate the error by separating into two terms:

$$\left\| \hat{F} - F^\circ \right\|_{2, P_X} \leq \left\| \hat{F} - \hat{f} \circ \Gamma \right\|_{2, P_X} + \left\| \hat{f} \circ \Gamma - F^\circ \right\|_{2, P_X}.$$

The second term can be bounded by (I.2), so we evaluate the first term. Since $\hat{f} \in \Psi(L, W, S, B)$ is $(BW)^L$-lipschitz continuous, for any $X \in [0, 1]^{d \times \infty}$, we have

$$\left| \hat{F}(X) - \hat{f} \circ \Gamma(X) \right| = \left| \hat{f}(\hat{\Gamma}(X)) - \hat{f}(\Gamma(X)) \right| \leq (BW)^L \left\| \hat{\Gamma}(X) - \Gamma(X) \right\|_\infty.$$

Since $X \in [0, 1]^{d \times \infty}$, it holds

$$\left\| \hat{\Gamma}(X) - \Gamma(X) \right\|_\infty = \max_{m=1, \ldots, d_{\max}} \left| \sum_{s=0}^{U-1} (k(s))_m X_{i_m, -s} - \delta_{j_m}(s) X_{i_m, -s} \right|$$

$$\leq \max_{m=1, \ldots, d_{\max}} \sum_{s=0}^{U-1} |(k(s))_m - \delta_{j_m}(s)|$$

$$\leq U\epsilon.$$

By setting $\epsilon = 2^{-T}/U$, we have

$$\left|\hat{F}(X) - \hat{f} \circ \Gamma(X)\right| \leq \left\|\hat{\Gamma}(X) - \Gamma(X)\right\|_\infty \leq 2^{-T}$$

for any $X \in [0,1]^{d \times \infty}$. Therefore, it holds

$$\begin{aligned}
\left\|\hat{F} - F^\circ\right\|_{2,P_X} &\leq \left\|\hat{F} - \hat{f} \circ \Gamma\right\|_{2,P_X} + \left\|\hat{f} \circ \Gamma - F^\circ\right\|_{2,P_X} \\
&\leq \sup_{X \in [0,1]^{d \times \infty}} \left|\hat{F}(X) - \hat{f} \circ \Gamma(X)\right| + \left\|\hat{f} \circ \Gamma - F^\circ\right\|_{2,P_X} \\
&\lesssim 2^{-T}.
\end{aligned}$$

Finally, we evaluate the parameters $L, W, S, B$ which controls the class of $k \in \Psi'(W', B)$. Since $\|a\|_{wl^\alpha} = \sup_j j^\alpha \bar{a}_j^{-1} \leq 1$, it holds

$$d_{\max} := \left|\left\{(i,j) \mid \exists s \in \mathbb{N}_0^{d \times \infty}, s_{ij} \neq 0, \gamma(s) < T\right\}\right| \leq T^{1/\alpha}.$$

Therefore, we have

$$W' = d_{\max} \cdot \log^2 \epsilon^{-1} \lesssim T^{2+1/\alpha},$$
$$\log B \sim \log \epsilon^{-1} \log\left(U^2 \log \epsilon^{-1}\right) \lesssim T^2.$$

This completes the proof. $\qquad\qquad\qquad\qquad\qquad\qquad\qquad\qquad\qquad\qquad\qquad\quad\square$

## I.2 PROOF OF THEOREM 4.5

*Proof of Theorem 4.5.* For $T > 0$, we define

$$I_j(T,\gamma) := \{i \mid (i,j) \in I(T,\gamma)\} = \left\{i_1^{(j)}, \ldots, i_{|I_j|}^{(j)}\right\},$$
$$r_{\max}(T,\gamma) := \max\{j \in [J] \mid I_j(T,\gamma) \neq \emptyset\},$$

Note that $r_{\max}(T,\gamma) \sim T^{1/\alpha}$ since $a_{ij} = \Omega(j^\alpha)$.

Theorem I.2 implies that there exist an embedding layer Emb, an FNN $f_1 \in \Psi(L, W, S, B)$ and a convolution layer $g_1 \in \mathcal{C}(U, D, L', W', S, B)$ with

$$M = 1, \quad \log U \sim T, \quad D \sim T^{1/\alpha},$$
$$L \sim T, \quad W_1 \sim T^{1/\alpha},$$
$$L' \sim \max\left\{T^{2/\alpha}, T^2\right\}, \quad W' \sim T^{1/\alpha} 2^{T/a^\dagger},$$
$$S \sim T^{2/\alpha} \max\left\{T^{2/\alpha}, T^2\right\} 2^{T/a^\dagger}, \quad \log B \sim T^{1/\alpha},$$

such that

$$f_1 \circ g_1 \circ \mathrm{Emb}(X)_i = [x_i^\top, \widehat{\mu}_i(X), \underbrace{0, \ldots, 0}_{d_{\max} \text{ elements}}, \underbrace{-1, \ldots, -1}_{r_{\max} \text{ elements}}]^\top,$$

for all $i \in \mathbb{Z}$, where $\widehat{\mu}_i(X)$ satisfies

$$|\widehat{\mu}_i(X)_{-t} - (\mu_i(X) - 1)| \lesssim 2^{-T}.$$

Intuitively, the $i$-th elements for $i = 3, \ldots, 2 + d_{\max}$ are used to store the feature $X_{t-i,j}$ for $j \in [d]$, and the $i$-th elements for $i = 3 + d_{\max}, \ldots, 2 + d_{\max} + r_{\max}$ are buffers to store which elements are already selected. Note that, for any $i \leq r_{\max}$, it holds

$$\widehat{\mu}(X)_{\pi_\lambda(i)} - \widehat{\mu}(X)_{\pi_\lambda(i+1)} \gtrsim (\mu(X)_{\pi_\lambda(i)} - 2^{-T}) - (\mu(X)_{\pi_\lambda(i+1)} + 2^{-T}) \gtrsim T^{-\beta/\alpha},$$

and $\widehat{\mu}(X)_t \in [-1, 0]$ for all $t \in [0 : V]$.

In the following, we set

$$U = V.$$

Let us set $\chi_T \sim \dfrac{T \log 2 + 2 \log V}{T^{-\beta/\alpha}}$. Using Lemma F.3, we see that, there exists a neural network $\phi_{\exp} \in \Psi(L, W, S, B)$ with

$$L \lesssim T^{2(1+\beta/\alpha)} \log^2 V, \quad W \lesssim T^{1+\beta/\alpha} \log V, \quad S \lesssim T^{2(1+\beta/\alpha)} \log^2 V, \quad \log B \lesssim T^{2(1+\beta/\alpha)} \log^2 V,$$

such that, for any $x \leq 0$, it holds

$$|\phi_{\exp}(\chi_T x) - \exp(\chi_T x)| \leq 2^{-2T^{1+\beta/\alpha}}/V^3.$$

Moreover, using Lemma F.3 again, we see that there exists a neural network $\phi_\times$ with
$$L \lesssim T^{2(1+\beta/\alpha)} \log^2 V, \quad W \lesssim 1, \quad S \lesssim T^{2(1+\beta/\alpha)} \log^2 V, \quad \log B \lesssim T^{1+\beta/\alpha} \log V,$$
such that, for any $0 \leq x \lesssim V^2 \exp\left(T^{1+\beta/\alpha}\right)$, $0 \leq y \lesssim 1$, it holds
$$|\phi_\times(x, y) - xy| \leq 2^{-2T^{1+\beta/\alpha}}/V^3.$$
Then, for any $x \leq 0$ and $y \in [0, 1]$, it holds
$$|\phi_\times(\phi_{\exp}(\chi_T x), y) - \exp(\chi_T x)y| \leq |\phi_\times(\phi_{\exp}(\chi_T x), y) - \phi_{\exp}(\chi_T x)y| + |\phi_{\exp}(\chi_T x)y - \exp(\chi_T x)y|$$
$$\leq 2^{-2T^{1+\beta/\alpha}}/V^3 + 2^{-2T^{1+\beta/\alpha}}/V^3$$
$$\lesssim 2^{-2T^{1+\beta/\alpha}}/V^3.$$

Then, let us define $f_1'$ be an FNN layer such that it holds
$$f_1' \circ f_1 \circ g_1(X) \circ \mathrm{Emb}(X)_i = [\phi_\times(\phi_{\exp}(\widehat{\mu}_i(X)), x_i), \underbrace{0, \ldots, 0}_{d_{\max} \text{ elements}}, \underbrace{-1, \ldots, -1}_{r_{\max} \text{ elements}}]^\top.$$

Additionally, we define
$$Z_m \coloneqq (f_m' \circ g_m \circ f_m) \circ \cdots \circ (f_1' \circ g_1 \circ f_1) \circ \mathrm{Emb}(X),$$
for $m \in [1 : r_{\max}]$. We construct remaining layers $f_2, g_2, f_2', \ldots, f_{r_{\max}+1}, g_{r_{\max}+1}, f_{r_{\max}+1}'$ to make them satisfying
$$Z_m = [\phi_\times(\phi_{\exp}(\widehat{\mu}_i(X)), x_i), \widehat{X}_{i_1^{(1)}, j_1}, \ldots, \widehat{X}_{i_{|I_1|}^{(1)}, j_1}, \ldots, \widehat{X}_{i_1^{(m)}, j_m}, \ldots, \widehat{X}_{i_{|I_m|}^{(m)}, j_m}, \underbrace{0, \ldots, 0}_{d_{\max} - \sum_{j=1}^m |I_j| \text{ elements}},$$
$$\widehat{j}_1/V, \ldots, \widehat{j}_m/V, \underbrace{-1, \ldots, -1}_{r_{\max} - m \text{ elements}}]^\top,$$
where $\widehat{X}_{i_k^{(jm)}, j_m}, \widehat{j}_m$ are the approximation of $\widehat{X}_{i_k^{(jm)}, j_m}, \widehat{j}_m$ $(m = 1, \ldots, M; k = 1, \ldots, |I_{j_m}|)$ respectively such that
$$\left|\widehat{X}_{i_k^{(jm)}, j_m} - X_{i_k^{(jm)}, j_m}\right| \lesssim 2^{-T}, \quad \left|\widehat{j}_m/V - j_m/V\right| \lesssim 2^{-3T^{1+\beta/\alpha}}/V^5.$$
Then, we see that
$$Z_M = [x_i^\top, \widehat{\mu}_i(X), \widehat{X}_{i_1^{(1)}, j_1}, \ldots, \widehat{X}_{i_{|I_1|}^{(1)}, j_1}, \ldots, \widehat{X}_{i_1^{(r_{\max})}, j_{r_{\max}}}, \ldots, \widehat{X}_{i_{|I_{r_{\max}}|}^{(r_{\max})}, j_{r_{\max}}}, \widehat{j}_1/V, \ldots, \widehat{j}_M/V]^\top.$$
Hence, Lemma I.3 shows that there exists a FNN $f_M' \in \Psi(L, W, S, B)$ with
$$L \lesssim \max\left\{T^{2/\alpha}, T^2\right\}, \quad W \lesssim T^{1/\alpha} 2^{T/\alpha^\dagger},$$
$$S \lesssim T^{2/\alpha} \max\left\{T^{2/\alpha}, T^2\right\} 2^{T/\alpha^\dagger}, \quad \log B \lesssim T^{1/\alpha},$$
such that
$$\|f_M'(Z_M) - f\|_2 \lesssim 2^{-T}.$$

The same discussion as Theorem I.2 gives the desired result.

In the following, we construct an FNN $f_m$ and a convolution layer $g_m$ for $m \in [1 : r_{\max}]$. The proof mainly divided into two parts: (i) obtaining $\widehat{X}_{i_k^{(m)}, j_m}$, i.e., the approximation of important features $X_{i_k^{(m)}, j_m}$ $(k = 1, \ldots, |I_m|)$ and (ii) getting $\widehat{j}_m$, i.e., recording which token $j_m$ was selected.

**Picking up the important features $X_{i_k^{(m)}, j_m}$ $(k = 1, \ldots, |I_m|)$** Due to Lemma F.1 and the fact that $j_m \in [0 : V]$ is an index such that $\mu_{t-j}$ $(\widehat{\mu}_{t-j})$ is the largest in $\mu_{t-j}$ $(\widehat{\mu}_{t-j})$ $(j \neq j_1, \ldots, j_{m-1})$, for any $t \in [0 : V]$ with $t \neq t_0$, it holds
$$\left|\frac{\sum_{j=0}^V X_{i,j} \exp(\chi_T \cdot \widehat{\mu}_{t-j}) \cdot (1 - \mathbb{I}_S(j))}{\sum_{j=0}^V \exp(\chi_T \cdot \widehat{\mu}_{t-j}) \cdot (1 - \mathbb{I}_S(j))} - X_{i,j_m}\right| \leq 2V^2 \exp\left(-\chi_T \cdot T^{-\beta/\alpha}\right) \lesssim 2^{-T},$$
where $S = \{j_1, \ldots, j_{m-1}\}$. Now, let us approximate
$$\frac{\sum_{j=0}^V X_{i,j} \exp(\chi_T \cdot \widehat{\mu}_{t_i}) \cdot (1 - \mathbb{I}_S(j))}{\sum_{j=0}^V \exp(\chi_T \cdot \widehat{\mu}_{t-i}) \cdot (1 - \mathbb{I}_S(j))} = \frac{\frac{1}{V} \sum_{j=0}^V X_{i,j} \exp(\chi_T \cdot \widehat{\mu}_{t_i}) \cdot (1 - \mathbb{I}_S(j))}{\frac{1}{V} \sum_{j=0}^V \exp(\chi_T \cdot \widehat{\mu}_{t-i}) \cdot (1 - \mathbb{I}_S(j))}$$

using neural networks. Using Lemma F.4, we can see that there exists a neural network $\phi_* \in \Psi(L, W, S, B)$ with

$$L \lesssim T^{2(1+\beta/\alpha)} \log^2 V, \quad W \lesssim T^{3(1+\beta/\alpha)} \log^3 V, \quad S \lesssim T^{4(1+\beta/\alpha)} \log^4 V, \quad \log B \lesssim T^{1+\beta/\alpha} \log V,$$

such that, for any $x \in [\exp(-\chi_T), \exp(\chi_T)], y \in [0, V], x' > 0, y' > 0$, it holds

$$\left| \phi_*(x', y') - \frac{y}{x} \right| \lesssim 2^{-T} + V^2 2^{T^{1+\beta/\alpha}} (|x - x'| + |y - y'|).$$

Next, Lemma F.5 implies that there exists a neural networks $\phi'_n \in \Psi'(1, B)$ and $\phi_n \in \Psi(L, W, S, B)$ $(n = 1, \ldots, N)$ with

$$N \lesssim T^{1+\beta/\alpha} \log T \log V,$$

$$L \lesssim T^{5(1+\beta/\alpha)} \log T \log^5 V, \quad W \lesssim 1,$$

$$S \lesssim T^{5(1+\beta/\alpha)} \log T \log^5 V, \quad \log B \lesssim T^{3(1+\beta/\alpha)} \log T \log^3 V,$$

such that, for any $t, x, \widehat{x} \in [0, 1]$, it holds

$$\left| \sum_{n=0}^{N} \phi'_n(t) \phi_n(\widehat{x}) - \exp\left( -\frac{V^2(\frac{1}{\alpha} \log T + 2T^{1+\beta/\alpha} + 2\log V) \cdot \sin^2\left(\frac{\pi}{2}(t-x)\right)}{2} \right) \right|$$

$$\lesssim T^{-1/\alpha} 2^{-2T^{1+\beta/\alpha}} / V^2 + T^{1+\beta/\alpha} V^3 |x - \hat{x}|.$$

Since

$$\exp\left( -\frac{V^2(\frac{1}{\alpha} \log T + 2T^{1+\beta/\alpha} + 2\log V) \sin^2\left(\frac{\pi}{2}(t-x)\right)}{2} \right)$$

$$\begin{cases} \leq T^{-1/\alpha} 2^{-2T^{1+\beta/\alpha}} / V^2 & (|t - x| \geq 1/V), \\ = 1 & (t = x), \end{cases}$$

we have

$$\left| \exp\left( -\frac{V^2(\frac{1}{\alpha} \log T + 2T^{1+\beta/\alpha} + 2\log V) \sin^2\left(\frac{\pi}{2}(t-x)\right)}{2} \right) - \mathbb{I}_{\{x\}}(t) \right| \lesssim 2T^{-1/\alpha} 2^{-2T^{1+\beta/\alpha}} / V^2.$$

Therefore, we have

$$\left| \sum_{n=1}^{N} \phi'_n(t) \phi_n(\widehat{x}) - \mathbb{I}_{\{x\}}(t) \right| \lesssim T^{-1/\alpha} 2^{-(1+\beta/\alpha)T} / V^2 + T^{1+\beta/\alpha} V^3 |x - \hat{x}|.$$

Summing up over $x = j_1/V, \ldots, j_{m-1}/V$, we have

$$\left| \sum_{m'=1}^{m-1} \sum_{i=1}^{I} \phi_0^{(j_{m'}, i)}(t) \phi_1^{(j_{m'}, i)}(\hat{j}_{m'}/V) - \mathbb{I}_S(t) \right|$$

$$\lesssim r_{\max}\left( T^{-1/\alpha} 2^{-2T^{1+\beta/\alpha}} / V^2 + T^{1+\beta/\alpha} V^3 \left| \hat{j}_{m'}/V - j_{m'}/V \right| \right)$$

$$\lesssim 2^{-2T^{1+\beta/\alpha}} / V^2.$$

Combining the results above, we have

$$\left| \frac{1}{V} \sum_{j=0}^{V} \phi_\times(\phi_{\exp}(X_{i,j}, \chi_T \widehat{\mu}_{t-j})) \cdot \left( 1 - \sum_{m'=1}^{m-1} \sum_{i=1}^{I} \phi_0^{(j_{m'}, i)}(t) \phi_1^{(j_{m'}, i)}(\hat{j}_{m'}/V) \right) \right.$$

$$\left. - \frac{1}{V} \sum_{j=0}^{V} X_{i,j} \exp(\chi_T \cdot \widehat{\mu}_{t-j}) \cdot (1 - \mathbb{I}_S(j)) \right|$$

$$\lesssim \frac{1}{V} \sum_{j=0}^{V} \left( \left| \phi_\times(\phi_{\exp}(X_{i,j}, \chi_T \cdot \widehat{\mu}_{t-j})) \cdot \left( \sum_{m'=1}^{m-1} \sum_{i=1}^{I} \phi_0^{(j_{m'}, i)}(t) \phi_1^{(j_{m'}, i)}(\hat{j}_{m'}/V) - \mathbb{I}_S(j) \right) \right| \right.$$

$$\left. + |(\phi_\times(\phi_{\exp}(X_{i,j}, \chi_T \widehat{\mu}_{t-j})) - X_{i,j} \exp(\chi_T \cdot \widehat{\mu}_{t-j})) \mathbb{I}_S(j)| \right)$$

$$\lesssim 2^{-2T^{1+\beta/\alpha}} / V^2$$

Similarly, we have

$$\left| \frac{1}{V} \sum_{j=0}^{V} \phi_{\exp}(\chi_T \cdot \widehat{\mu}_{t-j}) \cdot \left( 1 - \sum_{m'=1}^{m-1} \sum_{i=1}^{I} \phi_0^{(j_{m'},i)}(t) \phi_1^{(j_{m'},i)}(\hat{j}_{m'}/V) \right) \right.$$
$$\left. - \frac{1}{V} \sum_{j=0}^{V} \exp(\chi_T \cdot \widehat{\mu}_{t-j}) \cdot (1 - \mathbb{I}_S(j)) \right|$$

$$\lesssim 2^{-2T^{1+\beta/\alpha}}/V^2$$

Using the facts that

$$\exp(-\chi_T) \le \frac{1}{V} \sum_{t=0}^{V} \exp(\chi_T \cdot \widehat{\mu}[t]) \le 1,$$

$$\frac{1}{V} \sum_{t=0}^{V} u[t] \exp(\chi_T \cdot \widehat{\mu}[t]) \le \frac{1}{V} \sum_{t=0}^{V} \exp(\chi_T \cdot \widehat{\mu}[t]),$$

we have

$$\left| \phi_* \left( \frac{1}{V} \sum_{j=0}^{V} \phi_\times (X_{i,j}, \phi_{\exp}(\widehat{\chi}_T \cdot \mu_{t-j})) \cdot \left( 1 - \sum_{m'=1}^{m-1} \sum_{i=1}^{I} \phi_0^{(j_{m'},i)}(t) \phi_1^{(j_{m'},i)}(\hat{j}_{m'}/V) \right), \right. \right.$$
$$\left. \frac{1}{V} \sum_{j=0}^{V} \phi_{\exp}(\widehat{\chi}_T \cdot \mu_{t-j}) \cdot \left( 1 - \sum_{m'=1}^{m-1} \sum_{i=1}^{I} \phi_0^{(j_{m'},i)}(t) \phi_1^{(j_{m'},i)}(\hat{j}_{m'}/V) \right) \right)$$
$$\left. - \frac{\sum_{t=0}^{V} u[t] \exp(\chi_T \cdot \widehat{\mu}[t]) \cdot (1 - \mathbb{I}_S(j))}{\sum_{t=0}^{V} \exp(\chi_T \cdot \widehat{\mu}[t]) \cdot (1 - \mathbb{I}_S(j))} \right|$$

$$\lesssim 2^{-T} + V^2 2^{T^{1+\beta/\alpha}} \cdot 2^{-2T^{1+\beta/\alpha}}/V^2 \lesssim 2^{-T}.$$

Overall, we can see that, there exist neural networks $\phi_O \in \Psi'(L, W, S, B)$ and $\phi_A, \phi_B, \phi_C \in \Psi(L, W, S, B)$ with

$$L \lesssim T^{5+1/\alpha+5\beta/\alpha} \log T \log^5 V, \quad W \lesssim T^{3+1/\alpha+3\beta/\alpha} \log T \log^3 V,$$
$$S \lesssim T^{5+1/\alpha+5\beta/\alpha} \log T \log^5 V, \quad \log B \lesssim T^{3+1/\alpha+3\beta/\alpha} \log T \log^3 V,$$

such that

$$\max_{i \in \left\{ i_1^{(m)}, \dots, i_{|I_m|}^{(m)} \right\}} \left| \underbrace{\phi_C \left( \sum_{j=0}^{V} \phi_O(j/V) \phi_A(Z_{m-1}) \phi_B(Z_{m-1}[-j]) \right)}_{=: \widehat{X}_{i,j_m}} - X_{j_m,i} \right| \lesssim 2^{-T}.$$

**Recording which token was picked up** Similar discussion as above shows that there exist neural networks $\phi'_O \in \Psi'(L, W, S, B)$ and $\phi'_A, \phi'_B, \phi'_C \in \Psi(L, W, S, B)$ with

$$L \lesssim T^{5+1/\alpha+5\beta/\alpha} \log T \log^5 V, \quad W \lesssim T^{3+1/\alpha+3\beta/\alpha} \log T \log^3 V,$$
$$S \lesssim T^{5+1/\alpha+5\beta/\alpha} \log T \log^5 V, \quad \log B \lesssim T^{3+1/\alpha+3\beta/\alpha} \log T \log^3 V,$$

such that

$$\left| \phi'_C \left( \sum_{j=0}^{V} \phi'_O(j/V) \phi'_A(Z_{m-1}) \phi'_B(Z_{m-1}[-j]) \right) - \sin\left( \frac{\pi}{4} \frac{j_m}{V} \right) \right| \lesssim 2^{-T}.$$

Lemma F.3 shows that there exists a neural network $\phi_{\arcsin} \in \Psi(L, W, S, B)$ with

$$L \lesssim T^{2(1+\beta/\alpha)} \log^2 V, \quad W \lesssim 1, \quad S \lesssim T^{2(1+\beta/\alpha)} \log^2 V, \quad \log B \lesssim T^{1+\beta/\alpha} \log V,$$

for any $x \in [0, \pi/4]$, it holds

$$|\phi_{\arcsin}(x) - \arcsin(x)| \lesssim 2^{-3T^{1+\beta/\alpha}}/V^5.$$

Using this network, we can obtain $\widehat{j}_m/V$ such that $\left|\widehat{j}_m/V - j_m/V\right| \lesssim 2^{-3T^{1+\beta/\alpha}}/V^5$.

**Finishing the proof**  We can easily see that, constructing the weight matrix in the convolution layers appropriately, we can obtain $Z_m$ from $Z_{m-1}$ using the neural networks constructed above. This completes the proof.  $\square$

## J  PROOF OF THEOREM 4.6

### J.1  PREPARATION

In this subsection, we prove the following theorem.

**Theorem J.1.** *Let $\hat{F} \in \mathcal{S}(M, U, D, L, W, S, B)$ be an ERM estimator which minimizes the emprical cost. Then, for any $\delta \in (0, 1)$, it holds that*

$$R_{l,r}(\hat{F}, F^\circ) \lesssim \inf_{F \in \mathcal{S}} \frac{1}{r - l + 1} \sum_{i=l}^{r} \|F_i - F_i^\circ\|_{2,P_X}^2 + \frac{1}{n} \cdot M^2 L(S + D) \log\left(\frac{DULWB}{\delta}\right) + \delta.$$

To prove the theorem, we use the following proposition.

**Proposition J.2** (Theorem 5.2 in Takakura & Suzuki (2023))**.** *For a given class $\mathcal{F}$ of functions from $[0, 1]^{d \times \infty}$ to $\mathbb{R}^\infty$, let $\hat{F} \in \mathcal{F}$ be an ERM estimator which minimizes the empirical cost. Suppose that there exists a constant $R > 0$ such that $\|F^\circ\|_\infty \leq R$, $\|F\|_\infty \leq R$ for any $F \in \mathcal{F}$, and $\mathcal{N}(\mathcal{F}, \delta, \|\cdot\|_\infty) \geq 3$. Then, for any $0 < \delta < 1$, it holds that*

$$R_{l,r}(\hat{F}, F^\circ) \lesssim \inf_{F \in \mathcal{F}} \frac{1}{r - l + 1} \sum_{i=l}^{r} \|F_i - F_i^\circ\|_{2,P_X}^2 + (R^2 + \sigma^2) \frac{\log \mathcal{N}(\mathcal{F}, \delta, \|\cdot\|_\infty)}{n} + (R + \sigma)\delta,$$

*where $\mathcal{N}(\mathcal{F}, \delta, \|\cdot\|)$ is the $\delta$-covering number of the space $\mathcal{F}$ associated with the norm $\|\cdot\|$, defined by*

$$\mathcal{N}(\mathcal{F}, \delta, \|\cdot\|) := \inf \{m \in \mathbb{N} \mid \exists F_1, \dots, F_m \in \mathcal{F}, \forall F \in \mathcal{F}, \exists i \in [m] \text{ s.t. } \|F - F_i\| \leq \delta\}.$$

Thanks to this proposition, the problem to obtain the upper bound of the excess risk of the estimator $\hat{F}$ is reduced to the problem to evaluate the covering number of the function class $\mathcal{S}$. The covering number of the function class $\mathcal{S}$ can be evaluated as follows.

**Theorem J.3** (Covering number of SSMs)**.** *The covering number of the function class $\mathcal{S}(M, U, D, L, W, S, B)$ can be bounded as*

$$\log \mathcal{N}(\mathcal{S}(M, U, D, L, W, S, B), \delta, \|\cdot\|_\infty) \lesssim M^2 L(S + D^2) \log\left(\frac{DULWB}{\delta}\right).$$

This theorem implies that the upper bound of the covering number of the function class $\mathcal{S}$ polynomially increases with respect to the embedding dimensions $D$, the number of layers $M, L$ and the sparsity $S$ of the parameters. This result is similar to the result by Takakura & Suzuki (2023) on the covering number of Transformers.

A large difference of the covering number between the SSMs and Transformers is the dependence on the window size $U$; the covering number of the SSMs depends on $U$ logarithmically, while that of the Transformers does not depend on $U$. This is because SSMs sum up the tokens in the convolution without normalization. Whereas it is preferred that the covering number does not depend on $U$, the logarithmic dependence on $U$ is not a serious problem for the estimation ability, as we will see later.

In the following, we prove Theorem J.3. First of all, we introduce the lemma below, which is useful to evaluate the covering number.

**Lemma J.4.** *Let $\{f_\theta\}_{\theta \in \Theta}$ be a parametrized function class from $[0, 1]^{d \times \infty}$ to $\mathbb{R}^\infty$. Suppose that the parameter space $\Theta$ satisfies $\Theta \subseteq [-B, B]^D$ for some $B > 0, D > 0$. Additionally, suppose that*
$$|\{\theta \mid \theta \neq 0, \theta \in \Theta\}| \leq S.$$
*Moreover, assume that there exists a constant $r > 0$ such that*
$$\left\|f_\theta - f_{\tilde{\theta}}\right\|_\infty \leq r\|\theta - \tilde{\theta}\|_\infty \quad \text{for any } \theta, \tilde{\theta} \in \Theta.$$

*Then, it holds*

$$\log \mathcal{N}(\mathcal{F}, \delta, \|\cdot\|_\infty) \le S \log\left(\frac{rBD}{\delta}\right).$$

The following lemma is drawn from Takakura & Suzuki (2023), which evaluates the norm of the output of FNN, the lipschitz constant with respect to the input, and the lipschitz constant with respect to the parameters.

**Lemma J.5** (Lemma E.3 in Suzuki (2018)). *Suppose that two FNNs $f, \widetilde{f}$ with $L$ layers and $W$ hidden units is given by*

$$f(x) := (A_L \sigma(\cdot) + b_L) \circ \cdots \circ (A_1 \sigma(x) + b_1),$$
$$\widetilde{f}(x) := (\widetilde{A}_L \sigma(\cdot) + \widetilde{b}_L) \circ \cdots \circ (\widetilde{A}_1 \sigma(x) + \widetilde{b}_1),$$

*where $\sigma$ is the ReLU activation function. Assume that for any $l = 1, \ldots, L$, it holds*

$$\|A_l\|_\infty \le B, \quad \left\|\widetilde{A}_l\right\|_\infty \le B, \quad \|b_l\|_\infty \le B, \quad \left\|\widetilde{b}_l\right\|_\infty \le B.$$

*Additionally, let $r \ge 1$ be a constant.*

1. *For any $x \in \mathbb{R}^{D \times \infty}$ with $\|x\|_\infty \le r$, it holds*
$$\|f(x)\|_\infty \le (2BW)^L r.$$

2. *For any $X, X' \in \mathbb{R}^{D \times \infty}$, it holds*
$$\|f(x) - f(x')\|_\infty \le (BW)^L \|X - X'\|_\infty.$$

3. *Assume that, for any $l = 1, \ldots, L$, it holds*
$$\left\|A_l - \widetilde{A}_l\right\|_\infty \le \delta, \quad \left\|b_l - \widetilde{b}_l\right\|_\infty \le \delta.$$
   *Then, for any $x \in \mathbb{R}^D$ with $\|x\|_\infty \le r$, it holds*
$$\left\|f(x) - \widetilde{f}(x)\right\|_\infty \le 2(2BW)^L r \cdot \delta.$$

We also evaluate them for the convolution layers.

**Lemma J.6.** *Suppose that two convolution layers $g, \widetilde{g}$ with window size $U$ and embedding dimention $D$ is given by*

$$g(X) := \beta(X) * (W_V X),$$
$$\widetilde{g}(X) := \widetilde{\beta}(X) * (\widetilde{W_V} X).$$

*Let $r \ge 1$ be a constant. Assume that it holds*

$$\|W_V\|_\infty \le B, \quad \left\|\widetilde{W}_V\right\|_\infty \le B,$$

*and, for any $h = 0, \ldots, H$ and $X \in \mathbb{R}^{d \times \infty}$ with $\|X\|_\infty \le r$, it holds*

$$\|\beta(X)\|_1 \le c, \quad \left\|\widetilde{\beta}(X)\right\|_1 \le c,$$

*for some $B \ge 1, c \ge 1$. Then, the following statements hold.*

1. *For any $X \in \mathbb{R}^{D \times \infty}$ with $\|X\|_\infty \le r$, it holds*
$$\|g(X)\|_\infty \le BDrc.$$

2. *Suppose that $X, X' \in \mathbb{R}^{D \times \infty}$ satisfies $\|X\|_\infty \le r, \|X'\|_\infty \le r$ and*
$$\|\beta(X) - \beta(X')\|_1 \le \kappa \|X - X'\|_\infty$$
   *for some $\kappa \ge 0$[1]. Then, it holds*
$$\|g(X) - g(X')\|_\infty \le \left(B^2 rc + Br \cdot \kappa\right) \|X - X'\|_\infty.$$

3. *Assume that, for any $h = 0, \ldots, H$, it holds*
$$\left\|W_V - \widetilde{W}_V\right\|_\infty \le \delta, \quad \left\|\beta(X) - \widetilde{\beta}(X)\right\|_1 \le \iota\delta.$$

---

[1] If the filter is data-dependent, then $\kappa = 0$.

*for $\iota > 0$. Then, it holds*

$$\|g(X) - \widetilde{g}(X)\|_\infty \le \left(Br^2 c + (Br)^2 \cdot \iota\right) \cdot \delta.$$

*Proof.* We use frequently the following three inequalities:

$$\|WX\|_\infty \le \|W\|_1 \|X\|_\infty \le D \cdot \|W\|_\infty \|X\|_\infty,$$
$$\|\beta * X\|_\infty \le \|\beta\|_1 \|X\|_\infty,$$

where $W \in \mathbb{R}^{D \times D}, X \in \mathbb{R}^{D \times \infty}, Y \in \mathbb{R}^{D \times \infty}, \beta \in \mathbb{R}^{D \times U}$.

**Proof of 1**   We have

$$
\begin{aligned}
\|g(X)\|_\infty &= \|(\beta(X) * (W_V X))\|_\infty \\
&\le \|\beta(X) * (W_V X)\|_\infty \\
&\le \|W_V X\|_\infty \cdot \|\beta(X)\|_1 \\
&\le BDr \cdot c \le BDrc.
\end{aligned}
$$

**Proof of 2**   We have

$$
\begin{aligned}
\|g(X) - g(X')\|_\infty &= \|\beta(X) * (W_V X) - \beta(X') * (W_V X')\|_\infty \\
&\le \|\beta(X) * (W_V X) - \beta(X') * (W_V X)\|_\infty \\
&\quad + \|(\beta(X') * (W_V X)) - (\beta(X') * (W_V X'))\|_\infty \\
&\le \|((\beta(X) - \beta(X')) * (W_V X))\|_\infty + \|(\beta(X') * (W_V(X - X')))\|_\infty \\
&\le \|\beta(X) - \beta(X')\|_1 \cdot \|W_V X\|_\infty + \|\beta(X')\|_1 \cdot \|W_V(X - X')\|_\infty \\
&\le Br \cdot \kappa \|X - X'\|_\infty \cdot Br + Br \cdot c \cdot B \|X - X'\|_\infty \\
&= \left(B^2 rc + Br \cdot \kappa\right) \|X - X'\|_\infty.
\end{aligned}
$$

**Proof of 3**   We have

$$
\begin{aligned}
\|g(X) - \widetilde{g}(X)\|_\infty &= \left\| \beta(X) * (W_V X) - \widetilde{\beta}(X) * \left(\widetilde{W}_V X\right) \right\|_\infty \\
&\le \left\| \beta(X) * (W_V X) - \widetilde{\beta}(X) * (W_V X) \right\|_\infty \\
&\quad + \left\| \widetilde{\beta}(X) * (W_V X) - \widetilde{\beta}(X) * \left(\widetilde{W}_V X\right) \right\|_\infty \\
&\le \left\| \left(\beta(X) - \widetilde{\beta}(X)\right) * (W_V X) \right\|_\infty + \left\| \widetilde{\beta}(X) * \left(\left(W_V - \widetilde{W}_V\right) X\right) \right\| \\
&\le \left\| \beta(X) - \widetilde{\beta}(X) \right\|_1 \cdot \|W_V X\|_\infty + \left\| \widetilde{\beta}(X) \right\|_1 \cdot \left\| \left(W_V - \widetilde{W}_V\right) X \right\|_\infty \\
&\le Br \cdot \iota\delta \cdot Br + Br \cdot c \cdot \delta r \\
&= \left(Br^2 c + (Br)^2 \cdot \iota\right)\delta.
\end{aligned}
$$

$\square$

Subsequently, we evaluate the lipschitz constant of the composition of the layers with respect to the input and the parameters.

**Lemma J.7.** *Let $(f_1, \widetilde{f}_1), \dots, (f_M, \widetilde{f}_M)$ be pairs of two FNNs which satisfy the same condition of the pair $(f, \widetilde{f})$ in Lemma J.5. Additionally, let $(g_1, \widetilde{g}_1), \dots, (g_M, \widetilde{g}_M)$ be convolution layers which satisfy the same condition of the pair $(g, \widetilde{g})$ in Lemma J.6. Suppose $R > 0$ be a constant, and $F, \widetilde{F} \colon [0, 1]^{d \times \infty} \to \mathbb{R}^\infty$ are two functions defined by*

$$F := \mathrm{clip}_R \circ f_M \circ g_M \circ \cdots \circ \mathrm{clip}_R \circ f_1 \circ g_1,$$

$$\widetilde{F} := \mathrm{clip}_R \circ \widetilde{f}_M \circ \widetilde{g}_M \circ \cdots \circ \mathrm{clip}_R \circ \widetilde{f}_1 \circ \widetilde{g}_1.$$

*Moreover, assume that $B \ge 1, c \ge 1, r \ge 1$. Then, it holds*

$$\left\| F(X) - \widetilde{F}(X) \right\|_\infty \le 2^{M+1} (2BW)^{ML} (BDRc)^{2M} (1 + \kappa)^M (1 + \iota) \cdot \delta.$$

*Proof.* For $m = 1, \ldots, M$, we define

$$F_m := \text{clip}_R \circ f_m \circ g_m \circ \cdots \circ \text{clip}_R \circ f_1 \circ g_1, \quad \widetilde{F}_m := \text{clip}_R \circ \widetilde{f}_m \circ \widetilde{g}_m \circ \cdots \circ \text{clip}_R \circ \widetilde{f}_1 \circ \widetilde{g}_1,$$

and $F_0 := \text{id}, \widetilde{F}_0 := \text{id}$. Then, it holds

$$F_m = \text{clip}_R \circ f_m \circ g_m \circ F_{m-1}, \quad \widetilde{F}_m = \text{clip}_R \circ \widetilde{f}_m \circ \widetilde{g}_m \circ \widetilde{F}_{m-1}$$

for $m = 1, \ldots, M$. Note that $\|F_m\|_\infty \leq R$ and $\left\|\widetilde{F}_m\right\|_\infty \leq R$ for any $m = 1, \ldots, M$ due to the clipping.

For any $X \in \mathbb{R}^{d \times \infty}$ with $\|X\|_\infty \leq r$ and $m = 1, \ldots, M$, we have

$$\left\|F_m(X) - \widetilde{F}_m(X)\right\|_\infty = \left\|\text{clip}_R \circ f_m \circ g_m \circ F_{m-1}(X) - \text{clip}_R \circ \widetilde{f}_m \circ \widetilde{g}_m \circ \widetilde{F}_{m-1}(X)\right\|_\infty$$

$$= \left\|f_m \circ g_m \circ F_{m-1}(X) - \widetilde{f}_m \circ \widetilde{g}_m \circ \widetilde{F}_{m-1}(X)\right\|_\infty$$

$$(\because \text{clip}_R \text{ is 1-lipschitz continuous.})$$

$$\leq \left\|f_m \circ g_m \circ F_{m-1}(X) - \widetilde{f}_m \circ g_m \circ F_{m-1}(X)\right\|_\infty$$

$$+ \left\|\widetilde{f}_m \circ g_m \circ F_{m-1}(X) - \widetilde{f}_m \circ \widetilde{g}_m \circ F_{m-1}(X)\right\|_\infty$$

$$+ \left\|\widetilde{f}_m \circ \widetilde{g}_m \circ F_{m-1}(X) - \widetilde{f}_m \circ \widetilde{g}_m \circ \widetilde{F}_{m-1}(X)\right\|_\infty.$$

For the first term, since $\|g_m \circ F_{m-1}(X)\| \leq (BDRc)^2$ due to the first argument of Lemma J.6, using the third argument of Lemma J.5, we have

$$\left\|f_m \circ g_m \circ F_{m-1}(X) - \widetilde{f}_m \circ g_m \circ F_{m-1}(X)\right\|_\infty \leq 2(2BW)^L (BDRc)^2 \cdot \delta.$$

For the second term, the second argument of Lemma J.5 and the third argument of Lemma J.6 yield

$$\left\|\widetilde{f}_m \circ g_m \circ F_{m-1}(X) - \widetilde{f}_m \circ \widetilde{g}_m \circ F_{m-1}(X)\right\|_\infty \leq (BW)^L \|g_m \circ F_{m-1}(X) - \widetilde{g}_m \circ F_{m-1}(X)\|_\infty$$

$$\leq (BW)^L \cdot \left(2BR^2 c + (BR)^2 \cdot \iota\right) \cdot \delta.$$

For the thrid term, the third argument of Lemma J.5 and the third argument of Lemma J.6 imply

$$\left\|\widetilde{f}_m \circ \widetilde{g}_m \circ F_{m-1}(X) - \widetilde{f}_m \circ \widetilde{g}_m \circ \widetilde{F}_{m-1}(X)\right\|_\infty$$

$$\leq (BW)^L \left\|\widetilde{g}_m \circ F_{m-1}(X) - \widetilde{g}_m \circ \widetilde{F}_{m-1}(X)\right\|_\infty$$

$$\leq (BW)^L \cdot \left(2B^2 Rc + BR \cdot \kappa\right) \cdot \left\|F_{m-1}(X) - \widetilde{F}_{m-1}(X)\right\|_\infty.$$

Let $\lambda_1, \lambda_2$ be the constants defined by

$$\lambda_1 := \left(2(2BW)^L (BDRc)^2 + (BW)^L \cdot \left(2BR^2 c + (BR)^2 \cdot \iota\right)\right) \cdot \delta$$

$$\lambda_2 := (BW)^L \cdot \left(2B^2 Rc + BR \cdot \kappa\right).$$

Then, we have

$$\left\|F_m(X) - \widetilde{F}_m(X)\right\|_\infty \leq \lambda_1 + \lambda_2 \cdot \left\|F_{m-1}(X) - \widetilde{F}_{m-1}(X)\right\|_\infty.$$

This implies

$$\left\|F_m(X) - \widetilde{F}_m(X)\right\|_\infty + \frac{\lambda_1}{\lambda_2 - 1} \leq \lambda_2 \cdot \left(\left\|F_{m-1}(X) - \widetilde{F}_{m-1}(X)\right\|_\infty + \frac{\lambda_1}{\lambda_2 - 1}\right).$$

Thus, by induction, we have

$$\left\|F_m(X) - \widetilde{F}_m(X)\right\|_\infty + \frac{\lambda_1}{\lambda_2 - 1} \leq \lambda_2^m \cdot \left(\left\|F_0(X) - \widetilde{F}_0(X)\right\|_\infty + \frac{\lambda_1}{\lambda_2 - 1}\right) = \frac{\lambda_2^m \cdot \lambda_1}{\lambda_2 - 1}.$$

Since $\lambda_2 > 1$, it holds

$$\left\|F_m(X) - \widetilde{F}_m(X)\right\|_\infty \leq \lambda_1 \cdot \frac{\lambda_2^m - 1}{\lambda_2 - 1} = \lambda_1 \cdot \left(1 + \lambda_2 + \cdot + \lambda_2^{m-1}\right) \leq m \lambda_1 \lambda_2^{m-1}.$$

Now, using

$$\lambda_1 \leq 3(2BW)^L (BDRc)^2 (1 + \iota) \cdot \delta, \quad \lambda_2 \leq 2(2BW)^L (BDRc)^2 (1 + \kappa),$$

we have
$$\left\|F(X) - \widetilde{F}(X)\right\|_\infty \le M\lambda_1\lambda_2^{M-1} \le 2^{M+1}(2BW)^{ML}(BDRc)^{2M}(1+\kappa)^M(1+\iota)\cdot\delta,$$
which completes the proof. $\qquad\square$

Finally, we prove Theorem J.3.

*Proof of Theorem J.3.* In the model we consider, it holds
$$\kappa = 0,$$
$$\iota \le 2U\cdot(2BW')^{L'},$$
$$c \le U(2BW')^{L'}.$$
Therefore, we have
$$\left\|F(X) - \widetilde{F}(X)\right\|_\infty \le 2^{M+1}(2BW)^{ML}\big(BDRU(2BW')^{L'}\big)^{2M}\cdot\left(2\cdot2U(2BW')^{L'}\right)\cdot\delta$$
$$= 2^{M+3}(2BW)^{ML}(2BW')^{(2M+1)L'}(BDRU)^{2M+1}\cdot\delta.$$
The number of parameters in a FNN is $2W^2L$. Additionally, the number of parameters in a convolution layer is $2D^2$. Moreover, the number of nonzero parameters in whole network is bounded by $M(S+2D^2)$. Therefore, the covering number can be evaluated as
$$\log\mathcal{N}(\mathcal{S}(M,U,D,L,W,S,B),\delta,\|\cdot\|_\infty)$$
$$\le M(S+2D^2)$$
$$+ \log\left(\frac{M(2W^2L+D^2)\cdot B\cdot2^{M+3}(2BW)^{ML}(2BW')^{(2M+1)L'}(BDRU)^{2M+1}}{\delta}\right)$$
$$\lesssim M^2L(S+D^2)\log\left(\frac{DULWB}{\delta}\right),$$
which completes the proof. $\qquad\square$

## J.2 PROOF OF THEOREM 4.6

Theorem 4.5 implies that, for any $T > 0$, there exists an SSM $F \in \mathcal{S}(M,U,D,L,W,S,B)$ with
$$M = T^{1/\alpha}, \quad U = V, \quad D \sim T^{c_{\alpha,\beta}}\log^2 V,$$
$$L \sim T^{c_{\alpha,\beta}}\log^5 V, \quad W \sim 2^{T/a^\dagger}T^{c_{\alpha,\beta}}\log^3 V,$$
$$S \sim 2^{T/a^\dagger}T^{c_{\alpha,\beta}}\log^5 V, \quad \log B \sim T^{c_{\alpha,\beta}}\log^3 V,$$
such that $\|F - F^\circ\|_{2,P_X} \lesssim 2^{-T}$. Therefore, it holds
$$\|F - F^\circ\|_{2,P_X}^2 \le 2^{-2T}.$$

Next, Theorem J.3 shows that it holds
$$\log N(\mathcal{S},\delta,\|\cdot\|_\infty) \lesssim 2^{T/a^\dagger}T^{2/\alpha+4c_{\alpha,\beta}}(\log V)^{13}\log\frac{1}{\delta}.$$

Using Theorem J.1, we can show that
$$R(\hat{F},F^\circ) \lesssim 2^{-2T} + \frac{2^{T/a^\dagger}T^{2/\alpha+4c_{\alpha,\beta}}(\log V)^{13}\log\frac{1}{\delta}}{n} + \delta.$$
By setting $T = \frac{a^\dagger}{2a^\dagger+1}\log n$ and $\delta = 1/n$, we have
$$R(\hat{F},F^\circ) \lesssim n^{-\frac{2a^\dagger}{2a^\dagger+1}}(\log n)^{1+2/\alpha+4c_{\alpha,\beta}}\log^{13}V.$$

## K ADDITIONAL DETAILS ON THE EXPERIMENTS

All the code was implemented in Python 3.10.14 with Pytorch 1.13.1 and CUDA ver 11.7. The experiments were conducted on Ubuntu 20.04.5 with A100 PCIe 40GB.

Genomic Benchmark dataset (Grešová et al., 2023) is given with the Apache License Version 2.0 and can be accessed from `https://github.com/ML-Bioinfo-CEITEC/genomic_benchmarks`. The pretrained model of Hyena is given with the Apache License Version 2.0 and can be accessed from `https://github.com/HazyResearch/safari?tab=readme-ov-file`.

For the training and evaluation of models, we utilized the code provided at `https://colab.research.google.com/drive/1wyVEQd4R3HYLTUOXEEQmp_I8aNC_aLhL`.

We used the dataset `human_enhancers_cohn` of Genomic Benchmark dataset. As for the pretrained model of Hyena, we used `hyenadna-tiny-1k-seqlen`. The model was fine-tuned for 100 epochs. Then, we sampled 20 different test sequences whose correct probability is larger or equal to 0.95. For each sequence, we repeatedly mask the tokens that maximize the correct probability. The error bar is calculated by the standard deviation of these 20 samples. The source code for the experiment is `downstream_finetune.py` and `downstream_mask.py`, which can be found in the supplemental material. Finetuning needs around one hour, and masking needs around 90 minutes.

