# OpenReview forum: "State Space Models are Provably Comparable to Transformers in Dynamic Token Selection"
_ICLR.cc/2025/Conference — ICLR 2025 Poster_

### Official Review · Reviewer_Byu5 · 2024-10-27

**Soundness:** 3
**Presentation:** 2
**Contribution:** 2
**Rating:** 6
**Confidence:** 4

**Summary:**

This work presents a theoretical investigation into the capabilities of State Space Models (SSMs) for dynamic token selection. While previous research has established that standalone SSM layers exhibit inferior performance compared to Transformers, the authors demonstrate that SSMs combined with Fully-connected Neural Networks (FNNs) achieve comparable capabilities in dynamic token selection. The theoretical contributions are substantiated through rigorous mathematical analyses across three fundamental scenarios: input copying, associative recall, and non-parametric regression.

**Strengths:**

1. **The paper is well-motivated:** while prior theoretical works focus on standalone SSM layers and their limitations compared to Transformers, the authors insightfully observe that practical architectures combine SSMs with FNN layers and raise the important question in [page 2, line 77].
2. **Theoretical rigor:** The paper presents a rigorous theoretical analysis to demonstrate their claims, grounded in well-established mathematical foundations with detailed proofs.

**Weaknesses:**

1. **Lack of empirical validation:** The paper lacks sufficient experimental evidence for its theoretical claims. As the paper did not discuss whether SSMs can be optimized efficiently (limitation section), there are serious concerns whether SSMs can actually demonstrate comparable dynamic token selection capability to Transformers in practical scenarios, or at least in synthetic tasks which they studied theoretically (input copying, associative recall).

2. **Imbalanced analysis of scenarios:** Among the three scenarios (input copying, associative recall, and non-parametric regression) studied in this paper, the paper allocates excessive attention to non-parametric regression, which is less central to the paper's primary contribution - SSMs' dynamic token selection capabilities. A more concise treatment of non-parametric regression would have allowed for deeper analysis of the more fundamental scenarios.

**Questions:**

1. This paper theoretically compares input copying capabilities against [1] and associative recall capabilities against [2]. However, unlike [1] and [2] which provide comprehensive experimental validations, this paper lacks empirical results. To substantiate the theoretical findings, the authors should provide **experimental results** comparing the performance of (i) SSM, (ii) SSM+FNN, and (iii) Transformer on both input copying and associative recall tasks.
2. **The choice of non-parametric regression** as a case study for demonstrating the dynamic token selection capabilities of SSMs **requires further justification.** This analysis appears loosely related to the paper's main contribution and occupies a disproportionate amount of space. While valuable, it would be more suitable as a separate study. The authors should either better justify this choice or focus on more relevant scenarios that directly showcase dynamic token selection capabilities of SSMs.
    * Instead of non-parametric regression, the paper would benefit from extending its theoretical analysis to the synthetic tasks introduced in Mamba [3] - a seminal work that has become one of the most representative studies in the SSM literature. Specifically, its **selective copying** and **induction heads** tasks align with input copying and associative recall capabilities, respectively. Including these analyses would strengthen the connection between the paper's theoretical framework and contemporary SSM research.
3. It seems that the problem setting of this paper (Section 2, Appendix A) for theoretical analysis **overlooks the structural constraints of the state matrix A** (e.g., diagonal or other structured forms) that enable efficient SSM training, which was one of the key points for the practical success of SSMs. While prior work [1] analyzing generalized SSMs without structural constraints was suitable for claiming SSM limitations, this paper's goal of establishing SSM-Transformer comparability requires considering these practical aspects. Consequently, without incorporating the structural constraints, the theoretical claims may not translate to implementable architectures in practice. Please correct me if I'm wrong.
4. The theoretical foundation appears inconsistent: while the paper references the theoretical results from [1] which analyzes a single SSM layer [page 2, line 84], the current analysis assumes two SSM layers with additional FNN layers [page 5, line 222] (please correct me if I'm wrong). As the superior performance of (multi-layer) SSM+FNN over (single-layer) standalone SSM is self-evident (adding layers naturally increases modeling capacity), the current theoretical analysis alone offers limited practical insights. The paper would benefit from **extended theoretical analyses in terms of architectural design choices,** such as comparing SSM, SSM+FNN, and Transformer architectures under equal parameter budgets. Furthermore, an examination of optimal parameter allocation between SSM and FNN components under fixed budgets would be valuable.




[1] Repeat after me: Transformers are better than state space models at copying
[2] Laughing hyena distillery: Extracting compact recurrences from convolutions
[3] Mamba: Linear-Time Sequence Modeling with Selective State Spaces

---

> ### Author Response · Authors · 2024-11-21
>
> Thank you for your detailed and valuable feedbacks. We respond to specific concerns and questions below.
>
> > Lack of empirical validation: The paper lacks sufficient experimental evidence for its theoretical claims. As the paper did not discuss whether SSMs can be optimized efficiently (limitation section), there are serious concerns whether SSMs can actually demonstrate comparable dynamic token selection capability to Transformers in practical scenarios, or at least in synthetic tasks which they studied theoretically (input copying, associative recall).
>
> > This paper theoretically compares input copying capabilities against [1] and associative recall capabilities against [2]. However, unlike [1] and [2] which provide comprehensive experimental validations, this paper lacks empirical results. To substantiate the theoretical findings, the authors should provide **experimental results** comparing the performance of (i) SSM, (ii) SSM+FNN, and (iii) Transformer on both input copying and associative recall tasks.
>
> Thank you for your constructive suggestion. To empirically validate that SSMs combined with FNNs actually exhibit dynamic token selection ability, we conducted experiments on input copying and associative recall. The results can be found in https://drive.google.com/file/d/1m9nL_6XnxESmOE-DxIbzlfQZuxjcrw5w/view?usp=sharing. Additionally, we described the details in Appendix C.
>
> From the results, we can see that multi-layer SSMs perform better than single-layer SSMs, particularly when the dimension of the hidden states are small and the FNN layers are sufficiently expressive. This empirical observation match our theoretical findings. Indeed, our theory demonstrates that SSMs combined with preceding and following FNN layers have dynamic token selection ability, which means that it is essential to alternately apply SSMs and FNNs to solve these tasks. Additionally, in the input copying task, we compare SSMs and Transformers, and find that multi-layer SSMs combined with FNN layers perform similarly to Transformers.
>
> > Imbalanced analysis of scenarios: Among the three scenarios (input copying, associative recall, and non-parametric regression) studied in this paper, the paper allocates excessive attention to non-parametric regression, which is less central to the paper's primary contribution - SSMs' dynamic token selection capabilities. A more concise treatment of non-parametric regression would have allowed for deeper analysis of the more fundamental scenarios.
>
> > The choice of non-parametric regression as a case study for demonstrating the dynamic token selection capabilities of SSMs requires further justification. This analysis appears loosely related to the paper's main contribution and occupies a disproportionate amount of space. While valuable, it would be more suitable as a separate study.
>
> We appreciate your insightful comment. In our paper, non-parametric regression is one of the key contributions, alongside the two synthetic tasks (input copying and associative recall). First, this problem setting, like the two synthetic tasks, supports our claim that SSMs possess dynamic token selection abilities comparable to Transformers. Indeed, the functions we consider (piecewise $\gamma$-smooth functions) exhibit a smoothness structure that varies depending on the input. If the function is non-smooth for a certain token, the token is important for that function. This indicates that, for piecewise $\gamma$-smooth functions, the position of essential tokens change depending on the input, which means that the models are required to adaptively select which token to extract based on the input.
>
> Moreover, the analysis of non-parametric regression provides two insights that differ from those gained through the analysis of synthetic tasks:
>
> 1. **SSMs’ capabilities on more general tasks:** Theoretical analyses of specific synthetic tasks provide an intuitive understanding of the model's abilities but are somewhat limited in scope. Through the analysis of non-parametric regression, we can gain a broader understanding of SSMs' ability to estimate much general functions belonging to a certain class.
> 2. **Learnability with finite data:** In the analysis of synthetic tasks, we discussed the number of parameters required to solve the tasks, but it is not trivial whether such a model can be trained effectively on a finite dataset. In our statistical analysis of non-parametric regression, we evaluate the number of data for SSMs to estimate the function. As a result, we demonstrate that SSMs combined with FNNs can estimate functions with the same amount of data as Transformers.
>
> Since there was lack of explanation of the motivation to consider piecewise $\gamma$-smooth function class, in Appendix E, we add further intuitive description with figures. Moreover, to connect our theory and real-world applications, we provided some examples of practical tasks that can be seen as examples of piecewise $\gamma$-smooth functions.

---

> ### Author Response · Authors · 2024-11-21
>
> > Instead of non-parametric regression, the paper would benefit from extending its theoretical analysis to the synthetic tasks introduced in Mamba [3] - a seminal work that has become one of the most representative studies in the SSM literature. Specifically, its **selective copying** and **induction heads** tasks align with input copying and associative recall capabilities, respectively. Including these analyses would strengthen the connection between the paper's theoretical framework and contemporary SSM research.
>
> Thank you for your constructive suggestion. As for selective copying, using similar techniques, we can show that $O(\mathrm{poly} \log(L)) $parameters are sufficient to solve it by SSMs combined with FNNs, where $L$ is the sequence length. We add the details in Appendix D in the revised version, and formalize it in Theorem D.2. Regarding induction heads, we can also prove almost the same result as associative recall. We also add this point in the revised version, and formally state it in Theorem D.1.
>
> > It seems that the problem setting of this paper (Section 2, Appendix A) for theoretical analysis **overlooks the structural constraints of the state matrix A** (e.g., diagonal or other structured forms) that enable efficient SSM training, which was one of the key points for the practical success of SSMs. While prior work [1] analyzing generalized SSMs without structural constraints was suitable for claiming SSM limitations, this paper's goal of establishing SSM-Transformer comparability requires considering these practical aspects. Consequently, without incorporating the structural constraints, the theoretical claims may not translate to implementable architectures in practice.
>
> Thank you for your important feedback. In this paper, we did not assume any specific structure for the matrix $A$. However, in all the theoretical results of our study, $A$ can be assumed to be a block diagonal matrix expressed as $A = \mathrm{diag}(D_1, \dots, D_k)$, where $D_1, \dots, D_k$ are $2 \times 2 $ matrices (please refer to Appendix A for details). Investigating how the structure of $A$ can be further restricted is left as future work. We added this point in the limitation paragraph in the revised version.
>
> > The theoretical foundation appears inconsistent: while the paper references the theoretical results from [1] which analyzes a single SSM layer [page 2, line 84], the current analysis assumes two SSM layers with additional FNN layers [page 5, line 222] (please correct me if I'm wrong). As the superior performance of (multi-layer) SSM+FNN over (single-layer) standalone SSM is self-evident (adding layers naturally increases modeling capacity), the current theoretical analysis alone offers limited practical insights. The paper would benefit from extended theoretical analyses in terms of architectural design choices, such as comparing SSM, SSM+FNN, and Transformer architectures under equal parameter budgets.
>
> Thank you for your insightful question to make our contribution clear. As you pointed out, it is almost obvious that multi-layer SSMs + FNNs are more expressive than a single SSM layer. We also remarked, in the last paragraph of page 1, that SSMs do not perform well in dynamic token selection compared to Transformers. What we investigate in this paper is **how much the expression ability increases when SSMs are combined with FNNs** and **whether SSMs + FNNs can be as expressive as Transformers.** As a result, in Lemma 3.3, we showed that SSMs with FNN layers placed before and after them demonstrate equivalent capability to Transformers in dynamic token extraction. Based on this, we proved Theorem 3.1, 3.2 and Theorem 4.6, which shows that SSMs combined with FNNs are comparable to Transformers in various tasks. In our theorems, we provide the qualitative analysis on the sufficient number of parameters to solve the tasks, and show that those for SSMs + FNNs are similar to Transformers up to poly-log factor. Thus, our theoretical analysis provides the insights on how architectural design affect its expression ability.
>
> > Furthermore, an examination of optimal parameter allocation between SSM and FNN components under fixed budgets would be valuable.
>
> We appreciate your valuable feedback. While we provided a quantitative evaluation of the network size (e.g., layers, width, SSMs’ dimensions) that are sufficient for SSMs + FNNs to solve the task, this is just one particular parameter construction and is not definitive. Rigorous discussion on the optimal parameter allocation requires developing a different theoretical framework, which lies beyond the scope of this study. Practically, it would be effective to determine the network parameter settings through methods such as cross-validation.
>
> We would be happy to clarify any concerns or answer any questions that may come up during the discussion period. We would greatly appreciate it if you could consider increasing the score once all concerns have been resolved.

---

> > ### Comment · Reviewer_Byu5 · 2024-11-24
> >
> > Thank you for your detailed response. I appreciate your efforts to address most of my concerns, and I am currently going through the revised paper. One more question: do you have any empirical results for non-parametric regression as well?

---

> ### Author Response · Authors · 2024-11-30
>
> Thank you for your reply and question.
>
> > do you have any empirical results for non-parametric regression as well?
> >
>
> In addition to the experiments on two synthetic tasks, we conducted an experiment on nonparametric regression. The result is provided at the following URL:
>
> https://drive.google.com/file/d/1tPaay3awkh5TZPx9vog3pOwAHbOQ-U_A/view?usp=sharing
>
> In this experiment, we consider the following setting: let us consider the situation where inputs are given as sequences of words from the set $\mathcal W$. For each word $w\in\mathcal W$, we assign a value $r_w \in [0, 1]$. When the input sequence $X=[w_1, w_2, \ldots, w_L]$, the output $y$ is given as $y=f(\max(r_{w_1}, \ldots, r_{w_L}))$, where $f$ is a non-linear function. The function to be estimated $X\mapsto f(\max(r_{w_1}, \ldots, r_{w_L}))$ can be framed as a piecewise $\gamma$-smooth function. Indeed, the values assigned to the words can be seen as the importance of each token, and the input to the smooth function $f$ is the token with the highest importance.
>
> From the result, we can see that two-layer SSMs (with FNNs) perform better than one-layer SSMs, similar to the two synthetic tasks. This implies that it is essential to alternate multiple SSM layers and FNN layers to leverage dynamic token selection ability.

---

> > ### Author Response · Authors · 2024-12-03
> >
> > As the discussion period is ending soon, we wanted to follow up to check whether our response has adequately addressed your concerns for an improved score. If there are any remaining questions or concerns, we would be happy to provide further clarification.
> >
> > Thank you for your time and consideration.

---

> > > ### Comment · Reviewer_Byu5 · 2024-12-03
> > >
> > > Thank you for your clarification. While the paper has certain limitations in its practical application, I believe it provides a good starting point for the theoretical exploration of SSMs. Based on this, I have adjusted my score from 5 to 6.

---

### Official Review · Reviewer_8hKM · 2024-11-02

**Soundness:** 2
**Presentation:** 3
**Contribution:** 3
**Rating:** 6
**Confidence:** 3

**Summary:**

This paper explores the performance comparison between state space models and transformers in dynamic token selection. The study shows that the ability of SSMs combined with  FNNs to extract key tokens is comparable to that of transformers.

**Strengths:**

- The article presents a novel perspective by demonstrating that the combination of SSMs with nonlinear layers can simulate the dynamic token selection mechanism of transformers.
- It provides a detailed theoretical proof with a rigorous mathematical derivation process.
- The proposed theory is validated through experimental results.

**Weaknesses:**

- There is a lack of relevant experiments to validate the theoretical effectiveness, with the only experiments conducted on DNA base sequences. More experiments in settings such as LLMs are needed.
- The integration of SSMs and FNNs may increase the model's complexity, which could impact training and inference times.
- Although the article compares SSMs with transformers, it does not discuss comparisons with other sequence models, such as RNNs or LSTM.

**Questions:**

- Are there additional experimental results, for instance, would applying the improvements to Mamba enhance model performance?
- Are there any ablation study results demonstrating the effectiveness of adding FFNs?

---

> ### Author Response · Authors · 2024-11-21
>
> Thank you for your detailed and valuable feedback. We respond to specific concerns and questions below.
>
> > There is a lack of relevant experiments to validate the theoretical effectiveness, with the only experiments conducted on DNA base sequences. More experiments in settings such as LLMs are needed.
> >
>
> > Are there any ablation study results demonstrating the effectiveness of adding FFNs?
> >
>
> Thank you for your constructive suggestion. To empirically validate our theoretical findings that demonstrates the effectiveness of adding FNNs, we conducted experiments on input copying and associative recall. The results can be found in https://drive.google.com/file/d/1m9nL_6XnxESmOE-DxIbzlfQZuxjcrw5w/view?usp=sharing. Additionally, we described the details of the experiments in Appendix C.
>
> From the results, we can see that multi-layer SSM perform better than single-layer SSMs, particularly when the dimension of the hidden states is small and the FNN layers have sufficient expressive power. In our theory, we demonstrated that SSMs can exhibit dynamic token selection ability when it is paired with FNN layers before and after it. This implies that it is essential to alternate multiple SSM layers and FNN layers to leverage dynamic token selection ability. Thus, the experimental results of the experiments match our theory.
>
> > The integration of SSMs and FNNs may increase the model's complexity, which could impact training and inference times.
> >
>
> We appreciate your important comment. As we described in line 75, in practical applications, SSMs are combined with FNNs. Therefore, it is natural to analyze such a setting. Moreover, we showed that SSMs can solve tasks with O(poly log(L)) parameters, where L is a sequence length. This implies that, while combining SSMs with FNNs increases the computational cost, the increase does not offset the difference between the computational cost of SSMs, O(L) (or O(L log L)), and that of Transformers, O(L^2). Therefore, the advantage of SSMs compared to Transformers is still preserved.
>
> > Although the article compares SSMs with transformers, it does not discuss comparisons with other sequence models, such as RNNs or LSTM.
> >
>
> Thank you for your valuable feedback. As recent studies have intensely focused on SSMs as an alternative to Transformers, we intensively analyze SSMs.
>
> > Are there additional experimental results, for instance, would applying the improvements to Mamba enhance model performance?
> >
>
> We appreciate your important comment. The aim of this paper is to investigate whether SSMs combined with FNNs are comparable to Transformers. Exploring whether data-dependent filters improves model performance is beyond the scope of our research.
>
> We would be happy to clarify any concerns or answer any questions that may come up during the discussion period. We would greatly appreciate it if you could consider increasing the score once all concerns have been resolved.

---

> ### Comment · Reviewer_8hKM · 2024-11-26
>
> Thank you for your response. I have decided to maintain my rating score.

---

### Official Review · Reviewer_uwcy · 2024-11-04

**Soundness:** 3
**Presentation:** 3
**Contribution:** 2
**Rating:** 5
**Confidence:** 3

**Summary:**

This paper provides a theoretical investigation into the capabilities of State Space Models (SSMs) compared to Transformers in their ability to dynamically extract tokens based on the input. The authors prove that SSMs combined with Feedforward Neural Network (FNN) layers can achieve performance comparable to Transformers in three cases: input copying, associative recall, and nonparametric regression.

**Strengths:**

- The paper establishes a rigorous theoretical foundation, demonstrating that SSMs combined with FNNs can emulate the dynamic token selection mechanism of Transformers. This theoretical analysis bridges a gap in understanding the potential of SSMs in sequence modeling.

- The paper evaluates the proposed method across multiple tasks—input copying, associative recall, and nonparametric regression—showing that SSMs can achieve performance on par with Transformers.

**Weaknesses:**

- The paper is challenging to read due to dense technical details. It would be helpful if the authors could include a figure or remark after each theorem to illustrate the reasoning and provide an intuitive explanation.

- The input-independent SSMs used in this paper are somewhat outdated, given recent developments in selective SSMs (e.g., Mamba) and their variants. Additionally, the authors compare a combination of SSM and FNN with only the self-attention component of Transformers. This comparison may not be entirely fair, as the strength of a Transformer block lies in the combination of self-attention and FNN layers.

- A deeper comparison of the results presented here with those in [A] is necessary. In fact, the authors of [A] consider a more general version of selective SSMs, called Generalized State Space Models (GSSMs), which includes SSMs with FNN as a subclass. The results in [A] show that GSSMs do not outperform Transformers on tasks such as input copying, whereas the current paper suggests a different perspective.

Reference: [A] Jelassi, Samy, et al. "Repeat after me: Transformers are better than state space models at copying." arXiv preprint arXiv:2402.01032 (2024).

**Questions:**

- Could the authors clarify the differences between the results in this paper and those presented in [A] (see weaknesses)?

- Could the authors discuss the difficulty of generalizing these results to selective SSMs?

- In Line 405, could the authors explain the origin of the expression inside the big-O notation?

---

> ### Author Response · Authors · 2024-11-21
>
> Thank you for your detailed and valuable feedback. We respond to specific concerns and questions below.
>
> > The paper is challenging to read due to dense technical details. It would be helpful if the authors could include a figure or remark after each theorem to illustrate the reasoning and provide an intuitive explanation.
>
> Thank you for your important feedback. As for Section 3, as you pointed out, the intuition behind the theorems is insufficient. The theoretical essence lies in Lemma 3.3, which demonstrates that SSMs are capable of performing dynamic token selection. We revised the paper, and emphasized the connection between Theorem 3.1/3.2 and Lemma 3.3. Specifically, we add some explanation that follows Theorem 3.1/3.2 about how these theorems relate to Lemma 3.3.
>
> Regarding Section 4, as you pointed out, the explanation of the problem setup and its motivation was insufficient. In the revised version, at the beginning of Section 4, we included a detailed description of the motivation to consider this setup. Moreover, in Appendix E, we added some illustrations of the function class with some figures. Furthermore, to connect our theory and real-world applications, we added some examples of practical tasks that can be understood using our theory.
>
> > The input-independent SSMs used in this paper are somewhat outdated, given recent developments in selective SSMs (e.g., Mamba) and their variants.
>
> We appreciate your important point. As you pointed out, recently, various advanced methods have been proposed, including selective SSMs such as Mamba. However, our theory suggests that **even simple SSMs, which do not incorporate such development, are comparable to Transformers**, when combined with FNNs.
>
> Our main contribution in this paper is to provide a theoretical analysis from our novel perspective of "analyzing SSMs combined with FNNs". To this end, we established mathematical tools through the analysis for a simple architecture with conventional SSMs. Such a theoretical foundation for a simple model can be easily extended to complicated models that incorporate various recent developments. On the other hand, if we jump straight into analyzing complicated modern models, we would lose the essential capabilities of SSMs, which could hinder correct theoretical understanding. As you stated, as a future direction, it is essential to address the important research question of how modern SSMs provide advantages over older ones.
>
> > Additionally, the authors compare a combination of SSM and FNN with only the self-attention component of Transformers. This comparison may not be entirely fair, as the strength of a Transformer block lies in the combination of self-attention and FNN layers.
>
> As stated in the last paragraph on the first page, SSMs without FNN layers lack the capability of dynamic token selection. Therefore, the performance of SSMs without FNNs does not match that of the attention mechanism. Based on this fact, in this paper, instead of directly comparing SSMs and attention mechanism, we analyze the SSMs combined with FNNs. As a result, we show that SSMs’ expressive power is significantly enhanced when combined with FNN layers, making them comparable to the attention mechanism. Thus, our purpose is not to compare SSMs alone with the attention mechanism.
>
> > A deeper comparison of the results presented here with those in [A] is necessary. In fact, the authors of [A] consider a more general version of selective SSMs, called Generalized State Space Models (GSSMs), which includes SSMs with FNN as a subclass. The results in [A] show that GSSMs do not outperform Transformers on tasks such as input copying, whereas the current paper suggests a different perspective.
>
> > Could the authors clarify the differences between the results in this paper and those presented in [A] (see weaknesses)?
>
> We appreciate your insightful feedback. Jellasi et al. (2024) show that a single-layer SSM (with potential nonlinear operations before and after) cannot efficiently solve input copying. On the other hand, our discussion involves two-layer SSMs combined with FNNs, which does not contradict their results.
>
> In our theory, **it is essential that there are FNN layers before and after the second SSM layer**. In Lemma 3.3, we demonstrated that an structure of “FNN layer + SSM layer + FNN layer” can extract tokens where the inner product between the key and query is large, similar to a Transformer. Additionally, Jellasi et al. (2024) show that a two-layer Transformers can efficiently solve input copying. By applying Lemma 3.3 and replacing the first-layer attention with a single-layer SSM and the second-layer attention with FNN + SSM + FNN, we can derive our results. Since there was a lack of explanation, we have included additional description in the revised version.

---

> ### Author Response · Authors · 2024-11-21
>
> > Could the authors discuss the difficulty of generalizing these results to selective SSMs?
>
> Thank you for your valuable comment. While we primarily analyzed SSMs with data-independent filters, extending this to selective SSMs (i.e., SSMs with data-dependent filters) is straightforward. This is because selective SSMs are more expressive than SSMs with data-independent filters, and it is easy to construct selective SSMs that give the same output as SSMs with data-independent filters.
>
> > In Line 405, could the authors explain the origin of the expression inside the big-O notation?
>
> As written in line 177, the number of parameters in the model is expressed as $O(M(LW^2 + D^2))$. The number of parameters in line 405 is obtained by substituting the parameter count from Theorem 4.5 into this expression.
>
> We would be happy to clarify any concerns or answer any questions that may come up during the discussion period. We would greatly appreciate it if you could consider increasing the score once all concerns have been resolved.

---

> > ### Comment · Reviewer_uwcy · 2024-11-27
> >
> > Thank you, Authors, for your responses, which helped me understand the paper better. However, I still believe that comparing FFN+SSM+FFN with self-attention alone is unfair. As I mentioned before, the expressive power of Transformers comes from the combination of self-attention and FFN, not self-attention alone. It would be fair if a comprehensive comparison is done for FFN+SSM+FFN and self-attn+FFN. Overall, I would like to maintain my score.

---

> ### Author Response · Authors · 2024-11-30
>
> Thank you for your response. Since we have concerns that there might be misunderstandings, we would like to provide some remarks.
>
> We agree that the expressive power of Transformers comes from the combination of self-attention and FFN, and admit that the FNN layers would play important roles in enhancing the Transformers' capabilities. Moreover, we do not deny that there would be some tasks that Transformers can solve and SSMs combined with FNNs cannot.
>
> In this paper, **we do not intend to show that SSMs combined with FNNs can defeat Transformer/attention**. Rather, we claim that FNN + SSM + FNN is as expressive as the attention mechanism, **which is a key part of Transformer**. More concretely, through the analysis of the three tasks, we showed that SSMs combined with FNNs can perform dynamic token selection, which is an important capability of the attention mechanism. Therefore, we consider that the reviewer's criticism that "comparing FFN+SSM+FFN with self-attention alone is unfair" would not apply to our situation.
>
> Furthermore, we note that, by increasing width or depth of FNNs before and after SSM layers, we can make FNNs + SSMs + FNNs as expressive as FNNs + Attention + FNNs. In other words, our results show that, in solving the three tasks we consider, attention layers can be replaced by SSM layers without significant changes to the architecture (while hyper-parameters such as width and depth are changed), i.e., we can use an architecture alternating SSM layers and FNNs, instead of Transformers (alternating attention layers and FNNs).

---

### Official Review · Reviewer_Z8kP · 2024-11-07

**Soundness:** 4
**Presentation:** 4
**Contribution:** 2
**Rating:** 6
**Confidence:** 3

**Summary:**

This paper theoretically studies the expressive power of State Space Models (SSMs) and compares them with transformers. This papers consider three representative tasks in language modeling: input coping, associative recall, and non-parametric regression. For each task, this paper gives an architectural configuration on model depth, width, vocabulary size, weight norm/sparsity, etc. that can accomplish the task up to a given precision. In particular, the SSM architectures considered in this paper involve nonlinear layers, which are shown to improve the expressiveness of SSMs by enabling dynamic token selection.

**Strengths:**

+ The topic of this paper is both timely and important. SSMs have become increasingly popular and are widely used in many language and vision applications. Providing theoretical guarantees for these models is therefore crucial.

+ The paper is well-written, with precise and clear mathematical notations and definitions. While the claims are intricate, they are supported by comprehensive explanations and well-motivated interpretations.

+ The synthetic tasks considered in the theoretical analysis are popular yet insightful choices for analyzing language models. The results are solid and offer valuable insights into the comparative strengths of different sequence architectures. I reviewed the proofs for Theorems 3.1 and 3.2, and they appear correct.

**Weaknesses:**

- The assumptions in this paper appear to deviate from practical scenarios. Specifically, the paper assumes that the convolutional filter formed by SSMs has a defined "window size," whereas in practice, this window size is often as large as the sequence length. Additionally, it relies on a particular parameterization of SSMs (Lns 151–152) that has not been commonly adopted in real-world applications.

- While the piecewise smooth function class is convenient for theoretical comparison with prior work, it would be helpful to provide more background on this assumption. In particular, the authors should relate this function class to practical scenarios, explaining how the piecewise gamma-smooth function class aligns with real-world applications.

- The results in the paper seem more pertinent to understanding how combining convolution and nonlinear MLPs can perform synthetic sequence tasks while being not closely and directly related to the SSM models.

- The paper's proof techniques and settings closely resemble those in [1]. It remains unclear how SSMs differ from transformers in solving comparable tasks with similar complexity. Furthermore, Lemma 5.1 seems disconnected from the proofs of earlier results, failing to elucidate the underlying mechanism that distinguishes SSMs from transformers.

- A minor point: the analysis highlights a potential technical contribution, suggesting that nonlinear layers play an essential role in SSMs. This claim would be more convincing if supported by empirical evidence.

[1] Approximation and estimation ability of transformers for sequence-to-sequence functions with infinite dimensional input.

**Questions:**

1. In HiPPO theory [1], the authors demonstrate that certain parameterizations of A, B, C in SSMs can guarantee long-range modeling capabilities. Do the authors see any connection between the proposed parameterization (Lines 151–152) and the HiPPO framework, or similar properties in modeling long-range dependencies?

2. This paper focuses on input-independent SSMs. However, as shown in [2], data-dependent convolution can significantly enhance associative recall performance. Do the authors believe that the use of multi-layer SSMs with nonlinearity is sufficient to compensate this performance gap?

3. The proofs of Theorems 1 and 2 appear to follow a similar structure. Does this imply that the underlying network construction could be shared between the two results?

[1] HiPPO: Recurrent Memory with Optimal Polynomial Projections

[2] Zoology: Measuring and Improving Recall in Efficient Language Models

---

> ### Author Response · Authors · 2024-11-21
>
> Thank you for your detailed and valuable feedback. We respond to specific concerns and questions below.
>
> > The assumptions in this paper appear to deviate from practical scenarios. Specifically, the paper assumes that the convolutional filter formed by SSMs has a defined "window size," whereas in practice, this window size is often as large as the sequence length.
>
> Thank you for your important point. As you pointed out, in the typical use of SSMs, we use a window size that matches the sequence length. However, for mathematical simplicity, this paper adopts a finite window size.
>
> In the two synthetic tasks discussed in Section 3, we consider cases where the window size matches the sequence length, which aligns with typical usage. In the nonparametric regression task discussed in Section 4, since we assume an infinitely long sequence, the assumption of the finite window size deviates from standard SSM usage. However, with slight modifications to conventional SSMs, it is possible to reproduce the cases with a finite window size. We have add the details in Appendix B of the revised version.
>
> > Additionally, it relies on a particular parameterization of SSMs (Lns 151–152) that has not been commonly adopted in real-world applications.
>
> As noted in line 161 and Appendix A, the parameterization considered in this study can be reproduced using standard SSMs with the filter $CA^{t-n}B + D\delta_{t-n}$. In this research, we did not take into account specific constraints imposed on matrix $A$ in real-world applications. Instead, we found that $A$ can be constrained to a block diagonal matrix expressed as $A = \mathrm{diag}(D_1, \dots, D_k)$, where $D_1, \dots, D_k$ are $2 \times 2$ matrices (see Appendix A for details). Investigating how the structure of $A$ can be further restricted is left as future work. We have added this point to the limitation paragraph in the revised version.
>
> > While the piecewise smooth function class is convenient for theoretical comparison with prior work, it would be helpful to provide more background on this assumption. In particular, the authors should relate this function class to practical scenarios, explaining how the piecewise gamma-smooth function class aligns with real-world applications.
>
> We thank you for your insightful suggestion. As you pointed out, there were a lack of description on the significance of piecewise gamma smooth functions and the insights derived from the theory. We have added further clarification. Specifically, in the beginning of Section 4, we included an additional explanation on the motivation to consider the piecewise $\gamma$-smooth function. Moreover, in Appendix E, we added a detailed explanation with figures to illustrate the motivation for piecewise gamma smooth functions.
>
> > The paper's proof techniques and settings closely resemble those in [1]. It remains unclear how SSMs differ from transformers in solving comparable tasks with similar complexity. Furthermore, Lemma 5.1 seems disconnected from the proofs of earlier results, failing to elucidate the underlying mechanism that distinguishes SSMs from transformers.
>
> We appreciate your valuable feedback. The primary difference between Takakura & Suzuki (2023) and our study lies in showing how SSMs perform dynamic token selection. Transformers leverage their attention mechanism’s ability to “prioritize the parts where the dot product between key and query is large” to dynamically adjust the tokens extracted based on the input. The proof in Takakura & Suzuki (2023) heavily relies on this capability of the attention mechanism. In contrast, SSMs do not explicitly have a mechanism with such capabilities. However, in Lemma 5.1 (Lemma 3.3 in the revised version), we demonstrated that a SSM layer combined with FNN layers before and after it can prioritize the tokens with high dot product between key and query, similar to the attention mechanism.
>
> Although the significance of Lemma 5.1 was previously explained at the beginning of Section 5, as you pointed out, the explanation of its relevance to the three tasks was insufficient. This may have made our contribution less clear. We revised the paper, and change the position of Lemma 5.1 (currently Lemma 3.3) to emphasize its importance. Moreover, after Theorem 3.1, 3.2 and 4.5, we add some description about the connection between Lemma 3.3 and the theoretical analysis of three tasks.

---

> ### Author Response · Authors · 2024-11-21
>
> > A minor point: the analysis highlights a potential technical contribution, suggesting that nonlinear layers play an essential role in SSMs. This claim would be more convincing if supported by empirical evidence.
>
> Thank you for your constructive suggestion. To empirically validate our theoretical findings, we conducted experiments on input copying and associative recall. The results can be found in https://drive.google.com/file/d/1m9nL_6XnxESmOE-DxIbzlfQZuxjcrw5w/view?usp=sharing. Additionally, we described the details of the experiments in Appendix C.
>
> From the results, we can see that multi-layer SSMs perform better than single-layer SSMs when the dimension of the hidden states are small and the FNN layers have sufficient expressive power. In our theory, we demonstrated that FNN + SSM + FNN performs dynamic token selection similar to Transformers. This implies that it is essential to alternate multiple SSM layers and FNN layers to leverage dynamic token selection ability. Thus, the results of the experiments match our theory.
>
> > 1. In HiPPO theory [1], the authors demonstrate that certain parameterizations of A, B, C in SSMs can guarantee long-range modeling capabilities. Do the authors see any connection between the proposed parameterization (Lines 151–152) and the HiPPO framework, or similar properties in modeling long-range dependencies?
>
> We do not intend to associate our parametrization with the HiPPO framework. Our parametrization is specialized in replacing a attention layer with a SSM layer combined with the FNN layers before and after it. We leave it as future work to investigate whether parametrization like HiPPO, which is applied in real-world scenarios, is beneficial for the tasks, similar to our parametrization.
>
> > 2. This paper focuses on input-independent SSMs. However, as shown in [2], data-dependent convolution can significantly enhance associative recall performance. Do the authors believe that the use of multi-layer SSMs with nonlinearity is sufficient to compensate this performance gap?
>
> Thank you for your important question. We focused our analysis on the expression ability and demonstrated that SSMs combined with FNNs are comparable to Transformers in this aspect. We believe that we cannot rule out the possibility of a performance gap between SSMs with and without data-dependent convolution based on this result alone. The main reason is that we have not analyzed the optimization dynamics of the models. When models are optimized using gradient-based methods, SSMs with data-dependent filters may possess better inductive biases than SSMs with data-independent filter.
>
> We also believe that the approximation and estimation theories presented in this paper could contribute to the development of models with improved inductive biases.
>
> > 3. The proofs of Theorems 1 and 2 appear to follow a similar structure. Does this imply that the underlying network construction could be shared between the two results?
>
> As you pointed out, in Theorems 3.1 and 3.2, the proofs are conducted using similar parameterizations. This would suggest that the network construction for solving these tasks could be shared. Indeed, both proofs use the structure presented in Lemma 3.3 to prioritize the tokens with high query-key inner products.
>
> We would be happy to clarify any concerns or answer any questions that may come up during the discussion period. We would greatly appreciate it if you could consider increasing the score once all concerns have been resolved.

---

> ### Comment · Reviewer_Z8kP · 2024-12-03
>
> I appreciate the authors' detailed response. They did an excellent job of clearly defining the scope of their paper and elucidating the theoretical framework. While my concern regarding the gap between the theoretical and practical settings remains (albeit a minor one), I maintain my rating and remain positive about the quality of this work.

---

### Author Response · Authors · 2024-11-20
**For All the Reviewers**

We appreciate the detailed feedback and suggestions for improving our paper. We incorporated all the comments into the revised version. As for the changes in the revised paper, please refer to the blue text in the current PDF. Below, we provide some information we would like to share with all reviewers.

1. The reviewers pointed out that the intuition behind our theory is not clear. In this paper, for all the task we consider (input copying, associative recall, and nonparametric regression), the essence of our theory lies in the fact that SSMs, when combined with preceding and following FNN layers, possess a **dynamic token selection ability**. In other words, like Transformers, SSMs can determine positions of a sequence to focus on depending on the input. **This is formally demonstrated in Lemma 3.3** (Lemma 5.1 before revision). In this lemma, we show that FNN + SSM + FNN (i.e., SSM layer with FNN layers applied before and after) can extract tokens with high inner product of keys and queries, just as Transformers do. Using this lemma, we proved the theories for the three tasks (Theorems 3.1, 3.2, and 4.5).

    Thus, Lemma 3.3 (formerly Lemma 5.1) represents our most significant contribution, but it may have been unclear in the original version. To emphasize the importance of Lemma 3.3, we moved its location from Section 5 to the end of Section 3. Additionally, we clarified the connection between this lemma and each theorem by explicitly describing its relevance following Theorems 3.1, 3.2, and 4.5.

2. The reviewers also pointed out that our explanation of the motivation for considering nonparametric regression and the piecewise $\gamma$-smooth function class, as well as the intuitive description of their definitions, was insufficient.

    To address the reviewers' concerns, at the beginning of Section 4, we added a brief explanation of the motivation for considering piecewise $\gamma$-smooth functions. Additionally, to make the definition of the function class more intuitive, **we included a graphical explanation in Appendix E**. Furthermore, we provided **examples of real-world tasks that can be modeled as piecewise $\gamma$-smooth functions**.

3. The reviewers suggested that we should empirically validate our theory through experiments. **We conducted experiments** comparing a single-layer SSM with a multi-layer SSM on the input copying task and the associative recall task. The experimental results can be accessed via the following URL. Additionally, the details of the experiments are provided in Appendix C.

    https://drive.google.com/file/d/1m9nL_6XnxESmOE-DxIbzlfQZuxjcrw5w/view?usp=sharing

    The graphs plot the accuracy as the dimensionality of the hidden state varies. The graph on the left corresponds to input copying, while the graph on the right corresponds to associative recall. The number in parentheses following "FNN" represents the depth of the FNN layer. The results indicate that, when the dimension of the hidden state is small and the FNN layers have sufficient expressive power, the multi-layer SSM outperforms the single-layer SSM. **This aligns with our theoretical observation**, which states that alternating multiple SSM and FNN layers enhances the dynamic token selection ability.

We also respond to the comments from each reviewer. If there are further questions and comments, we would be pleased to discuss them during the discussion period.

---

### Meta-Review · Area_Chair_vcd6 · 2024-12-19

**Metareview:**

This paper provides a theoretical analysis of the expressive capabilities of modern neural net architectures relying on State Space Models (SSMs) when combined with dense layers, comparing their performance to Transformers. The authors provide theoretical guarantees that SSMs, when integrated with nonlinear layers, can achieve comparable performance to Transformers. The work addresses an important gap in the theoretical understanding of increasingly popular class of SSM-based neural architectures and demonstrates their potential as competitive sequence modelling architectures.

The paper makes several notable contributions. It provides rigorous analysis showing that SSMs with nonlinear layers can emulate dynamic token selection, a hallmark of Transformer architectures. The selection of tasks is well-motivated, offering concrete benchmarks that test fundamental aspects of sequence modeling. The authors also provide explicit configurations (model depth, width, sparsity, and weight norms) necessary to achieve precision in these tasks.

The paper is recommended for acceptance, as it addresses a timely and important topic in sequence modeling. State Space Models (SSMs) are increasingly adopted in practice, yet their theoretical underpinnings remain limited. This work bridges a significant gap by proving that SSMs, when integrated with fully connected neural networks (FNNs), can achieve expressiveness comparable to Transformers in tasks requiring dynamic token selection. The paper provides meaningful advancements in the understanding and application of SSMs as a component of modern neural architectures.

**Additional Comments On Reviewer Discussion:**

During the rebuttal phase, there was productive and detailed discussion between the authors and reviewers regarding the modeling choices and the specifics of the neural architectures presented in the paper. Several reviewers requested empirical validation of certain theoretical claims. In response, the authors conducted additional experiments to address these concerns, incorporating the results into the appendix.

---

### Decision · Program_Chairs · 2025-01-22

Accept (Poster)